# Visual timing-tuned responses in human association cortices and response dynamics in early visual cortex

Evi Hendrikx [1,3], Jacob M. Paul [1,2,3], Martijn van Ackooij[1], Nathan van der Stoep [1] & Ben M. Harvey [1✉]

Quantifying the timing (duration and frequency) of brief visual events is vital to human perception, multisensory integration and action planning. Tuned neural responses to visual event timing have been found in association cortices, in areas implicated in these processes. Here we ask how these timing-tuned responses are related to the responses of early visual cortex, which monotonically increase with event duration and frequency. Using 7-Tesla functional magnetic resonance imaging and neural model-based analyses, we find a gradual transition from monotonically increasing to timing-tuned neural responses beginning in the medial temporal area (MT/V5). Therefore, across successive stages of visual processing, timing-tuned response components gradually become dominant over inherent sensory response modulation by event timing. This additional timing-tuned response component is independent of retinotopic location. We propose that this hierarchical emergence of timing-tuned responses from sensory processing areas quantifies sensory event timing while abstracting temporal representations from spatial properties of their inputs.

[1] Experimental Psychology, Helmholtz Institute, Utrecht University, Heidelberglaan 1, Utrecht 3584 CS, Netherlands. [2] Melbourne School of Psychological Sciences, University of Melbourne, Redmond Barry Building, Parkville 3010 VIC, Australia. [3] These authors contributed equally: Evi Hendrikx, Jacob M. Paul. ✉email: b.m.harvey@uu.nl

Quantifying spatial and temporal information of events is vital to perceiving and interacting with our environment. For example, accurately determining the duration and frequency of sub-second events is crucial for multisensory integration and motor action planning[1,2]. Currently, predominant models for the brain's estimation of an event's timing rely on central neural processes responding to passing time, like ticking pacemakers or the dynamics of neural activity[3–7]. However, following converging behavioral, computational and neuroimaging results[8–16], we hypothesized that the brain may derive a representation of event timing from the dynamics of neural responses to that event in sensory processing areas, rather than relying on specialized central pacemakers or processes.

Visual responses are strongly modulated by event timing[17,18], but it remains unclear how we quantify the timing of sensory events from the neural responses in sensory cortices. Early visual cortical responses increase monotonically but sub-linearly with event duration and frequency (and therefore decrease with event period, i.e. 1/frequency). This can be described in terms of the summed amplitude of transient and sustained neural responses respectively[17,18], for a stimulus of fixed strength (e.g. fixed contrast). But these early visual response components' amplitudes both also increase with stimulus strength, so don't unambiguously represent event timing. Nevertheless, their amplitudes provide a signal from which event timing can be quantified: the ratio of sustained to transient response amplitude gives event duration (if both scale similarly with stimulus strength), while the sum of transient responses over time is proportional to event frequency (at fixed contrast or after contrast normalization). Following our hypothesis that the brain's representation of a visual event's timing emerges from response dynamics in retinotopically organized early visual areas, we predicted that these monotonic responses would be restricted to the stimulus's retinotopic location.

Beyond early visual areas, an extensive network throughout the human association cortices shows visual timing-tuned responses, with maximum response amplitudes at specific preferred durations (the time from event onset to offset) and periods (the time between repeating event onsets; i.e., 1/frequency)[19]. Tuned neural responses are common throughout sensory and cognitive processing, and are closely linked to the perception of many stimulus features[20,21], including visual motion[22], somatosensory vibrational frequency[23], and other quantities like numerosity[24–26] (recently reviewed by ref. [27]). Even for visual event timing, changes in duration-tuned responses and duration perception are linked[28–30]. These timing-tuned responses are topographically mapped, such that the preferred durations and periods of neural populations gradually progresses across the cortical surface[19,31]. These maps are found in areas that also show tuning and mapping for other quantities, including visual numerosity, visual object size, and haptic numerosity[32–34]. These areas have a more abstracted representation of multiple quantities that likely allows their neural responses to interact regardless of the spatial, temporal, or sensory origins[33,34], potentially underlying perceptual interactions[35,36]. As these quantity-tuned responses are topographically mapped by the state of the stimulus quantity, rather than its position, we predicted that timing-tuned responses would not be restricted to the stimulus' position on the sensory organ, but encoded in an abstracted, quantity-based frame of reference.

Regarding the computation of timing-tuned responses, it seems likely that they are derived from the dynamics of early sensory neural responses. However, it is unclear how and where tuned representations to visual event timing are computed and transformed from monotonic responses in early visual cortices. Computational models for other quantities suggest that excitatory and inhibitory non-linear monotonic responses can be compared

to give numerosity-tuned responses[37,38]. Indeed, monotonic responses of early visual areas that closely follow numerosity are likely to be transformed into tuned responses to numerosity in lateral occipital areas[39]. A range of motor event timing-dependent neural response profiles, from timing-tuned to monotonic, has been described in macaque premotor cortex[40,41]. These tuned and monotonic neurons are intermixed and together transform from abstract, timing-tuned response to action-triggering thresholds to determine motor action timing.

Here we ask whether and how monotonic and tuned neural responses to sub-second visual event timing are related throughout the brain's hierarchy of both timing maps and visual field maps. We further ask how both monotonic and tuned responses to timing are related to visual position preferences. We answer these questions by reanalyzing ultra-high-field (7T) functional magnetic resonance imaging (fMRI) data that was acquired during the presentation of repetitive visual events that gradually varied in event duration and/or period in our previous study[19]. We compare the fits of monotonically increasing and tuned neural response models throughout the brain and investigate how these relate to visual spatial selectivity throughout the visual field map hierarchy. We find that through this visual hierarchy timing-tuned response components gradually become dominant over monotonic responses. This additional timing-tuned response component is independent of retinotopic location

## Results

**Transition from monotonic to tuned responses to timing along the visual field map hierarchy**. We first asked whether and how monotonic and tuned neural responses to sub-second visual event timing are related throughout the hierarchy of both timing maps and visual field maps. We hypothesized that visual event timing-tuned responses are likely to be derived from the inherent modulation of visual responses by event timing: monotonic increases in early visual responses with event duration and frequency. This predicts a transition from monotonic to tuned response functions through the visual field map hierarchy.

We focus on events with durations and periods from 50 to 1000 ms for several reasons. First, it is feasible to sample this limited range in a single experiment. Second, previous studies using fMRI[17–19,28–31], animal neurophysiology[40,41], and psychophysics[11–15] have used similar ranges, allowing us to relate our findings to previous literature. Finally, these studies have shown both monotonic[17,18,40] and tuned[19,31,41] neural responses within this range.

To determine whether each visual field map's responses followed monotonic or tuned functions, we recorded neural responses to visual events of variable timing. During fMRI scanning, we showed a circle repeatedly appearing and disappearing (visual events). Because of fMRI's slow time scale, we varied event timing gradually in both duration and/or frequency, allowing the resulting measurements to distinguish response amplitudes to different event timings. Both event duration and period ranged from 50 to 1000 ms in 50 ms steps, and any event's duration was always less than its period so that the event ended before the next began and there was never more than one event happening. Event duration changes are inevitably coupled with changes in either mean display luminance (if events begin at regular intervals) or event period (if each event is immediately followed by another). We therefore characterized four stimulus configurations (Fig. 1b and Supplementary Movie 1) to distinguish between responses to event duration, event period and display luminance. The constant luminance configuration matched duration and period so an event was always ongoing. The constant duration configuration fixed all events' durations at

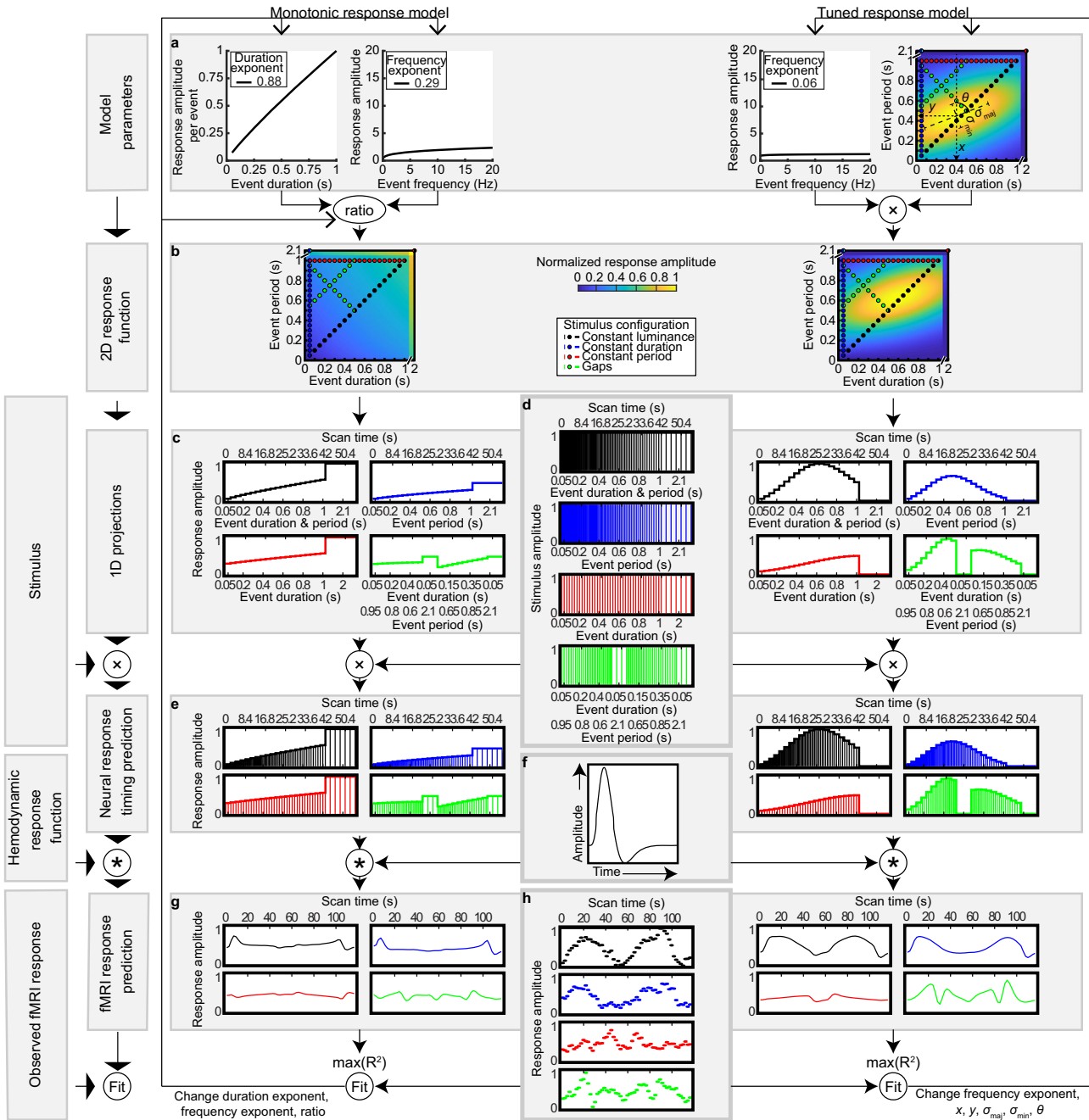

**Fig. 1 Monotonic and tuned response model fitting procedures. a** Monotonic response model (left) and tuned response model (right). In each case, a compressive exponent parameter captures a non-linear scaling with increases in event duration (left) or frequency (middle left and middle right). The monotonic response predictions for event duration and frequency changes are combined as a weighted sum. **b** Together, these make a prediction of the per-event response amplitude for every event timing shown in the stimulus (dots). **c** This gives a prediction of the response amplitudes that would be seen for each stimulus condition for a specific candidate set of response model parameters. **d** The times of event offsets in each stimulus condition, which vary in frequency. **e** Combining the per-event response amplitudes predicted by the response models (in **c**) and the times of the event offsets (in **d**) gives neural response amplitude predictions for each condition, equal to the amount of color under the curve. **f** The hemodynamic response function. **g** The neural response amplitude predictions (in **e**) are convolved with a hemodynamic response function (in **f**) to get the predicted fMRI response time courses. Note here that these predictions are for both ascending and descending sweeps of duration and/or period, while neural response predictions (in **c** and **e**) are for ascending sweeps only. **h** The recorded fMRI response time course for an example voxel. The monotonic and tuned predictions were compared to the recording from each voxel. The free parameters of both the monotonic and tuned response models were found that maximized the correlation ($R^2$) between predicted and measured fMRI response time courses. For the monotonic response model these parameters were the compressive exponents on duration and frequency components, and the weighting of these two components (ratio). This ratio is the ratio of the scaling factors for the two components in a general linear model. For the tuned response model, the free parameters were the compressive exponent on event frequency, preferred duration ($x$), preferred period ($y$), response function extent along its major ($\sigma_{maj}$), and minor ($\sigma_{min}$) axes, and major axis orientation ($\theta$).

50 ms while their periods varied. The constant period configuration fixed all events' periods at 1000 ms while their durations varied. Finally, the gaps configuration varied both duration and period to sample timings absent in other configurations. Participants made no duration or period judgments but reported when white circles were presented rather than black (performance > 80%). This happened pseudo-randomly, equally frequently for all timings.

This stimulus set let us disambiguate monotonic and tuned response models by testing these models' predictions against the response amplitudes seen for many combinations of event duration and frequency. For each voxel, we used both monotonic and tuned response models (Fig. 1 and Supplementary Fig. 1) to predict the time course of response amplitudes to the presented event timings. Each of these response models uses a set of candidate response model parameters to predict the neural response amplitudes to every combination of event duration and frequency (Fig. 1a, b). At every event's offset, we evaluated this prediction (Fig. 1c, d) to generate a prediction of the neural response time course that would be seen for this set of candidate response model parameters (Fig. 1e). Convolving this with a hemodynamic response function (Fig. 1f) predicted an fMRI response time course (Fig. 1g), which we correlated to the measured responses (Fig. 1h). We repeated this for a large set of candidate response model parameters to find those parameters that most closely predict each voxel's response time course. We then compared the goodness of fit of the resulting monotonic and tuned response models on cross-validated data to distinguish between these models' performance despite differences in model complexity.

We validated our procedure's ability to distinguish monotonic and tuned responses by generating simulated responses[42] that followed either monotonic or tuned response functions with a broad and homogenous range of response function parameters and different levels of noise (Supplementary Fig. 2). This showed that monotonic responses were almost always correctly identified when their variance explained ($R^2$) was above 0.2 in the data on which the models were fit. Tuned responses were also reliably identified using the same variance explained threshold of 0.2. In both cases, responses were more likely to be classified as monotonic where the best fitting tuned response function approximated a monotonic response function: where its preferred duration or period estimate was outside the presented range, its extent was very large, or its compressive exponent on event frequency was high.

Our subsequent analyses, therefore, excluded voxels where the variance explained was 0.2 or less for both models in the data on which the models were fit (to avoid selecting voxels using the cross-validated fit values we compared). This excludes many voxels in each visual field map (often far from the retinotopic location of the stimulus) where responses do not systematically vary with timing, while including voxels where responses were convincingly modulated by event timing. We also classified voxels where tuned models gave preferred duration or period estimates outside the presented range (i.e. below 60 ms or above 990 ms) as showing monotonic responses, setting their variance explained to zero to avoid using a tuned response model that approximates a monotonic response function.

We then compared these models' cross-validated fits, in data independent from that on which model parameters were fit. We found a clear modulation of responses by visual event timing throughout the visual field map hierarchy, with the monotonic response model best capturing responses of occipital areas and the tuned response model best capturing most of the responses in the parietal and frontal areas (Fig. 2a and Supplementary Fig. 5).

We used standard visual field mapping and population receptive field (pRF) modelling procedures[43,44] to determine the preferred visual field positions of every voxel. We grouped these voxels into visual field maps (Fig. 2 and Supplementary Figs. 3, 4), then took the median model fit (cross-validated variance explained) in each visual field map and used a three-factor ANOVA to assess how model fits differed between visual field maps, models and participants. Model fits ($R^2$) differed between visual field maps ($F_{(20, 1109)} = 15.96$, $p < 0.001$, $\eta_p^2 = 0.23$) but not between models ($F_{(1, 1128)} = 0.15$, $p = 0.695$, $\eta_p^2 = 0.00$). However, there was an interaction between visual field map and model ($F_{(20, 1109)} = 20.93$, $p < 0.001$, $\eta_p^2 = 0.28$). Post hoc multiple comparisons demonstrated that early visual and lateral visual field maps had significantly better fits for the monotonic response model (V1, V2, V3, hV4, LO1, LO2, and TO1; Fig. 3, Table 1). In contrast, the responses of several parietal and frontal visual field maps were better captured by the tuned response model than the monotonic response model (IPS1, IPS2, IPS3, IPS4, IPS5, sPCS1, sPCS2, and iPCS; Fig. 3, Table 1). Ventral stream visual field maps (VO1, VO2, and PHC), and visual field maps that lay in between the early visual (monotonic) and parietal/frontal (tuned) maps (TO2, V3AB, and IPS0) showed no significant difference between the fit of the monotonic and tuned response model (Fig. 3 and Table 1). Similar results were found at variance explained thresholds of 0, 0.2, and 0.4 (Supplementary Figs. 6 and 7 and Supplementary Tables 1–6).

Strikingly, the amount of variance explained (i.e. the goodness of model fit) by both models increased along the early visual and dorsal stream visual field map hierarchy from V1 to TO1 (hMT+) (Fig. 3a), after which the tuned response model fit better. Tuned response model fits then remained relatively stable across frontal and parietal visual field maps while monotonic response model fits declined for more anterior visual field maps. As a result of these two changes, the difference in model fits (i.e. the additional variance explained by a timing-tuned response component) increased from occipital to frontal visual field maps (Fig. 3b).

In the ventral stream visual field maps (VO1, VO2, and PHC) the maximum variance explained by either response model was lower than in other visual field maps, and few hemispheres showed any voxels reaching the variance explained threshold of 0.2. As such, the responses of these visual field maps are not convincingly affected by visual event timing. We have therefore excluded them from further analysis.

**Changes in response model parameters between visual field maps.** Both response models can capture very different relationships between event timing and response amplitude using different free parameters. Differences between visual field maps in monotonic response model parameters[17,18] and between timing maps in tuned response model parameters[19] have been described in previous studies, tuned model parameters in the visual field maps have not been described. We, therefore, grouped the voxels in each visual field map, took the median model parameter in each visual field map, and used a two-factor ANOVA to assess how these parameters differed between visual field maps and participants. Here we do not analyze how response model parameters change within visual field maps as the distribution of timing response function preferences is better characterized with a timing map, which has a complete set of response properties in a particular region of the brain. We have done this elsewhere[19]. The set of voxels within a visual field map represents a variable sample from the set within a complete timing map.

In the monotonic response model, the ratio of the response amplitudes of the duration component divided by the frequency

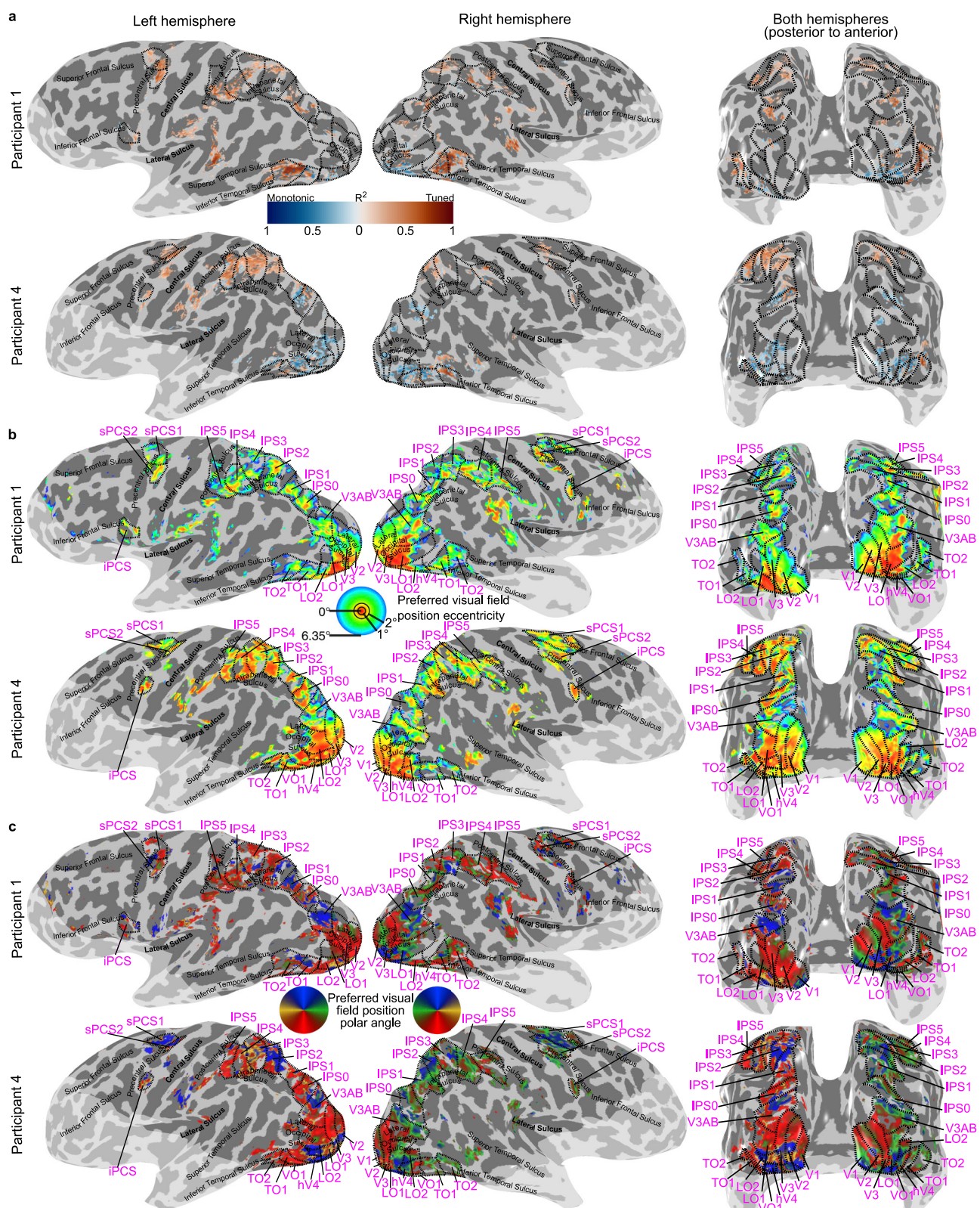

**Fig. 2 Progressions of best-performing model on timing-selective responses and visual field position preferences. a** Cross-validated variance explained of the best-performing model within each voxel is displayed (averaged across both cross-validation splits, for voxels with a variance explained above 0.2 for the best fitting model). The variance explained of the monotonic response model is given in blue and of the tuned response model is given in red. The intensity of the color relates to the magnitude of the variance explained. **b** Eccentricity preferences for voxels with over 0.1 variance explained by the response model. **c** Polar angle preferences for voxels with over 0.1 variance explained by the response model. Visual field map borders are shown as black dashed lines, and named in magenta text. The light shaded region is outside the fMRI recording volume. See also Supplementary Figs. 3–5. Images in **b** and **c** adapted from Harvey et al., 2020[19].

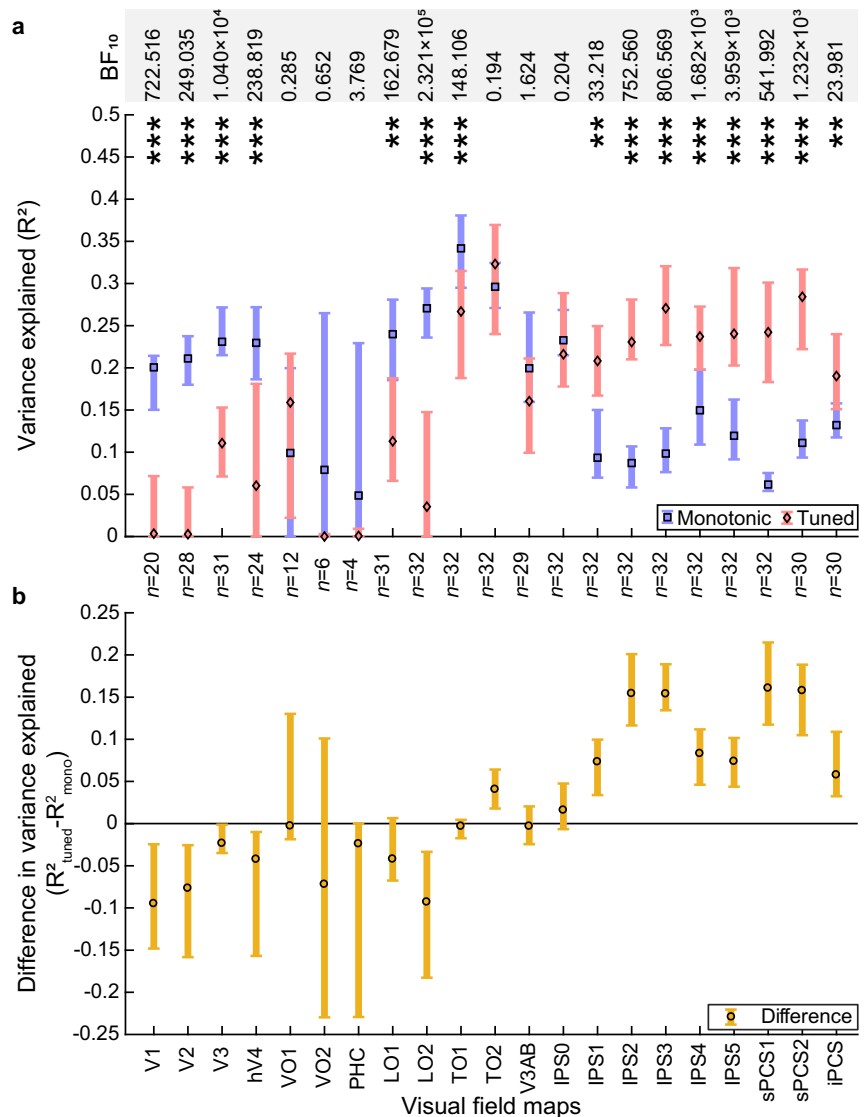

**Fig. 3 Comparison of fits of monotonic and tuned response models in each visual field map. a** Cross-validated variance explained by monotonic and tuned response models. **b** Difference in cross-validated variance explained between models (tuned – monotonic) progressively changes from monotonic to tuned up the visual hierarchy. Markers show the median variance explained of each visual field map, error bars show the 95% confidence interval, computed from 1000 bootstrap iterations. Only the voxels that had a variance explained above 0.2 for the best fitting model in the data on which the models were fit are included. Significant differences between the models (two-sided Wilcoxon signed-rank test, false discovery rate (FDR) corrected for multiple comparisons): $*p < 0.05$ $**p < 0.01$, and $***p < 0.001$. $n$ indicates the number of hemispheres included in the comparison in each visual field map (for both panels). Bayes factors ($BF_{10}$) for each paired comparison are presented in the top row. Source data are provided as a Source Data file.

component differed significantly between visual field maps ($F_{(17, 406)} = 5.52$, $p < 0.001$, $\eta_P^2 = 0.19$) (Fig. 4a). Post hoc multiple comparisons showed this ratio progressively decrease from early to later visual field maps. This is consistent with previous reports[17], where this result is proposed to reflect a faster decrease in the amplitudes of sustained than transient neural response components through the visual processing hierarchy. The second model parameter, the compressive exponent on event duration, also differed significantly between visual field maps ($F_{(17, 451)} = 1.95$, $p = 0.013$, $\eta_P^2 = 0.07$) (Fig. 4b), though there was no evidence of a systematic progression through the visual hierarchy. The final parameter, the compressive exponent on event frequency, also differed significantly between visual field maps ($F_{(17, 451)} = 11.51$, $p < 0.001$, $\eta_P^2 = 0.31$) (Fig. 4c). Post hoc multiple comparisons showed this parameter progressively decreased from early to later visual field maps (consistent with previous reports[18]), particularly maps where tuned models

predict responses better than monotonic models (Fig. 3). Therefore, up the visual hierarchy responses to repeated events are progressively integrated until response amplitudes no longer increase monotonically with event frequency and instead show tuned responses that peak at specific frequencies.

Similarly, for the tuned response model, the compressive exponent on event frequency differed between visual field maps ($F_{(17, 520)} = 49.89$, $p < 0.001$, $\eta_P^2 = 0.62$) (Fig. 4d), again decreasing from early to later visual field maps where tuned responses become dominant. This decrease is consistent with a decrease among the timing map hierarchy[19], which largely overlaps with the visual field map hierarchy from LO1 onwards. We found that the median preferred event duration and period differed significantly between visual field maps (duration: $F_{(17, 520)} = 3.17$, $p < 0.001$, $\eta_P^2 = 0.10$; period: $F_{(17, 520)} = 2.29$, $p = 0.003$, $\eta_P^2 = 0.07$) (Fig. 4e, f), though without a systematic hierarchical progression in either case, again in agreement with a

**Table 1 Descriptive and test statistics of the paired model comparisons within visual field maps.**

| Visual field map | Monotonic fit ($R^2$) | Tuned fit ($R^2$) | $Z$ statistic | Effect size ($r$) | $p$ value | $BF_{10}$ | Posterior distribution |
|---|---|---|---|---|---|---|---|
| V1 | 0.20 [0.15, 0.21] | 0.00 [0.00, 0.07] | −3.58 | −0.80 | $6 \times 10^{-4}$ | 722.516 | 1.394 [0.693, 2.079] |
| V2 | 0.21 [0.18, 0.24] | 0.00 [0.00, 0.06] | −4.01 | −0.76 | $1 \times 10^{-4}$ | 249.035 | 1.159 [0.640, 1.631] |
| V3 | 0.23 [0.22, 0.27] | 0.11 [0.07, 0.15] | −4.74 | −0.85 | $1 \times 10^{-5}$ | $1.040 \times 10^{4}$ | 1.937 [1.099, 2.671] |
| hV4 | 0.23 [0.19, 0.27] | 0.06 [0.00, 0.18] | −3.57 | −0.73 | $6 \times 10^{-4}$ | 238.819 | 0.969 [0.440, 1.568] |
| VO1 | 0.10 [0.00, 0.20] | 0.16 [0.02, 0.22] | 0.00 | −0.00 | 1 | 0.285 | 0.003 [−0.510, 0.509] |
| VO2 | 0.08 [0.00, 0.26] | 0.00 [0.00, 0.00] | −1.15 | −0.47 | 0.290 | 0.652 | 0.367 [−0.339, 1.248] |
| PHC | 0.05 [0.00, 0.23] | 0.00 [0.00, 0.01] | −1.83 | −0.91 | 0.084 | 3.769 | 1.616 [−0.035, 10.825] |
| LO1 | 0.24 [0.19, 0.28] | 0.11 [0.07, 0.19] | −3.00 | −0.54 | 0.003 | 162.679 | 0.665 [0.301, 1.061] |
| LO2 | 0.27 [0.24, 0.29] | 0.04 [0.00, 0.15] | −4.69 | −0.83 | $1 \times 10^{-5}$ | $2.321 \times 10^{5}$ | 1.523 [0.942, 2.080] |
| TO1 | 0.34 [0.29, 0.38] | 0.27 [0.19, 0.31] | −3.61 | −0.64 | $6 \times 10^{-4}$ | 148.106 | 0.821 [0.389, 1.303] |
| TO2 | 0.30 [0.27, 0.32] | 0.32 [0.24, 0.37] | −0.06 | −0.01 | 1 | 0.194 | 0.036 [−0.295, 0.354] |
| V3AB | 0.20 [0.16, 0.27] | 0.16 [0.10, 0.21] | −2.02 | −0.38 | 0.057 | 1.624 | 0.369 [0.008, 0.743] |
| IPS0 | 0.23 [0.21, 0.27] | 0.22 [0.18, 0.29] | −0.67 | −0.12 | 0.554 | 0.204 | 0.066 [−0.263, 0.400] |
| IPS1 | 0.09 [0.07, 0.15] | 0.21 [0.17, 0.25] | 3.22 | 0.57 | 0.002 | 33.128 | −0.652 [−1.051, −0.274] |
| IPS2 | 0.09 [0.06, 0.11] | 0.23 [0.21, 0.28] | 4.67 | 0.83 | $1 \times 10^{-5}$ | 752.560 | −1.475 [−1.992, −0.749] |
| IPS3 | 0.10 [0.08, 0.13] | 0.27 [0.23, 0.32] | 4.77 | 0.84 | $1 \times 10^{-5}$ | 806.569 | −1.579 [−2.124, −0.803] |
| IPS4 | 0.15 [0.11, 0.20] | 0.24 [0.20, 0.27] | 4.41 | 0.78 | $3 \times 10^{-5}$ | $1.682 \times 10^{3}$ | −1.269 [−1.868, −0.734] |
| IPS5 | 0.12 [0.09, 0.16] | 0.24 [0.20, 0.32] | 4.81 | 0.85 | $1 \times 10^{-5}$ | $3.959 \times 10^{3}$ | −1.848 [−2.391, −0.892] |
| sPCS1 | 0.06 [0.05, 0.08] | 0.24 [0.18, 0.30] | 4.94 | 0.87 | $1 \times 10^{-5}$ | 541.992 | −2.114 [−3.238, −1.023] |
| sPCS2 | 0.11 [0.09, 0.14] | 0.28 [0.22, 0.32] | 4.62 | 0.84 | $1 \times 10^{-5}$ | $1.232 \times 10^{3}$ | −1.694 [−2.422, −0.932] |
| iPCS | 0.13 [0.12, 0.16] | 0.19 [0.15, 0.24] | 3.18 | 0.58 | 0.002 | 23.981 | −0.655 [−1.104, −0.246] |

Data are median cross-validated variance explained [95% confidence interval of the median computed from 1000 bootstrap iterations]. These are the outcomes of a two-sided Wilcoxon signed-rank test (FDR corrected for multiple comparisons), and of Bayesian nonparametric pairwise comparisons ($BF_{10}$) with the posterior distribution given as median [95% credibility interval of the median as provided by JASP]

lack of progression through the timing map hierarchy[19]. The major and minor extent of the tuned response function also differed significantly between visual field maps (major: $F_{(17, 520)} =$ 2.67, $p < 0.001$, $\eta_p^2 = 0.08$; minor: $F_{(17, 520)} = 6.00$, $p < 0.001$, $\eta_p^2 = 0.17$) (Fig. 4g). Post hoc multiple comparisons showed a decrease in the major extent and an increase in the minor extent of the response function from early to later visual field maps, with the major extent highest and the minor response extent lowest where monotonic models predict responses better than tuned models. This may be because these monotonic responses have no response peak, so responses were best predicted by a function that responds similarly to many event timings to capture the large monotonic changes in amplitude with event frequency. Because of these simultaneous but opposite changes in the response function's major and minor extents, their ratio also differed between visual field maps ($F_{(17, 520)} = 11.59$, $p < 0.001$, $\eta_p^2 = 0.28$) (Fig. 4h) and progressed similarly to the response function's major extent, again consistent with changes through the timing map hierarchy.

The response function's orientation also differed significantly between visual field maps ($F_{(17, 520)} = 3.72$, $p < 0.001$, $\eta_p^2 = 0.11$) (Fig. 4i). Post hoc multiple comparisons showed a progressive decrease along the visual hierarchy, such that the response function was narrower in the duration direction in the early visual field maps (particularly where monotonic models fit better than tuned models) and narrower in the period direction in later visual field maps. Where the monotonic model fit better, again this orientation appears to describe the tuned response function that best captures monotonic changes in response amplitude. However, where tuned responses fit better, there is hierarchical transformation of timing-tuned responses to give a finer representation of event period than event duration (as in the timing map hierarchy[19]). This may provide better information for frontal premotor areas to plan precisely timed actions to interact with visual events.

We can visualize how these differences in response model parameters predict different response amplitudes to different event timings by taking the median response model parameters in each visual field map and rendering response functions with these parameters (Fig. 4j, k). We restrict this to possible event timings, where event duration is less than event period. These response functions change considerably between visual field maps. Nevertheless, in all visual field maps, these two response models predict very different response functions regardless of the response function parameters within that map.

**Relationship between model fits and retinotopic location.** Since the monotonic response model performed well in early visual field maps, it is likely that monotonic responses are computed from low-level stimulus properties. This predicts they would be limited to the retinotopic location of the stimulus (the central visual field representation), rather than elsewhere (the peripheral visual field representation). Furthermore, as timing tuned responses are topographically mapped by their timing preferences[19,31], we predicted that they would not be restricted to stimulus's retinotopic location but instead be encoded in an abstracted, quantity-based frame of reference. We, therefore, investigated whether the fits of the monotonic and tuned response models differed between central and peripheral representations within each visual field map. We excluded voxels with a cross-validated variance explained ($R^2$) of 0 in both models in the data on which the models were fit.

First, we visualized the progressions of variance explained of each response model throughout the visual field by plotting the variance explained against eccentricity (see Fig. 5 for a representative selection of visual field maps and Supplementary Fig. 8). Each model's fit decreased as the voxel's preferred visual position moved away from the stimulus area. This decrease was steep and sudden in early visual and lateral occipital visual field maps, which have smaller spatial receptive fields and where responses to event timing are best captured by monotonic response models. A decrease in model fits with eccentricity was also apparent in higher visual field maps, where the tuned response model begins to fit better, but this decrease was more gradual.

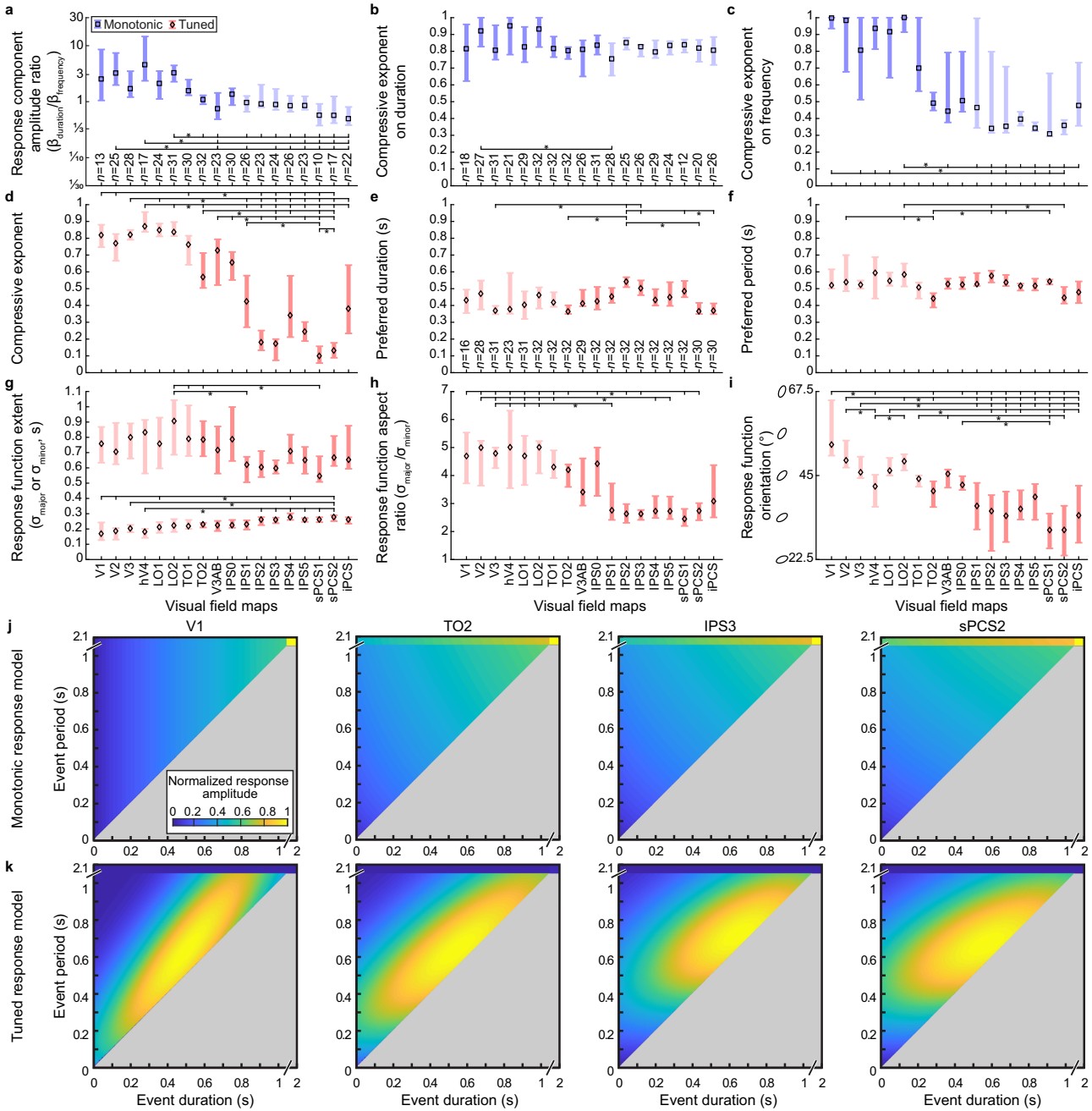

**Fig. 4 Comparison of monotonic and tuned response model parameters between visual field maps. a** In the monotonic response model, the ratio of response amplitudes of the duration component divided by the frequency component decreased along the visual field map hierarchy. **b** The duration component's compressive exponent did not systematically change along the visual hierarchy. **c** The frequency component's compressive exponent decreased along the visual hierarchy. **d** In the timing tuned response model, the compressive exponent on event frequency similarly decreased along the visual hierarchy. **e, f** The median preferred event duration and period did not systematically change along the visual hierarchy. **g, h** There was a decrease in the major extent and an increase in the minor extent of the response function (and consequently a decrease in their aspect ratio) along the visual hierarchy, particularly changing where tuned response models begin to dominate later visual field maps' responses. **i** The response function's orientation progressively rotated along the visual hierarchy, so the response function was narrower in the duration direction in the early visual field maps (particularly where monotonic models fit better than tuned models) and narrower in the period direction in the later visual field maps. Points show the median parameter value across recordings from different hemispheres, error bars show 95% confidence intervals computed from 1000 bootstrap iterations. Brackets and stars show significant differences in multiple comparisons between all maps: all brackets to the left of the star are significantly different from all brackets to the right of the star (two-sided Dunn's test, corrected for multiple comparisons, $p < 0.05$). $n$ indicates the number of hemispheres included for the monotonic (**b, c**) or tuned response model (**d–i**) in each visual field map. $n$ deviates for the monotonic component's ratio (**a**), as maps with all voxels' ratios of 0 or infinite were excluded. Light colors show visual field maps where the shown response model captured responses significantly less well than the other response model. **j, k** Visualizations of response functions given by the median response model's parameters in representative visual field maps. Source data are provided as a Source Data file.

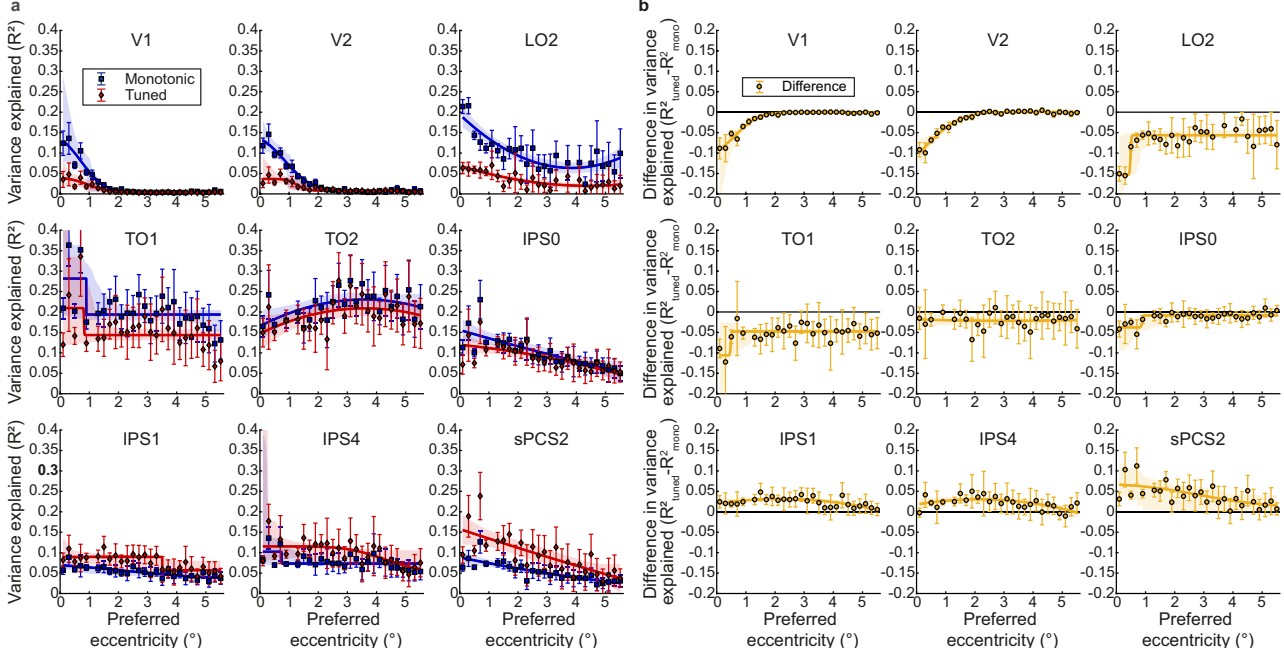

**Fig. 5 Progression of timing response model fits with preferred visual field eccentricity in a representative selection of visual field maps. a** Early and lateral occipital visual field maps show a sharp decrease of model fits moving away from the retinotopic representation of the stimulus position. This decrease then becomes more gradual where tuned response model fits begin to improve. **b** The difference between the response model fits (tuned - monotonic) also decreases with eccentricity in the early and lateral occipital visual field maps, but shows no consistent relationship with eccentricity after TO1. Markers show mean variance explained per eccentricity bin, error bars show the standard error of the mean. For all bins, $n \geq 50$ included voxels. Solid lines show the best fit to changes with eccentricity, shaded areas around these lines are their 95% confidence intervals. Note that the data for these plots are not thresholded to a variance explained above 0.2, but above 0 for the best fitting model in the data on which the models were fit. See also Supplementary Fig. 8. Source data are provided as a Source Data file.

The gradual progression from monotonic to tuned response model fits along the visual field map hierarchy suggests tuned responses could be derived from monotonic inputs. Furthermore, it is important to remember that both response models describe the responses per event, so even in a monotonic response model (where total response amplitude increases with event frequency) the response per event actually decreases with frequency. As a result of this strong relationship between frequency and response amplitude in both models, much of the fMRI response of a timing-tuned neural population can be captured by a monotonic response model. Furthermore, as timing-tuned responses are found overlapping with visual field maps it may be that a spatial representation responding monotonically to temporal contrast is intermixed with a temporal representation with timing-tuned responses. We therefore also assess how much extra response variance the tuned response model can capture beyond the prediction of a monotonic response model: the difference between the tuned and monotonic response model fits.

This difference between the monotonic and tuned response models' fits decreased with eccentricity in early visual and lateral occipital visual field maps where responses to event timing are best captured by the monotonic response model. After TO1 the difference in goodness-of-fit does not show a consistent relationship to the voxel's preferred visual position eccentricity.

Second, to assess the significance of the effect of eccentricity on timing response model fits within each map, we compared the variance explained in voxels near (<1°) and far (>2°) from the center of the visual field using three-way ANOVAs for monotonic response model fits, tuned response model fits, and their difference to assess how model fits differed between visual field maps, eccentricity range, and participants.

For the monotonic response model there was a main effect of eccentricity range ($F_{(1, 1138)} = 159.77$, $p < 0.001$, $\eta_p^2 = 0.13$) and map ($F_{(17, 1122)} = 61.27$, $p < 0.001$, $\eta_p^2 = 0.49$), and an interaction between eccentricity range and visual field map ($F_{(17, 1122)} = 7.27$, $p < 0.001$, $\eta_p^2 = 0.10$). This interaction demonstrated that the difference between eccentricity ranges differed between maps. Not all maps showed a difference between eccentricity ranges, but post hoc multiple comparisons demonstrated a higher variance explained near the central visual field representation than in the periphery in several maps (V1, V2, V3, hV4, LO1, LO2, V3AB, IPS0, IPS2, IPS3, sPCS2, and iPCS; Fig. 6a; Table 2).

For the tuned response model, there were main effects of eccentricity range ($F_{(1, 1138)} = 35.64$, $p < 0.001$, $\eta_p^2 = 0.03$) and map ($F_{(17, 1122)} = 34.88$, $p < 0.001$, $\eta_p^2 = 0.35$) but the interaction between eccentricity range and map did not reach significance ($F_{(17, 1122)} = 1.63$, $p = 0.0502$, $\eta_p^2 = 0.02$). Again, post hoc multiple comparisons demonstrated a higher variance explained near the central visual field representation than in the periphery in several maps (V1, V2, V3, hV4, LO1, LO2, V3AB, IPS3, IPS5, and sPCS2, iPCS; Fig. 6b; Table 2).

Finally, the difference in variance explained between the models revealed a main effect of eccentricity range ($F_{(1, 1138)} = 32.20$, $p < 0.001$, $\eta_p^2 = 0.03$) and map ($F_{(17, 1122)} = 46.59$, $p < 0.001$, $\eta_p^2 = 0.42$), and an interaction between the eccentricity range and map ($F_{(17, 1122)} = 3.68$, $p < 0.001$, $\eta_p^2 = 0.05$). Here, post hoc multiple comparisons demonstrated that the difference in variance explained between the models was larger near the center of the visual field than in the periphery only in V1, V2, V3, hV4, LO1, LO2, V3AB, and IPS0 (Fig. 6c; Table 2). Notably, these are areas in which the monotonic response model outperformed the tuned response model or they were not significantly different.

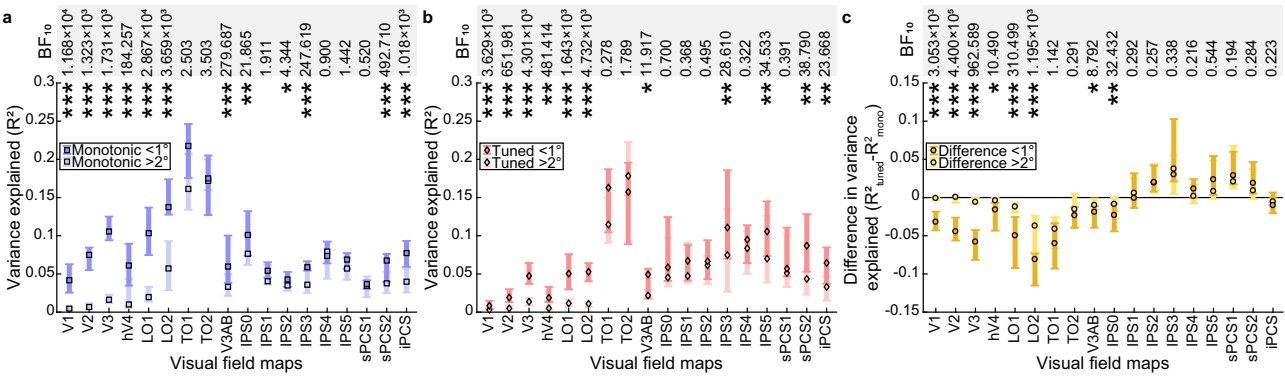

**Fig. 6 Model fits decrease with distance from the stimulus area, but the additional response variance explained by the timing-tuned model is not affected in visual field maps where the tuned response model fits better. a** Cross-validated variance explained by the monotonic response model in eccentricity ranges near and far from the stimulus area. **b** Cross-validated variance explained by the tuned response model in eccentricity ranges near and far from the stimulus area. In both cases, responses are clearer near the stimulus area in many visual field maps. **c** The difference in cross-validated variance explained between the two models was significant only in V1, V2, V3, hV4, LO1, LO2, V3AB, and IPS0, where the monotonic response model fit better than the tuned response model. Markers show the median variance explained in each visual field map, error bars show the 95% confidence interval, computed from 1000 bootstrap iterations. Only voxels that had a variance explained above 0 for the best fitting model in the data on which the models were fit are included. Significant differences between the near and far eccentricity ranges (two-sided Wilcoxon signed-rank test, FDR corrected for multiple comparisons): *$p < 0.05$ **$p < 0.01$, and ***$p < 0.001$. For hV4 $n = 28$ hemispheres included in the eccentricity range comparison. For LO1 and iPCS $n = 31$. For all other visual field maps $n = 32$. Bayes factors ($BF_{10}$) for each paired comparison are presented in the top row. Source data are provided as a Source Data file.

Areas in which the tuned response model outperformed the monotonic response model showed no significant difference between eccentricity ranges. The Bayes factors of the paired comparisons in the latter areas are below one, indicating that such a difference between eccentricity ranges is indeed absent. This suggests that the additional response component explained by tuning was location-independent.

**Transition from monotonic to tuned responses to timing along the timing map hierarchy.** In previous work, the locations of timing-tuned responses were defined as timing maps, as the tuned response model performed best over the brain as a whole[19]. Such timing maps overlap with visual field maps, but do not have the same borders so do not include the same set of voxels. In the previous analyses, we have used visual field map borders, which may include both a spatial representation responding monotonically to temporal contrast and a temporal representation with timing-tuned responses. The gradual transition from monotonic to increasingly timing-tuned responses along the visual field map hierarchy suggests a similar gradual transition along the timing map hierarchy. Therefore, to focus on the temporal representation more specifically, we compared the models specifically within the timing map borders. We excluded voxels where the cross-validated variance explained ($R^2$) was 0.2 or less for both models in the data on which the models were fit. We used a three-factor ANOVA to assess how model fits differed between timing maps, models, and participants. Model fits differed between timing maps ($F_{(9, 608)} = 25.33$, $p < 0.001$, $\eta_p^2 = 0.28$) and models ($F_{(1, 616)} = 66.40$, $p < 0.001$, $\eta_p^2 = 0.10$), and there was an interaction between maps and model ($F_{(9, 608)} = 16.94$, $p < 0.001$, $\eta_p^2 = 0.21$). Post hoc multiple comparisons demonstrated that responses of anterior maps (TLS, TPCI, TPCM, TPCS, TFI, and TFS) were better fit by the tuned response model (Fig. 7). However, posterior timing maps (TLO and TTOP) were better fit by the monotonic response model, which was not evident in previous analyses that did not compare models in each map separately. Responses in timing maps TTOA and TPO (which largely overlap with TO2 and IPS0, respectively) showed no

significant difference between the fits of the monotonic and tuned response models (Table 3).

We do not compare model fits between eccentricity ranges within timing maps because the voxel selection within the timing maps is not chosen to cover all visual field position eccentricities.

## Discussion

Tuned responses to sub-second visual event timing occur in a hierarchical network of topographically organized maps throughout the human association cortices[19]. Responses of early visual field maps are also modulated by event timing, monotonically increasing with event duration and frequency[17,18]. In the current study, we asked how these two sets of responses are related throughout the brain's hierarchy of both timing maps and visual field maps. We also assessed how the responses to timing and visual field position are related. Visual field maps and timing maps largely overlap but have different borders and so include different neural populations which may have slightly different properties. We found increasingly clear monotonic responses to visual event duration and frequency from primary visual cortex to lateral occipital and temporal-occipital visual field maps (LO1, LO2, and TO1). After this, we found a gradual transition from monotonic responses to tuned responses from posterior to anterior brain areas, both when separated into visual field maps and timing maps. We also found gradual transformations of response model parameters through the visual field map and timing map hierarchies. Both kinds of responses typically decreased when moving away from the retinotopic location of the stimulus. However, the difference between the model fits in areas where the tuned model outperformed the monotonic response model (i.e. the additional variance explained by the tuned response model) did not consistently depend on retinotopic location of the neural population.

We found that monotonic responses to visual stimulus event timing occurred in early visual field maps (as is also reported by Stigliani and colleagues[17]). More specifically, the amplitude of these monotonic responses accumulated sub-linearly with event duration and frequency (in line with Zhou and colleagues[18]).

**Table 2 Descriptive and test statistics of the paired eccentricity range comparisons within visual field maps for the monotonic response model, tuned response model, and the difference between their fits.**

| Visual field map | Near[a] | Far[b] | Z statistic | Effect size (r) | p value | BF$_{10}$ | Posterior distribution |
|---|---|---|---|---|---|---|---|
| Monotonic fit ($R^2$) | | | | | | | |
| V1 | 0.04 [0.03, 0.06] | 0.00 [0.00, 0.01] | 4.94 | 0.87 | $4 \times 10^{-6}$ | $1.168 \times 10^4$ | 1.874 [1.007, 2.678] |
| V2 | 0.07 [0.06, 0.08] | 0.01 [0.00, 0.01] | 4.94 | 0.87 | $4 \times 10^{-6}$ | $1.323 \times 10^3$ | 1.750 [0.891, 2.553] |
| V3 | 0.11 [0.09, 0.13] | 0.02 [0.01, 0.02] | 4.94 | 0.87 | $4 \times 10^{-6}$ | $1.731 \times 10^3$ | 2.227 [0.772, 2.920] |
| hV4 | 0.06 [0.03, 0.09] | 0.01 [0.01, 0.04] | 3.60 | 0.68 | $6 \times 10^{-4}$ | 184.257 | 0.889 [0.450, 1.339] |
| LO1 | 0.10 [0.07, 0.14] | 0.02 [0.01, 0.03] | 4.84 | 0.87 | $5 \times 10^{-6}$ | $2.867 \times 10^4$ | 2.129 [2.054, 3.171] |
| LO2 | 0.14 [0.13, 0.17] | 0.06 [0.03, 0.09] | 4.94 | 0.87 | $4 \times 10^{-6}$ | $3.659 \times 10^3$ | 1.890 [1.052, 2.690] |
| TO1 | 0.22 [0.18, 0.25] | 0.16 [0.13, 0.21] | 2.00 | 0.35 | 0.063 | 2.503 | 0.408 [0.053, 0.769] |
| TO2 | 0.18 [0.13, 0.21] | 0.17 [0.16, 0.20] | −1.96 | −0.35 | 0.064 | 3.503 | −0.420 [0.773, −0.072] |
| V3AB | 0.06 [0.04, 0.10] | 0.03 [0.02, 0.05] | 3.52 | 0.62 | $8 \times 10^{-4}$ | 279.687 | 0.858 [0.416, 1.268] |
| IPS0 | 0.10 [0.07, 0.13] | 0.08 [0.06, 0.10] | 2.92 | 0.52 | 0.006 | 21.865 | 0.581 [0.213, 0.957] |
| IPS1 | 0.05 [0.05, 0.07] | 0.04 [0.04, 0.06] | 1.81 | 0.32 | 0.082 | 1.911 | 0.364 [0.023, 0.713] |
| IPS2 | 0.04 [0.03, 0.05] | 0.03 [0.03, 0.04] | 2.23 | 0.39 | 0.040 | 4.344 | 0.456 [0.095, 0.834] |
| IPS3 | 0.06 [0.05, 0.07] | 0.04 [0.03, 0.06] | 3.78 | 0.67 | $4 \times 10^{-4}$ | 247.619 | 0.845 [0.446, 1.234] |
| IPS4 | 0.07 [0.06, 0.09] | 0.08 [0.04, 0.09] | 1.80 | 0.32 | 0.082 | 0.900 | 0.299 [−0.037, 0.669] |
| IPS5 | 0.07 [0.06, 0.08] | 0.06 [0.04, 0.07] | 1.76 | 0.31 | 0.083 | 1.442 | 0.347 [0.001, 0.705] |
| sPCS1 | 0.04 [0.03, 0.04] | 0.03 [0.02, 0.05] | 1.38 | 0.24 | 0.166 | 0.520 | 0.242 [−0.095, 0.586] |
| sPCS2 | 0.07 [0.04, 0.08] | 0.04 [0.02, 0.05] | 3.87 | 0.68 | $3 \times 10^{-4}$ | 492.71 | 0.926 [0.509, 1.363] |
| iPCS | 0.08 [0.06, 0.09] | 0.04 [0.03, 0.06] | 3.90 | 0.70 | $3 \times 10^{-4}$ | $1.018 \times 10^3$ | 0.965 [0.513, 1.445] |
| Tuned fit ($R^2$) | | | | | | | |
| V1 | 0.01 [0.00, 0.01] | 0.00 [0.00, 0.00] | 3.95 | 0.70 | $3 \times 10^{-4}$ | $3.629 \times 10^3$ | 0.987 [0.569, 1.479] |
| V2 | 0.02 [0.01, 0.03] | 0.00 [0.00, 0.01] | 4.19 | 0.74 | $1 \times 10^{-4}$ | 651.981 | 1.110 [0.617, 1.656] |
| V3 | 0.05 [0.04, 0.06] | 0.01 [0.01, 0.02] | 4.71 | 0.83 | $1 \times 10^{-5}$ | $4.301 \times 10^3$ | 1.468 [0.881, 2.078] |
| hV4 | 0.02 [0.01, 0.03] | 0.01 [0.00, 0.01] | 3.51 | 0.66 | 0.001 | 481.414 | 0.901 [0.461, 1.396] |
| LO1 | 0.05 [0.03, 0.08] | 0.01 [0.01, 0.02] | 4.82 | 0.87 | $1 \times 10^{-5}$ | $1.643 \times 10^3$ | 1.762 [1.019, 2.644] |
| LO2 | 0.05 [0.04, 0.06] | 0.01 [0.01, 0.01] | 4.94 | 0.87 | $1 \times 10^{-5}$ | $4.732 \times 10^3$ | 1.816 [0.853, 2.712] |
| TO1 | 0.16 [0.10, 0.19] | 0.11 [0.09, 0.15] | 0.79 | 0.14 | 0.432 | 0.278 | 0.143 [−0.187, 0.485] |
| TO2 | 0.16 [0.09, 0.20] | 0.18 [0.15, 0.22] | −1.96 | −0.35 | 0.074 | 1.789 | −0.378 [−0.731, −0.033] |
| V3AB | 0.05 [0.02, 0.06] | 0.02 [0.02, 0.03] | 2.62 | 0.46 | 0.014 | 11.917 | 0.524 [0.164, 0.896] |
| IPS0 | 0.06 [0.04, 0.12] | 0.05 [0.03, 0.10] | 1.61 | 0.28 | 0.149 | 0.700 | 0.278 [−0.057, 0.617] |
| IPS1 | 0.07 [0.04, 0.09] | 0.05 [0.04, 0.09] | 1.16 | 0.20 | 0.277 | 0.368 | 0.194 [−0.138, 0.543] |
| IPS2 | 0.07 [0.04, 0.09] | 0.06 [0.04, 0.07] | 1.10 | 0.20 | 0.286 | 0.495 | 0.239 [−0.089, 0.598] |
| IPS3 | 0.11 [0.07, 0.19] | 0.07 [0.03, 0.13] | 2.92 | 0.52 | 0.008 | 28.610 | 0.599 [0.212, 0.986] |
| IPS4 | 0.09 [0.06, 0.11] | 0.08 [0.05, 0.11] | 1.48 | 0.26 | 0.180 | 0.322 | 0.175 [−0.160, 0.525] |
| IPS5 | 0.11 [0.07, 0.14] | 0.07 [0.04, 0.13] | 2.79 | 0.49 | 0.010 | 34.533 | 0.600 [0.227, 0.981] |
| sPCS1 | 0.06 [0.05, 0.11] | 0.05 [0.03, 0.10] | 1.22 | 0.21 | 0.269 | 0.391 | 0.194 [−0.139, 0.533] |
| sPCS2 | 0.09 [0.05, 0.13] | 0.04 [0.02, 0.09] | 2.88 | 0.51 | 0.008 | 38.790 | 0.611 [0.241, 0.995] |
| iPCS | 0.06 [0.04, 0.08] | 0.03 [0.02, 0.05] | 2.98 | 0.53 | 0.007 | 23.668 | 0.603 [0.220, 0.990] |
| Difference ($R^2_{tuned} - R^2_{mono}$) | | | | | | | |
| V1 | −0.03 [−0.04, −0.02] | −0.00 [−0.00, 0.00] | −4.92 | −0.87 | $7 \times 10^{-6}$ | $3.053 \times 10^3$ | −2.697 [−3.538, −1.107] |
| V2 | −0.04 [−0.06, −0.03] | 0.00 [−0.01, 0.00] | −4.86 | −0.86 | $7 \times 10^{-6}$ | $4.400 \times 10^5$ | −1.841 [−2.394, −1.118] |
| V3 | −0.06 [−0.08, −0.04] | −0.01 [−0.01, −0.00] | −4.94 | −0.87 | $7 \times 10^{-6}$ | 962.589 | −2.328 [−3.231, −0.940] |
| hV4 | −0.02 [−0.04, −0.01] | −0.00 [−0.01, −0.00] | −2.53 | −0.48 | 0.026 | 10.490 | −0.540 [−0.943, −0.158] |
| LO1 | −0.05 [−0.09, −0.03] | −0.01 [−0.02, −0.01] | −3.68 | −0.66 | $8 \times 10^{-4}$ | 310.499 | −0.883 [−1.302, −0.464] |
| LO2 | −0.08 [−0.12, −0.07] | −0.04 [−0.07, −0.02] | −4.56 | −0.81 | $2 \times 10^{-5}$ | $1.195 \times 10^3$ | −1.264 [−1.782, −0.703] |
| TO1 | −0.06 [−0.09, −0.03] | −0.04 [−0.07, −0.03] | −1.94 | −0.34 | 0.104 | 1.142 | −0.332 [−0.688, 0.018] |
| TO2 | −0.02 [−0.04, −0.01] | −0.02 [−0.04, 0.01] | −0.77 | −0.14 | 0.559 | 0.291 | −0.153 [−0.484, 0.183] |
| V3AB | −0.02 [−0.04, −0.01] | −0.01 [−0.02, −0.00] | −2.80 | −0.50 | 0.013 | 8.792 | −0.491 [−0.854, −0.129] |
| IPS0 | −0.02 [−0.04, −0.01] | −0.01 [−0.02, 0.00] | −3.12 | −0.55 | 0.005 | 32.432 | −0.681 [−1.008, −0.237] |
| IPS1 | −0.00 [−0.01, 0.03] | 0.01 [−0.00, 0.03] | −0.95 | −0.17 | 0.471 | 0.292 | −0.157 [−0507, 0.176] |
| IPS2 | 0.02 [0.01, 0.04] | 0.02 [0.01, 0.04] | −1.12 | −0.20 | 0.393 | 0.257 | −0.131 [−0.470, 0.205] |
| IPS3 | 0.03 [0.02, 0.10] | 0.04 [0.00, 0.06] | 1.25 | 0.22 | 0.344 | 0.338 | 0.190 [−0.139, 0.531] |
| IPS4 | 0.01 [−0.00, 0.02] | 0.00 [−0.01, 0.02] | 0.73 | 0.13 | 0.559 | 0.216 | 0.089 [−0.245, 0.426] |
| IPS5 | 0.02 [0.00, 0.05] | 0.01 [−0.00, 0.05] | 1.27 | 0.22 | 0.344 | 0.544 | 0.247 [−0.091, 0.603] |
| sPCS1 | 0.03 [0.02, 0.06] | 0.02 [0.01, 0.07] | −0.15 | −0.03 | 0.881 | 0.194 | 0.021 [−0.322, 0.358] |
| sPCS2 | 0.02 [0.01, 0.05] | 0.01 [−0.00, 0.04] | 0.60 | 0.11 | 0.618 | 0.284 | 0.150 [−0.180, 0.481] |
| iPCS | −0.01 [−0.02, 0.01] | −0.01 [−0.01, 0.00] | −0.29 | −0.05 | 0.814 | 0.223 | −0.092 [−0.426, 0.241] |

Data are median cross-validated variance explained [95% confidence interval of the median computed from 1000 bootstrap iterations]. These are the outcomes of a two-sided Wilcoxon signed-rank test (FDR corrected for multiple comparisons), and of Bayesian nonparametric pairwise comparisons (BF10) with the posterior distribution given as median [95% credibility interval of the median as provided by JASP].
[a]The near eccentricity range contains voxels with an eccentricity <1°.
[b]The far eccentricity range contains voxels >2° and <5.6°.

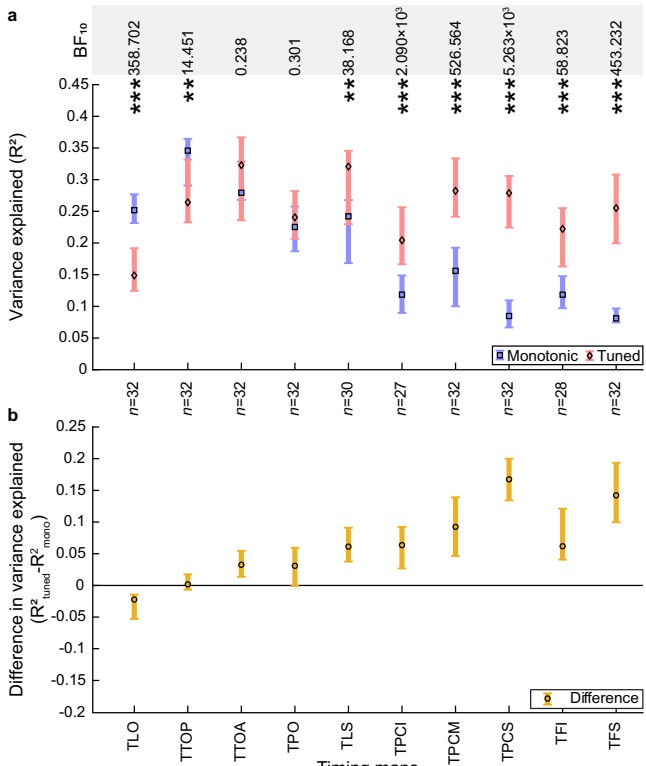

**Fig. 7 Comparison of fits of monotonic and tuned response models in each timing map. a** Cross-validated variance explained by the monotonic and tuned response models. **b** Difference in cross-validated variance explained between models (tuned – monotonic) progressively changes from monotonic to tuned up the timing map hierarchy. Markers show the median variance explained of each timing map, error bars show the 95% confidence interval, computed from 1000 bootstrap iterations. Only the voxels that had a variance explained above 0.2 for the best fitting model in the data on which the models were fit are included. Significant differences between the models (two-sided Wilcoxon signed-rank test, FDR corrected for multiple comparisons): *$p < 0.05$ **$p < 0.01$, and ***$p < 0.001$. $n$ indicates the number of hemispheres included in the comparison in each timing map (for both panels). Bayes factors (BF$_{10}$) for each paired comparison are presented in the top row. Source data are provided as a Source Data file.

In contrast to these previous experiments, we characterized the time between events in terms of the frequency of event onsets, rather than interstimulus interval (the time from one single event's offset to another single event's onset). This describes our stimulus design more straightforwardly because we use a repeating periodic stimulus that includes a large variation in event durations (the time from event onset to offset) for any particular frequency. However, unlike in Zhou and colleagues, we found no decrease in event duration's compressive exponent along the visual field map hierarchy. This appears to be because Zhou and colleagues separate an event's period into two components, event duration, and interstimulus interval. By doing this, the event period's compressive exponent may be reflected in both their components, while in our approach it is all captured by the event period component and so does not affect the event duration component.

The fits of the monotonic response model increased from the V1 to the TO visual field maps. This is in line with the relative increase of transient responses to temporal contrast (stronger for magnocellular inputs), compared to sustained responses to temporal contrast (stronger for parvocellular inputs), when moving up the visual field map hierarchy[17]. Indeed, in macaques, the inputs to neurons in area MT are mainly from the magnocellular stream (the TO visual field maps are functionally homologous to MT[45–47], which has benefits for an area specialized in motion processing. A greater proportion of magnocellular inputs increases the modulation of responses by stimulus timing and may likewise be beneficial for subsequently deriving representations of stimulus timing.

A transition from monotonic to tuned responses occurs in the TO2 and IPS0 visual field maps or, similarly, the TTOA and TPO timing maps. Strikingly, visual field map TO2 and timing map TTOA overlap with the human MT+[48,49]. This area appears to be required for visual event timing perception as transcranial magnetic stimulation here disrupts timing judgements[10]. A network of previously described areas then shows timing-tuned responses[19]. A similar network allows event duration decoding from multivoxel activity patterns[29]. Early visual and lateral occipital areas also allow such decoding, apparently from the monotonic responses we show in these areas. While we find widespread monotonic responses, in that study these were limited to the occipital pole, i.e. the early visual cortex. This may be because our monotonic responses can reflect an increase in response amplitude with event frequency (not only duration) and allow a nonlinear increase in response amplitude with duration. Previous fMRI experiments have also shown repetition suppression by repeated presentation of the same event duration (consistent with duration tuning) in the supramarginal gyrus of the inferior parietal lobule[28], near our TPCI and TLS timing maps. Here we extend these findings by demonstrating that timing-tuned responses may be derived from the response dynamics of visual field maps, demonstrating contributions of both tuned and monotonic response components in many areas, and demonstrating relationships between these responses and visual spatial responses.

Either monotonic or tuned responses should allow decoding of event timing, though only tuned responses should produce repetition suppression. Some spatial separation of neural populations with different timing preferences (for example in topographic maps[19,31]) is required to allow decoding of event timing[29], or measure different timing preferences at the spatial scale of fMRI[19,31]. Importantly, this does not require that an fMRI voxel contains only neurons with similar timing preferences, simply that neurons with different timing preferences are not homogeneously intermixed.

We had previously reported that the responses of the timing maps were better fit by tuned than monotonic functions[19]. In the current study, we find that the responses in the timing maps TLO and TTOP are better captured by monotonic rather than tuned response models. We had previously grouped all timing maps together in this comparison, and used a monotonic response model that assumed a linear accumulation of response amplitude with event duration[19]. Based on the results from the current study, we are no longer convinced that TLO and TTOP are topographically organized by preferred event timing because monotonic response functions have no timing preference.

Nevertheless, there may be timing-tuned neurons with specific timing preferences in TLO and TTOP. The transition from monotonic to tuned responses is remarkably gradual across the visual field map and timing map hierarchies. This suggests that tuned and monotonic timing-selective responses may not be mutually exclusive. Intermixed monotonic and motor timing-tuned neuronal populations have been found in macaque medial premotor cortex during a rhythm continuation task[40,41]. FMRI's spatial resolution groups the responses of large neural populations, which prevents us from distinguishing a hierarchical change in the proportions of intermixed monotonic and tuned

**Table 3 Descriptive and test statistics of the paired model comparisons within timing maps.**

| Timing map | Monotonic fit ($R^2$) | Tuned fit ($R^2$) | Z statistic | Effect size (r) | p value | $BF_{10}$ | Posterior distribution |
|---|---|---|---|---|---|---|---|
| TLO | 0.25 [0.23, 0.28] | 0.15 [0.12, 0.19] | −4.06 | −0.72 | $1 \times 10^{-4}$ | 538.702 | 1.080 [0.523, 1.536] |
| TTOP | 0.35 [0.29, 0.36] | 0.26 [0.23, 0.33] | −2.80 | −0.50 | 0.006 | 14.451 | 0.582 [0.194, 0.981] |
| TTOA | 0.28 [0.27, 0.33] | 0.32 [0.24, 0.37] | 0.50 | 0.09 | 0.614 | 0.238 | −0.115 [−0.456, 0.251] |
| TPO | 0.23 [0.19, 0.26] | 0.24 [0.21, 0.28] | 0.62 | 0.11 | 0.597 | 0.301 | −0.170 [−0.509, 0.168] |
| TLS | 0.24 [0.17, 0.27] | 0.32 [0.23, 0.35] | 3.34 | 0.61 | 0.001 | 38.168 | −0.684 [−1.101, −0.272] |
| TPCI | 0.12 [0.09, 0.15] | 0.20 [0.17, 0.26] | 4.01 | 0.77 | $1 \times 10^{-4}$ | $2.090 \times 10^3$ | −1.122 [−1.654, −0.605] |
| TPCM | 0.16 [0.10, 0.19] | 0.28 [0.24, 0.33] | 4.86 | 0.86 | $4 \times 10^{-6}$ | 526.564 | −1.826 [−2.648, −0.871] |
| TPCS | 0.08 [0.07, 0.11] | 0.28 [0.22, 0.31] | 4.94 | 0.87 | $4 \times 10^{-6}$ | $5.263 \times 10^3$ | −2.153 [−2.737, −1.365] |
| TFI | 0.12 [0.10, 0.15] | 0.22 [0.16, 0.26] | 3.44 | 0.65 | $1 \times 10^{-3}$ | 58.823 | −0.862 [−1.375, −0.384] |
| TFS | 0.08 [0.07, 0.10] | 0.26 [0.20, 0.31] | 4.94 | 0.87 | $4 \times 10^{-6}$ | 453.232 | −2.257 [−2.940, −0.856] |

Data are median cross-validated variance explained [95% confidence interval of the median computed from 1000 bootstrap iterations]. These are the outcomes of a two-sided Wilcoxon signed-rank test (FDR corrected for multiple comparisons) and of Bayesian nonparametric pairwise comparisons ($BF_{10}$) with the posterior distribution is given as median [95% credibility interval of the median as provided by JASP].

neurons from a hierarchical transformation of event timing representations of single neurons. Both of these possibilities have independent components reflecting tuning for event timing and the accumulation of neural response amplitude with event frequency. Regardless of whether these components are found in the responses of distinct neurons or contribute to the responses of the same neuron, we propose that the difference between the fits of the monotonic and tuned response models reflects neural tuning for event timing after accounting for the underlying frequency-dependence of sensory responses.

Furthermore, our model validation procedure (Supplementary Fig. 2) demonstrated that ground-truth monotonic responses are very rarely classified as tuned. This gives high confidence that visual field maps identified as showing tuned responses do indeed show tuned responses, and indeed their response function parameters in tuned response models should facilitate identification as tuned responses. However, ground-truth tuned responses can be misclassified as monotonic when their response function approximates a monotonic function: where the preferred timing is outside the presented timing range, the response function is very large, or the compressive exponent on event frequency is high. Therefore, for early visual field maps identified as showing monotonic responses, some borderline cases may actually have tuned responses that our procedure cannot reliably identify. Specifically, tuned response models fit here have high estimates of the compressive exponent and slightly larger response functions than later visual field maps showing tuned responses. This does not strongly affect our conclusions because such tuned response functions closely resemble monotonic response functions, there is a gradual transition from monotonic to tuned responses, and tuned responses may have monotonic components. Nevertheless, it is possible that tuned responses emerge slightly earlier in the visual hierarchy than our model comparison suggests.

The goodness of fit of both the monotonic and tuned response models are greatest near the retinotopic location of the visual field center, where the stimulus was presented, in most visual field maps. However, in visual field maps where the tuned response model outperforms the monotonic response model, the extra response variance captured by the tuned response model is independent of retinotopic organization. Based on the location-invariance of this additional tuned response component (beyond the monotonic frequency-dependent response component), the location-dependence of the tuned response model's fit may be rooted in the location-dependence of the monotonic, frequency-dependent response component.

The decrease in model fits with eccentricity is clearest in early visual field maps with small spatial pRFs. It is more gradual in

higher visual field maps with larger pRFs, which is expected as peripheral pRFs in these visual field maps also cover the central visual field where the stimulus was presented. Only the timing-tuned response component is independent of the retinotopic location of the neural population: monotonic components of the same neural population's response decrease with retinotopic distance from the stimulus location. In perception, visual duration aftereffects, which are thought to reflect duration-tuned neural responses[14], may have a limited though large spatial spread[50], though others studies show an unlimited spread within and between visual hemifields[15,51]. If the visual duration after-effect depends on effects on timing-tuned neural populations and is spatially limited, this appears inconsistent with retinotopically-independent timing-tuned response components. This spatial limit may result from interactions between timing-tuned neurons and retinotopically-specific monotonically responding neurons in the same neural population, or effects of adaptation on earlier monotonic responses from which timing-tuned responses are derived. Alternatively, timing-tuned response components might depend on retinotopic location if we mapped a larger area of the visual field: in studies quantifying the duration aftereffect's spatial spread, at 5° from the adapter it is around 50% of the strength at the adapted location[50], and our visual field mapping stimulus covered only 5.6° from the stimulus area. So, timing-tuned neurons may have large receptive fields that the stimulus must fall within[50]. However, analogous numerosity aftereffects in visual perception[52] appear to reflect changes in the responses of numerosity-tuned neurons[26,53] and have a similarly limited spatial spread[54], but numerosity-tuned neurons do not require spatial receptive fields overlapping the stimulus area[32,33,39,55]. Therefore, relationships between the spatial spread of perceptual quantity adaptation effects and the receptive fields of quantity-tuned neural populations may be more complex than they appear: rather than perceptual quantity aftereffects providing clear evidence demonstrating that quantity-tuned responses follow spatial receptive field properties, neural tuning for different quantities[33,34] and visual position[32,33,39,55] seem to be independent dimensions of neural responses.

Notably, we do not analyze responses in ventral stream visual field maps (VO1, VO2, and PHC) in detail. These are generally implicated in object processing, so we focused our data collection on the lateral, dorsal stream, and parieto-frontal visual field maps implicated in motion processing, multisensory integration, and attention control. We also investigate early visual field maps that provide their inputs. In the ventral visual field maps, the maximum variance explained by either response model was lower than in other visual field maps, so the responses here were not

convincingly modulated by visual event timing, and as a result we could not distinguish between response models or quantify response model parameters here. However, visual field mapping in the ventral visual field maps often uses stimuli including objects (faces, buildings, and other objects) to maximize responses[56]. Analogously, responses in the ventral visual field maps may be more strongly modulated by event timing if the events included objects rather than dots alone.

Previous studies of responses to timing have not compared the temporal modulation of responses to visual position selectivity on an individual subject level, which is important given the fine spatial scale of changes in visual position selectivity in visual field maps. A recent meta-analysis found dorsal-ventral gradients of responsivity to spatial and temporal tasks in the intraparietal sulcus and the right frontal operculum and posterior-anterior gradients around the supplementary motor area[57]. We expect this discrepancy in locations to be due to the difference in task demands, as the studies in the meta-analysis were only included if they used a task that was clearly linked to spatial or temporal processing. Conversely, the current study did not require the participants to make any timing-related judgements when measuring temporal modulation of neural responses, nor spatial judgements during visual field mapping. Still, we do find a posterior-anterior gradient, but across the entire brain, such that spatial responses to the temporal contrast of stimuli (i.e., monotonic responses) are found posterior, while timing-selective tuned responses are found more anterior.

The emergence of visual numerosity tuned responses shows striking similarities with the emergence of visual timing tuned responses we describe here. Both numerosity and event timing are quantities, but spatial and temporal quantities respectively. Both show tuned responses in higher-level association areas[19,32]. While these responses largely overlap, there are clear differences in their location. This suggests that, although the different quantities are not processed using the exact same neural populations, there may be similar computational mechanisms for estimating different quantities in the brain. In both cases, early visual responses monotonically follow numerosity and timing, and tuned responses emerge later[39]. However, the transition from monotonic responses to numerosity to numerosity-tuned responses takes place around the lateral occipital areas, rather than the temporal occipital areas that we see for timing. This transition is very sudden in the case of numerosity responses, in contrast to the gradual transition we see in the emergence of timing-tuned responses. As a result, the numerosity-tuned responses are not partly captured by or intertwined with monotonic responses. Therefore, only monotonic numerosity response model fits decrease when moving away from the retinotopic location of the center of the visual field, while the numerosity-tuned response model fits are consistent throughout the visual field.

Our experimental design cannot conclusively demonstrate that timing-tuned responses are derived from monotonic early visual responses, because we don't disrupt early visual responses and show effects on later timing-tuned responses. Nevertheless, several findings suggest timing-tuned responses are derived from monotonic early visual responses. First, almost all visual inputs to the cortex come through V1, which responds monotonically to event duration and frequency. No other known pathway could pass visual inputs to timing-tuned neurons. Second, monotonic response model fits gradually decrease as timing tuned model fits gradually increase through the visual hierarchy, suggesting a transformation from monotonic to tuned responses. Third, the first areas showing evidence for timing tuned responses are the temporal-occipital visual field maps of hMT+/V5[48]. Transcranial magnetic stimulation here decreases visual duration discrimination performance[10], which is thought to depend on duration-tuned neural responses[14].

Our experiments focus on events with durations and periods from 50 to 1000 ms. It remains unclear how the brain responds to visual event timings outside that range. For monotonic responses to event frequency, we would expect that neural response amplitudes summed over time would also increase with event frequency outside this range: each event should produce a neural response, the sum of which should increase with the number of events. Similarly, for monotonic responses to event duration, we would expect increases to continue as duration continues to increase. However, there are likely to be limits in both cases. At very high frequencies, humans cannot distinguish visual events, leading to flicker fusion, and we similarly cannot accurately perceive the durations of events that are faster than neural responses. At very low frequencies or for very long events (for example with periods or durations of minutes), the event's temporal structure is less perceptible, and we must remember how long ago the event began. Our monotonic response models make quantitative predictions outside the tested range, but these predictions are not tested and a single compressive exponent parameter seems unlikely to be able to predict responses over a very large range of time scales. For tuned responses, there is no evidence in the literature of visual event timing tuned responses with preferred durations or periods below 50 ms[19] or above 3000 ms[31] or indeed adaptation effects operating outside these ranges[14]. However, descriptions of visual event timing-tuned neural populations are relatively recent, and further research may reveal such responses.

To begin estimating stimulus timing from visual inputs requires an analysis of neural responses to temporal changes in those inputs. Early retinotopic visual areas will only show such responses at the retinotopic location of the stimulus. However, a true sense of event timing should generalize across all stimulus locations, separating timing information from other stimulus properties. Here we find that the human brain appears to transform simple monotonic responses to timing into tuned temporal representations, where the tuned component is independent of the event's location in the visual field. As such, similar to numerosity, the brain's responses to stimulus timing reflect a transformation and abstraction from low-level sensory information. This abstracted tuned representation of event timing is propagated into brain areas implicated in allocating attention, multisensory integration, and planning actions, suggesting it benefits a wide variety of cognitive functions.

## Methods

**Participants**. All experimental procedures were approved by the ethics committee of University Medical Center Utrecht. We collected data from eight participants (1 female, aged 25–35), which was also used in a previously published experiment[19]. All participants were right-handed and had normal or corrected to normal vision. Two participants were co-authors, familiar with the goals of the study. All participants were briefly trained in duration discrimination tasks before scanning, to encourage attention to stimulus timing and avoid learning or habituation effects at the start of data collection. All participants gave written informed consent and they were financially compensated for their time and travel expenses.

### Stimuli

*Timing mapping stimuli.* All visual stimuli were generated using MATLAB and Psychtoolbox-3[58]. The visual stimuli were presented on a $27.0 \times 9.5$ cm screen inside the MRI bore (resolution $1600 \times 538$ pixels) at 41 cm from the participant's eyes.

Participants were asked to fixate at the center of a red fixation cross that crossed the whole display on a gray background. Visual events comprised the presentation of a filled circle with a diameter of 0.4° that appeared for a variable duration at a variable temporal period[19]. The position of the circle changed pseudo-randomly between stimulus events, but was always within 0.75° of this fixation cross and 0.25° or more away (edge to edge) from the previous position. Every 21 s on average, the presented circle was white rather than the usual black. Participants had to respond by pushing a button when they saw this white target stimulus. No timing judgements were ever required.

Each event timing was repeated in a 2100 ms time frame, such that an entire TR (=2100 ms) contained the same event duration and period. The number of events presented within the 2100 ms between timing changes varied with event period and increments in event period sometimes fell slightly before or after 2100 ms. The maximum drift of event onset timing was only 300 ms, and the increments in event period were only 50 ms, so this deviation was not perceptible. The presented event timing was used for analysis.

To distinguish responses to specific event timings from responses to other stimulus parameters, we used four conditions where event duration and period were related in different ways (Supplementary Movie 1). First, the constant luminance condition, in which event duration and period were equal and both changed together (50–1000 ms, 50 ms steps) (Fig. 1b, d, black points). Second, the constant duration condition, in which event period changed (50–1000 ms, 50 ms steps), while event duration remained 50 ms (Fig. 1b, d, blue points). Third, the constant period condition, in which the event duration changed (50–1000 ms, 50 ms steps), while the period always remained 1000 ms (Fig. 1b, d, red points). In all the aforementioned conditions, increasing progressions of event duration and/or period were followed by a 16.8 s interval of events with a 2100 ms period. The event duration was 50 ms in the constant duration condition and 2000 ms in the other two conditions. These extreme timings help to distinguish very small response functions (which would respond briefly and with low amplitude in the 50–1000 ms range) and very large response functions (which respond continuously and with high amplitude) in fMRI responses[43,59]. Furthermore, these long event durations and periods produce little response from neural populations preferring sub-second timing. Conversely, populations whose response monotonically increases with duration, period, or mean luminance should respond most strongly to these stimuli. After this, the same event timings were presented in a decreasing order, followed by another 16.8 s interval of events with a 50 or 2000 ms duration and a 2100 ms period. We used a single model to capture responses to both increasing and decreasing event duration and/or period progressions. Including responses to both increasing and decreasing timings in the same model counterbalanced adaptation effects with stimuli that give both higher and lower responses preceding presentation of any timing.

Finally, the gaps condition was designed to sample further combinations of event periods and durations (Fig. 1b, d, green points). This stimulus configuration consisted of four progressions with timing changing in 50 ms steps. In chronological order: increasing event durations (50–500 ms) and decreasing event periods (950–500 ms); increasing event durations (50–500 ms) and increasing event periods (550–1000 ms); decreasing event durations (500–50 ms) and increasing event periods (500–950 ms); and decreasing event durations (500–50 ms) and decreasing event periods (1000–550 ms). Each of these progressions was separated by 6.3 s of events with 50 ms duration and 2100 ms period, and the last was followed by 14.7 s with that timing.

We tested each of 24 possible orders of all four stimulus configurations once per participant, so each participant's data included 24 scanning runs, each totaling 470.4 seconds and acquired in four sessions.

*Visual field mapping stimuli.* We acquired visual field mapping responses to examine the relationship between our voxels' responses to visual event timing and visual field position, and to delineate visual field maps. The visual field mapping paradigm was identical to that described in previous studies[32,33]. The stimulus consisted of drifting bar apertures at various orientations, which exposed a moving checkerboard pattern. The stimulus had a radius of 6.35°, much larger than the timing mapping stimuli (0.75° radius). Two diagonal red lines, intersecting at the center of the display, were again presented throughout the entire scanning run. Participants fixated the center of the cross and pressed a button when these lines changed color, and detected 80–100% of the color changes that were presented within each scanning run.

**fMRI data collection and pre-processing.** Acquisition procedures were similar to procedures described elsewhere[33,59]. Briefly, data was acquired with a 7T Philips Achieva scanner with a repetition time (TR) of 2100 ms, echo time (TE) of 25 ms, and a flip angle of 70°. The T1-weighted anatomical scans were automatically segmented with Freesurfer and manually edited to minimize segmentation errors using ITK-SNAP. The T2*-weighted functional scans were acquired using a 32-channel head coil at a resolution of $1.77 \times 1.77 \times 1.75$ mm, with 41 interleaved slices of $128 \times 128$ voxels. The resulting field of view was $227 \times 22 \times 72$ mm. We used a single shot gradient echo sequence with SENSE acceleration factor 3.0 and anterior-posterior encoding, plus a top-up scan with the opposite phase-encoding direction to correct for image distortion in the gradient encoding direction[60]. We also acquired a T1-weighted anatomical image with the same resolution, position, and orientation as the functional data. We used a 3rd-order image-based B0 shim of the functional scan's field of view (in-house IDL software, v6.3, RSI, Boulder, CO, USA). The anterior temporal and frontal lobes were excluded from acquisition due to the fact that 7T fMRI has a low response amplitude and large spatial distortions in these areas. For timing mapping, we used 24 runs, 224 TRs (470.4 s) each, separated into 4 sessions with typically 6 runs each. For visual field mapping, we used 8–10 runs, 182 TRs (382.2 s) each, in a single separate session.

The functional data was co-registered to the anatomical space with fMRI_preproc[61] using AFNI (afni.nimh.nih.gov[62]). A single transformation matrix

was constructed, incorporating all the steps from the raw data to the cortical surface model to reduce the number of interpolation steps to one. A T1 image with the same resolution, position, and orientation as the functional data was first used to determine the transformation to a higher resolution (1 mm isotropic) whole-brain T1 image (3dUnifize, 3dAllineate). For the fMRI data, we first applied motion correction to two series of images that were acquired using opposing phase-encoding directions (3dvolreg). Subsequently, we determined the distortion transformation between the average images of these two series (3dQwarp). We then determined the transformation in brain position between and within functional scans (3dNwarpApply). Then we determined the transformation that co-registers this functional data to the T1 acquired in the same space (3dvolreg). We applied the product of all these transformations to every functional volume to transform our functional data to the whole-brain T1 anatomy. We repeated this for each fMRI session to transform all their data to the same anatomical space.

The resulting data was imported into Vistasoft's mrVista framework[63]. For timing response data, we identified the parts of each scanning run where each stimulus configuration was presented and averaged these fMRI responses together across all runs and sessions[19]. We also separately averaged data from odd and even runs to allow cross-validation in subsequent modelling. For visual field mapping data, we averaged all scan runs together.

**Visual field mapping analysis and visual field map definitions.** We analyzed visual field mapping data following a standard pRF modelling procedure[43,44]. We identified visual field map borders based on reversals in the cortical progression of the polar angle of voxels' visual field position preferences, manually identifying these on an inflated rendering of each participant's cortical surface (Fig. 2 and Supplementary Figs. 3 and 4). These formed our main regions of interest. As well as the early visual field maps (V1, V2, V3, hV4), we identified higher visual field maps in the ventral occipital (VO1, VO2, PHC), lateral/temporal occipital (LO1-LO2, TO1-TO2), parietal association (V3A/B, IPS0-IPS5), and frontal (sPCS1-sPCS2, iPCS) cortices with reference to landmarks identified in previous studies[48,64–66].

**Timing response models.** The current study compared the fits of established monotonic[18] and tuned[19] models of neural responses to visual event timing in different brain areas (Fig. 1 and Supplementary Fig. 1)[67,68]. Each of these models describes a particular relationship between event timing and the neural response amplitude to every event. We predict response amplitudes on a per-event basis, but these responses accumulate over a few seconds due to fMRI's measurement of slow changes in blood flow and oxygenation. The models predicted neural response amplitudes at the offset of events, since this is when the information about the duration and period of the events was completely available to the participants.

These predictions were then convolved with a standard hemodynamic response function (HRF) to construct a prediction for the fMRI response. Then, any free parameters were fit to maximize the correlation between the predicted response and the actual, recorded data (variance explained), giving a fit neural response model. Since HRF parameters substantially differ between participants, the resulting neural response model was used to determine participant-specific HRF parameters as described elsewhere[44]. Specifically, given the neural response model parameters already fit, we found the set of HRF parameters for each participant that maximize the correlation between the predicted and observed fMRI responses over the entire recorded cortex where the neural response model explained more than 10% of the variance in the data. Using these participant-specific HRF parameters, the neural response model's free parameters were refit.

Given that the tuned response model has more free parameters than the monotonic response model, both model fits were cross-validated before comparison. This cross-validation was achieved by fitting each response model's free parameters on the even or odd scans and evaluating the resulting model's fit on the complementary half. Because fMRI response amplitudes change arbitrarily between scans and sessions[69], the scaling between the predicted fMRI time course and complementary scan data was refit during cross-validation.

*Monotonic response model.* The monotonic response model has previously been demonstrated to capture effects of event timing on fMRI responses in early visual areas better than simpler monotonic response models. This model has two components that scale independently with event duration and frequency (1/period)[17]. The frequency and duration components were each scaled by free compressive exponent parameters[18]. The neural response amplitude to each event was given by Eq. (1):

$$\text{Amplitude} \propto \text{Duration}^{expDur} \times \text{AmplitudeRatio} + \frac{\text{Frequency}^{expFreq}}{\text{Frequency}} \quad (1)$$

Where expDur and expFreq are the compressive exponent on duration and frequency respectively, in the range 0–1. AmplitudeRatio captures the relative amplitudes of the duration and frequency components, and was linearly solved by dividing the duration component's response amplitude by the frequency component's response amplitude from a general linear model. The compressive exponents were fit by testing a large set of candidate values to find the parameter combination that best predicted the measured response of each voxel.

We also tested simpler models: a model where there was a linear relationship between duration and response amplitude (so expDur was fixed at 1) and another

model that additionally had a linear relationship between frequency and response amplitude (so expFreq was also fixed at 1)[17]. In a comparison of cross-validated model fits (using a Wilcoxon signed-rank test of the medians in each map), the monotonic model with variable compressive exponents on event duration and frequency predicted responses more closely than monotonic models with linear scaling of response amplitude with event duration (visual field maps: $Z = 15.17$, $p < 0.001$, $n = 481$ pairs; timing maps: $Z = 13.77$, $p < 0.001$, $n = 284$ pairs) or with both event duration and frequency (visual field maps: $Z = 8.27$, $p < 0.001$, $n = 481$ pairs; timing maps: $Z = 8.80$, $p < 0.001$, $n = 284$ pairs). This finding is in line with those of Zhou and colleagues[18]. Therefore, all subsequent analyses used the model with variable compressive exponents as the best fitting monotonic response model.

Note that monotonically decreasing responses would imply a response that decreases with increasing duration or frequency, so these should not be found in the transient and sustained responses of early visual areas. Therefore, we restricted the response amplitudes of the duration and frequency components to positive values. If, for a certain voxel, one of the components' scaling factors (e.g. the response amplitude of the duration component) was fit as negative, we set this scaling factor to 0 and fit the response amplitude to the other component alone. If the other scaling factor (e.g. the response amplitude of the frequency component) was positive after any refitting, these scaling factors were used to compute the amplitude ratio in the final model. If the other scaling factor (e.g. the response amplitude of the frequency component) was negative after any refitting, it was also set to zero and the variance explained by the model in this voxel was zero. As a result, the amplitude ratio cannot be below zero. Since no monotonically decreasing responses were allowed, the monotonically increasing model will be referred to simply as the monotonic response model.

*Tuned response model.* In the tuned response model, the neural response amplitude to each event is described by a two-dimensional anisotropic Gaussian function. This response model describes the timing of each event in terms of duration and period (rather than frequency, i.e. 1/period) because both of these are expressed in seconds and the response function is a Gaussian function of these parameters. The model describes the response amplitudes to each event separately, so at the temporal resolution of fMRI these grouped response amplitudes over several events also increase with event frequency. Therefore the response is scaled by a compressive exponent on frequency. We chose the tuned model response function that best predicted neural responses throughout the timing map network[19]. The function can be described using the following Eqs. (2–4):

$$X = \left(\text{Duration} - \text{Duration}_{\text{pref}}\right) \times \cos(\theta) - \left(\text{Period} - \text{Period}_{\text{pref}}\right) \times \sin(\theta) \quad (2)$$

$$Y = \left(\text{Duration} - \text{Duration}_{\text{pref}}\right) \times \sin(\theta) - \left(\text{Period} - \text{Period}_{\text{pref}}\right) \times \cos(\theta) \quad (3)$$

$$\text{Amplitude} \propto e^{-0.5 \times \left(\left(\frac{Y}{\sigma_{\text{maj}}}\right)^2 + \left(\frac{X}{\sigma_{\text{min}}}\right)^2\right)} \times \frac{\text{Frequency}^{\text{expFreq}}}{\text{Frequency}} \quad (4)$$

The six free parameters of the model are: the preferred duration ($\text{Duration}_{\text{pref}}$) and the preferred period ($\text{Period}_{\text{pref}}$) around which the Gaussian function's mean is centered; the standard deviations along its major and minor axes ($\sigma_{\text{maj}}$ and $\sigma_{\text{min}}$); the angulation of its major axis ($\theta$); and the compressive exponent on frequency to which the response was scaled. The fitting procedure consisted of testing a large combination of free parameters, followed by a gradient descent between the best-fitting parameter combination and its neighbors.

After fitting this model, it is possible that some voxels are best described by a Gaussian function with a preferred duration and/or period on the boundaries and/or outside of the presented stimulus range (i.e. a duration or period below 60 ms or above 990 ms). However, it would be impossible to find the exact preferred duration and/or period belonging to this Gaussian[43]. Also, this would increase the risk that a tuned response model produces monotonic-like responses by simply adhering to large values for these free parameters. Therefore, the variance explained of the tuned response model in voxels where the preferred duration or period was outside the presented stimulus range was set to 0 during cross-validation.

The fit parameters of this tuned response model allowed us to define the borders of timing map regions of interest, as previously described[19]. Here we use the same timing maps defined in that study (Supplementary Fig. 5). Briefly, we took the variance explained values for each vertex on the cortical surface and performed surface-based clustering on these values. In some cases, we merged two adjacent clusters into a single map, or split a single large cluster into two parts where it contained two contiguous maps (common in TTOP/TTOA and TPCS/TPCM).

**Model comparisons.** We excluded from model comparison voxels for which the variance explained of both models was 0.2 or less in the data on which the models were fit. Then, the cross-validated model fits (i.e. fits in data independent from that used for model fitting) were statistically compared to each other in each visual field map and timing map. For each hemisphere, we then took the median model variance explained across the selected voxels in each map, separately for the two cross-validation splits. If a hemisphere did not have any voxels with a variance explained above 0.2 in a specific map and a specific cross-validation split, that map in that hemisphere and cross-validation split was excluded from subsequent

comparison ($n = 1130$ visual field map measurements; $n = 618$ timing map measurements).

To assess how model fits differed between maps, the median variance explained by the two models was then compared using a 3-factor ANOVA (factors: participant, map, and model; interaction: map and model) using MATLAB's anovan function. The interaction factor here demonstrated that different maps' responses were best captured by different response models. As the differences between variance explained of the two models were not normally distributed (Jarque–Bera test with FDR correction), post hoc two-sided Wilcoxon signed-rank tests with FDR correction were then used to assess which model best captured the responses in each map, using hemispheres and cross-validation splits as independent measures. Here, we calculated the effect size as $r = Z / \sqrt{n}$. Furthermore, in order to demonstrate the absence of a difference between models, we computed Bayes factors for each of these nonparametric paired comparisons in JASP[70] with 5 chains of 1000 iterations. As we had no expectations about the prior distribution, we used the default distribution provided by JASP (Cauchy with $r = 1/\sqrt{2}$).

**Model validation.** To test whether our model comparison procedure correctly identified monotonic and tuned responses, we performed a model validation process using simulated responses with a known ground-truth state. First, we generated simulated fMRI responses time courses for both monotonic and tuned responses, with a broad and homogenous distribution of response function parameters in each case, using the same procedure we use to generate candidate fMRI time course predictions during model fitting. We normalized these simulated responses by subtracting their mean amplitude and dividing the resulting amplitudes by their standard deviation, to give simulated responses with a mean amplitude of zero and standard deviation of one. We degraded these responses by adding normally distributed noise with standard deviations between 0 and 6, giving simulated signals with known signal-to-noise ratios from 1/6 to infinite (i.e. no noise). We repeated this twice to give pairs of simulated responses with the same signal but independent noise for our cross-validation procedure. We then multiplied these by the average standard deviation of the observed fMRI responses within our visual field maps and added the mean response amplitude of those observed responses, matching the amplitudes and ranges in our observed responses. We then passed these simulated responses through our response model fitting and comparison procedures, including cross-validation against data with the same signal but independent noise.

For both the simulated monotonic and tuned responses, we then compared the cross-validated variance explained from monotonic and tuned response models. We separated responses where the tuned response model returned preferred event duration and period estimates inside the presented range (which we take as evidence of tuned responses) and outside the presented range (which we do not take as evidence of tuned responses because the responses change monotonically within the presented timing range). For each of these cases, we determined the proportion of responses that were correctly classified as monotonic or tuned, as a function of both signal-to-noise ratio and cross-validated variance explained.

The parameters underlying our simulated data were broadly and homogeneously distributed. However, the tuned response model parameters estimated from our fMRI data had specific distributions that differed between brain areas. We, therefore, asked which ground-truth response function parameters would lead to incorrect response model classification in responses that would pass our threshold of 0.2 variance explained in the best fitting response model in the data on which the response model was fit. We determined the proportion of correctly identified responses for each of the ground-truth response function's parameters, both where preferred event duration and period were estimated inside and outside the presented range.

**Differences in response model parameters between visual field maps.** As well as differing in which response model best captured the measured responses, different parameters within either response model can predict very different responses to the presented event durations and frequencies. For each visual field map, we took the median parameter values from both response models, for each hemisphere and the two cross-validation splits. We excluded from this median voxels for which the variance explained in that cross-validation split and model was 0.2 or less in the data on which the model was fit. In the ratio of response amplitudes of the duration and frequency components of the monotonic response model, voxels without a positive response to one component (i.e. where the ratio of response amplitudes was either 0 or infinite) were excluded from this median. We then compared the population of median parameter values between visual field maps using a two-factor ANOVA (factors: participant, visual field map) using MATLAB's anovan function. As the data was not normally distributed for all parameters (Jarque–Bera test with FDR correction), to determine which visual field maps differed from each other for each parameter, we followed this by post hoc comparisons using Dunn's test[71] with Holm–Šidák correction for multiple comparisons[72].

**Progression of model fits within visual field maps.** To study whether the model fits changed throughout the visual field, variance explained was plotted against the distance to the center of the visual field (eccentricity), where the stimulus was

shown. To achieve this, we first grouped all voxels within a map across hemi-spheres and cross-validation splits. For each visual field map, we then binned voxels according to their preferred eccentricity. Each bin had a range of 0.2°, centered on eccentricities from 0.1° to 5.5°. We excluded voxels for which the variance explained of both models was 0 in the data on which the models were fit. For each bin, the mean cross-validated variance explained and its standard error were computed for each model. Bins with less than 50 voxels were excluded.

To visualize the progression of model variance explained across eccentricity, we fitted a cumulative Gaussian sigmoid function (i.e. greater variance explained at low eccentricities, near the stimulus) and a quadratic function (which also allows a maximum variance explained at higher eccentricity) to the progression in each visual field map. The free parameters for the sigmoid function were point of inflection, slope, maximum, and minimum. We fit the best sigmoid function, and computed the 95% confidence interval of this sigmoid from 10,000 bootstrap iterations. The free parameters for the quadratic function were intercept, slope, and quadratic term. We fit the best quadratic function and computed the 95% confidence interval of the quadratic function linearly using polyfit and polyconf in MATLAB. For visualizing the progression of model fits with eccentricity, we chose the function that was best correlated with the eccentricity bin means.

Note that the quadratic function can also capture a linear relation, so comparing these progressions is only useful for plotting the data rather than determining whether fits decrease with eccentricity. Therefore, to statistically compare the difference between model fits near and far from the center of the visual field, we computed the average variance explained in each hemisphere, each cross-validation split, and each map for eccentricities lower than 1° and eccentricities higher than 2° ($n = 1140$ visual field map measurements). We excluded voxels for which the variance explained of both models was 0 in the data on which the models were fit. To assess how the cross-validated model fits differed between these eccentricity ranges, these averages were then compared using a 3-factor ANOVA (factors: participant, visual field map, and eccentricity range; interaction: visual field map and eccentricity range) using MATLAB's anovan function. The interaction factor here demonstrated that the different visual field maps had better variance explained in different eccentricity ranges. The differences between variance explained at different eccentricity ranges were not normally distributed in all visual field maps (Jarque–Bera test with FDR correction). Therefore, we used post hoc two-sided Wilcoxon signed-rank tests with FDR correction to assess how variance explained differed with eccentricity range in each visual field map, using hemispheres and cross-validation splits as independent measures. Here, we calculated the effect size as $r = Z/\sqrt{n}$. Furthermore, in order to demonstrate the absence of a difference between eccentricity ranges, we computed Bayes factors for each of these nonparametric paired comparisons in JASP[70] with 5 chains of 1000 iterations. As we had no expectations about the prior distribution, we used the default distribution provided by JASP (Cauchy with $r = 1/\sqrt{2}$).

We also computed the difference in variance explained between the monotonic and tuned response models for each voxel (tuned minus monotonic). This quantified the component of the voxel's response that reflects timing tuning and cannot be explained by monotonic responses alone. The eccentricity progression of variance explained differences across the visual field map was also assessed using the statistical procedure described above.

**Reporting summary**. Further information on research design is available in the Nature Research Reporting Summary linked to this article.

## Data availability

The data set described in the study was also used in a previous publication[19]. Ethical constraints prevent us from sharing the medical imaging data sets (MRI scans) generated in the current study to public repositories. The structure of the brain is unique to the individual participant, in theory allowing the participant to be identified from these images, which may also contain medically sensitive findings. This is an interpretation of the EU's General Data Protection Regulation (GDPR) for medical images including MRI data. These raw data sets are available from the corresponding author upon reasonable request within a month, depending on agreements not to share these data publicly. Model parameters underlying all statistical analyses and response data for all model fitting are publicly available at the following DOIs: visual field map parameters (doi.org/https://doi.org/10.6084/m9.figshare.19146131)[73], timing map parameters (doi.org/https://doi.org/10.6084/m9.figshare.17122706)[74], parameters used during validation (doi.org/https://doi.org/10.6084/m9.figshare.17122727)[75], visual field map time series (doi.org/https://doi.org/10.6084/m9.figshare.19146092)[76], timing map time series (doi.org/https://doi.org/10.6084/m9.figshare.17122718)[77] and validation time series (doi.org/https://doi.org/10.6084/m9.figshare.17122748)[78]. Source Data plotted in the Figures are provided with this paper. Source data are provided with this paper.

## Code availability

The code that supports the findings of this study is available from the following repositories: vistasoft (https://github.com/vistalab/vistasoft)[63] vistasoftAddOns (github.com/benharvey/vistasoftAddOns, https://doi.org/10.5281/zenodo.5811114)[67]

fMRI_preproc (github.com/MvaOosterhuis/fMRI_preproc, https://doi.org/10.5281/zenodo.5811116)[61] MonoTunedTiming (github.com/evi-hendrikx/MonoTunedTiming, https://doi.org/10.5281/zenodo.6417921)[68].

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

## Acknowledgements

This work was supported by the Netherlands Organization for Scientific Research (452.17.012 to B.M.H.) and the Helmholtz Institute (PhD funding to E.H. & B.M.H.).

## Author contributions

Conceptualization, J.M.P and B.M.H.; methodology, E.H., J.M.P, M.A., and B.M.H.; software, E.H., J.M.P, M.A., and B.M.H.; validation, E.H., J.M.P., and B.M.H.; formal analysis, E.H., J.M.P and B.M.H.; investigation, E.H., J.M.P., and B.M.H.; data curation, J.M.P. and B.M.H.; writing – original draft, E.H. and B.M.H.; writing – review & editing, E.H., J.M.P, M.A., N.S., and B.M.H.; visualization, E.H., J.M.P., and B.M.H.; supervision, N.S. and B.M.H.; project administration, B.M.H.; funding acquisition, B.M.H.

## Competing interests

The authors declare no competing interests.
