## [Peer Review File · Nature Communications]

Visual timing-tuned responses in human association cortices
and response dynamics in early visual cortexREVIEWER COMMENTS

Reviewer #1 (Remarks to the Author):

In this paper the authors use computational modeling and 7T fMRI data to examine 2 models of timing responses in the visual system: a monotonic model and a time-tuned model of neural responses. The authors conclude that earlier visual areas behave monotonically and higher level areas show separable responses to event period and event duration – which they refer to as time-tuned responses.

While the topic of the paper is interesting, and this is a strong research group, the paper is lacking a theoretical framework and is missing important analyses and key results. Specifically, the results only show how much of the variance is explained for the 2 models - they don't show under which conditions the models succeed or fail, how model parameters vary across the visual system, or compare results to other temporal models of the visual system. Additionally, it seems that the data presented is a re-analysis of the data of Harvey et al, Current Biology, 2020, but is not stated as such. Therefore, I cannot recommend publication at this time.

Major Concerns

- 1) The paper has no theoretical framework and no statement of what hypothesis is being tested. If the goal of the paper is model comparison, then it should be stated as such. If the authors have a different goal, please make that explicit.
- 2) There is no description of the experiment or justification how the experimental manipulation could disambiguate models. I had to read the 2020 paper in order to see the stimuli and understand the experiment. The paper needs to be self-encapsulated even if the author reanalyze prior data from the lab. The authors also need to be transparent about the fact that the data in this paper has been published before.
- 3) There is no explanation of the models in figure 1 or biological justification of their validity. It is also unclear why it is important and what is implication if the neural response is monotonic or temporally tuned.
- 4) Related to (1) if the goal of the paper is model comparisons, the authors should compare the data to additional temporal models of the visual system including (i) standard linear model, (ii) 2-channel quadratic model (Stigliani 2017), (iii) temporal compression (Zhou 2018), (iv) dynamic divisive normalization (Zhou 2019), and (v) 2-channel model with compression and adaption (Stigliani 2019).
- 5) While they can find that the temporal tuned model better explains the variance of visual responses than the monotonic model, there is insufficient evidence that this reflects a qualitative rather than quantitative difference between early and higher level areas.
- 6) Models are useful because they explain how different voxels/regions behave under different conditions. So presenting the model fits as presented in the paper is necessary but this is just the first step of the results. The interesting results that are missing are (i) understanding how

model parameters vary across the visual system (ii) under which conditions the models succeed or fail, and (iii) what is the implication of these parameters on how different visual areas process visual stimuli with different durations and periods

7) The presentation of the results can be improved by (i) explaining the models and their biological basis, (ii) explaining the experiment and how it can disambiguate models, (iii) adding justifications and predicted outcomes for each section, and (iv) reporting all the intricate stats in tables rather than the main text; I found these long paragraphs of stats unreadable.

Reviewer #2 (Remarks to the Author):

This paper aimed to determine whether and where monotonic responses (i.e., a gradual increase of response according to the increase of period/duration) are transformed into timing-tuned responses and how the monotonic and time-tuned responses are related to the visual field position. To address this issue, the authors collected fMRI data while participants were exposed to visual stimuli of various frequencies and durations. The data was analyzed using a neural response model that combined a tuned response model (Harvey et al., 2020) with monotonic response models for duration and frequency. The relationship between monotonic /timing-tuned responses and location preferences was examined in each visual field map. The authors reported that there is a gradual transition from monotonic response to tuned response in the visual processing hierarchy: while early visual cortices responded monotonically, higher visual regions, beginning in area MT/V5, exhibited timing-tuned responses. Importantly, while monotonic responses in early visual cortices were tightly associated with the location preferences, timing-tuned responses were more abstract and observed independent of the location preferences.

Overall, I think this study was conducted with a sophisticated design and analysis, and the results were clearly presented. I find the main claim of this study is mostly supported by the results and the findings may potentially advance the knowledge of the field. Nevertheless, I have some major concerns regarding the methods and interpretations of the results. Also, there are some limitations in the data analysis which should clearly be stated in the discussion section (if correct). The details are listed below.

Comments:

How does the neural response model for monotonic response can simultaneously and independently capture the effect of 'temporal summation' (i.e., increase of BOLD response according to the increase of frequency in stimulus presentation) and the increase of BOLD response according to the increase of 'period'? Since the period is the inverse of frequency, the prediction for BOLD response would be opposite according to the change in stimulus frequency parameter. I might be missing something but would appreciate it if you could clarify.

Fig. 1. It is not clear to me why fMRI response predictions for the monotonic response for

duration and frequency have to be 'weighted sum' ('ratio' in panel g). To create fMRI response predictions, do they need to be integrated rather than being modeled separately? Please clarify.

Please clarify the rationale of using the liberal threshold of $R^2 = 0.2$ to map monotonic and timing-tuned responses. To me, $R^2 = 0.2$ appears arbitrary and too low to claim that the model was 'able to predict' the fMRI responses. I would like to see some justification to use this threshold.

One of my major concerns is that this study appears not providing any direct evidence that time-tuned responses were "derived" from monotonic responses. This claim requires some computational explanation regarding how time-tuned responses can be derived from monotonic responses and the supporting evidence for it. If the evidence is rather indirect, this claim should be toned down.

In this study, timing-tuned responses were not associated with the location preferences. How does this finding reconcile with the behavioral studies showing that timing-tuned cells may be spatially specific (Fulcher et al., 2016 Proc Roy Soc B; Li et al., 2021 Brain Res)?

There is an fMRI decoding study (Hayashi et al., 2018 Commun Biol) showing that, while the occipital cortex represents time by monotonic responses, parietal and frontal cortex represent time by multivoxel activity patterns (which may reflect timing-tuned representations as reported in the Hayashi et al., 2015 PLOS Biol). I think their results are highly relevant and consistent with the present study, so it would be important to discuss and clarify what the present study would advance the previous knowledge.

If I understand correctly, the neural response model for timing-tuned response would not be able to predict the response of the voxels if timing-tuned neurons with different timing preferences were intermixed in the same voxel (thus such voxels would be missed in the results). This point might be trivial for the authors but not necessarily so for the readers, so this limitation should be noted in the manuscript.

Fig. 1, legends: Legends for a. and b. are erroneously switched.

Reviewer #3 (Remarks to the Author):

This study uses forward-modelling of fMRI responses to infer the encoding of fast temporal properties of visual stimuli, duration and frequency (period). The authors compare a monotonic model where the fMRI response increases steadily with duration and frequency with a "tuned" model where neuronal populations are selective for these temporal stimulus properties. Comparisons across the extended hierarchy of visual cortical regions, they find that the monotonic model explains data in earlier visual regions but they observe greater evidence for the tuned model in higher parietal regions. This is an elegant and well-executed study and the findings should be of wider interest. However, I think there are a few issues that the authors need to resolve before publication (I apologise for the length of this review but while extensive, most of these comments can probably be dealt with fairly swiftly):

1. Validating models

The description of the model fit on page 30, lines 663-675 suggests a fairly complex decision tree dealing with contingencies for when particular parameter fits fail/fall outside the range. I'd be worried about some unforeseen consequences of this. To me this is a perfect case of where some validation with a simulated ground truth would be highly informative (see e.g. Lerma-Usabiaga et al, 2020, PLoS Comp Biol). This is generally good advice (trust me, I'd know...) but it is particularly crucial for novel models on as-yet untested tuning properties such as this. This need not be an enormous addition and perhaps can even be relegated to the Supplementary Information but some validation of this would help a great deal. One particular issue is how models perform depending on the signal-to-noise ratio. Does the pattern of results observed empirically simply arise because of different noise levels in higher vs early regions? More generally, it can also help instill confidence that the models can actually capture these parameters in the first place.

2. Comparing variance explained

In Figures 3 and 6, the authors show the variance explained by the two models across the range of brain areas tested. The results are quite clear but something is odd. Only voxels with $R^2 > 0.2$ for either model were included. I interpret this that at least one of the models was above threshold for the voxel to be included (logical-OR operation). It must be because otherwise the means of any condition could never be below this threshold.

However, this logical-OR data selection is susceptible to statistical artifacts because the data selection is not independent. Consider if condition A is on average lower than condition B: by selecting only the voxels that are above 0.2 in A or B you will artifactually inflate the mean of condition B. It amplifies the difference already there. As far as I can tell, it should not produce a difference where isn't one - so this is probably not a major concern - but it may mean that the distinction of the models isn't actually as clear as shown here. My suggestion is to redo this analysis but for -all- voxels with $R^2 > 0$ (as the authors already do in some of the later analyses in Fig 5). It doesn't really make sense to threshold the data here anyway if the aim is to compare the goodness-of-fit of the two models.

On a smaller but related point, it seems the authors take the arithmetic mean of R^2 across a ROI for each participant. R^2 distributions are distinctly non-normal so it makes more sense to use the median or another robust estimator of central tendency for that (but this probably makes no practical qualitative difference).

3. Clarity of results

The presentation of some of the results could be clearer. I understand why the authors want to show everything they show in Figure 1 but this diagram is not going to be easy for a reader to digest, especially someone who isn't versed in pRF-type analyses. For one thing, it would help adding titles over the three columns labelling them as monotonic duration, frequency, and the tuned model. More generally, I wonder if less isn't more here. You could have a figure outlining the modelling more conceptually in the main text, and relegate this more detailed one to the supplementary information. Or perhaps it would help to split this figure. In any case, it would probably help running this by some non-expert readers to aid understanding.

In similar vein, there are a lot of details in the text that detract more than they help

understanding. I'm not a big fan of tables but the statistical inferences reported across a large range of brain areas, complete with means, SDs, t-stats, and confidence intervals are a perfect example of information better shown as tables. Otherwise you have these massive blocks of statistical text, like in lines 168-183, 265-279, 285-297, 309-322, 360-369 (I hope I got them all). I'd turn these into tables and keep them in the Supplementary Info.

4. Eccentricity effects

The authors show interesting findings w.r.t. to how the models can explain data from peripheral pRFs far from the actual stimulus location. This raises an obvious question how this relates to cortical magnification and/or larger pRF sizes. In terms of functional significance, in peripheral vision spatial sensitivity is poorer while temporal sensitivity is enhanced. The authors already discuss a magno- vs parvocellular account but it isn't entirely clear what the interaction could be. Related to this, higher regions have larger receptive fields so more peripheral voxels will be stimulated by the central stimulus there while it is less the case in earlier regions. Could this explain the absence of the relationship with eccentricity in higher regions?

On that note, I am a bit confused by the confidence intervals in Figure 4a. Is the dashed lines we see for V1 and V2 really the upper bound of the CI for the fit here? I assume the lower bound is effectively at zero? The bins have narrow error bars and the means are far away from this CI. This seems implausible but I'm probably missing something. In general, since the dashed lines are impossible to discern in most of the panels I would suggest using a filled region to denote the confidence interval (transparent or behind the line plot) rather than the dashed lines.

5. Missing ROIs

The authors seemingly analyse every visual and timing-related region under the sun. For that reason it seems odd that as far as I can tell there is no mention of V4 anywhere in the manuscript. Also, how were the ROIs in Figure 4 chosen because some of them are absent from this plot as well (such as LO1). This is not a big issue but it should probably be discussed somewhere.

6. Typo on page 12, line 232: captured -> capture

7. Data availability

I don't want to make the authors' lives more difficult but given NC is championing data transparency I believe a statement that data will be shared upon "reasonable request" is not good enough. What requests are "reasonable"? Who decides this? I entirely understand that there are legal and ethical limits to data sharing but this must be specified clearly. Quoting the journal's guidelines:

"...reasons for controlled access (eg., privacy, ethical/legal issues), conditions of access must be described precisely including contact details for access requests, timeframe for response to requests, restrictions imposed on data use via data use agreements. [...] Restrictions on controlled access datasets including restrictions on downstream data reuse or authorship requirements must be clearly described in manuscript and to editors at the time of submission."

Sam Schwarzkopf
University of Auckland

REVIEWER COMMENTS

Reviewer #1 (Remarks to the Author):

In this paper the authors use computational modeling and 7T fMRI data to examine 2 models of timing responses in the visual system: a monotonic model and a time-tuned model of neural responses. The authors conclude that earlier visual areas behave monotonically and higher level areas show separable responses to event period and event duration – which they refer to as time-tuned responses.

While the topic of the paper is interesting, and this is a strong research group, the paper is lacking a theoretical framework and is missing important analyses and key results. Specifically, the results only show how much of the variance is explained for the 2 models - they don't show under which conditions the models succeed or fail, how model parameters vary across the visual system, or compare results to other temporal models of the visual system. Additionally, it seems that the data presented is a re-analysis of the data of Harvey et al, *Current Biology*, 2020, but is not stated as such. Therefore, I cannot recommend publication at this time.

We are happy to see that the reviewer finds our study interesting. We address the specific points mentioned above in our point-by-point reply below.

Major Concerns

1) The paper has no theoretical framework and no statement of what hypothesis is being tested. If the goal of the paper is model comparison, then it should be stated as such. If the authors have a different goal, please make that explicit.

We agree that our hypotheses and their predictions were not introduced clearly enough. Our hypothesis is that the visual timing-tuned responses that we have previously observed throughout the human association cortex are related to, and emerge from, responses of the early visual areas that monotonically increase with event duration and frequency (which have also been described recently). Methodologically, we test this hypothesis by comparing the predictions of monotonic and tuned models to the responses in different brain areas. The results show a gradual transition from monotonic to tuned responses across the visual processing hierarchy. As such, the goal of the paper is not model comparison, but rather the model comparisons are used to test the predictions of a specific theoretical framework.

We describe this hypothesis as follows in the abstract (new text underlined) (p. 2, lines 23 – 24):

“Here we ask how these timing-tuned responses are related to the responses of early visual cortex, which monotonically increase with event duration and frequency.”

We considered this hypothesis against predominant models for the brain's estimation of a sensory event's timing based on the assumption that there are central neural processes that respond to passing time (like ticking pacemakers or the dynamics of neural activity). We propose that the neural response to passing time is not a centralized process, but rather

emerges from the activity of early sensory areas. We now outline this distinction in the first paragraph of the introduction, as follows (p. 3, lines 38 - 43) :

“Currently, predominant models for the brain’s estimation of an event’s timing rely on central neural processes responding to passing time, like ticking pacemakers or the dynamics of neural activity (Creelman, 1962; Treisman, 1963; Church, 1984; Grondin, 2001, Hartcher-O’Brien et al., 2016). However, following converging behavioral, computational and neuroimaging results (Karmarkar & Buonomano, 2007; Ivry & Schlerf, 2008; Buetti et al., 2008; Morrone et al., 2005; Bruno et al., 2010; Bruno et al., 2015; Heron et al., 2015; Maarseveen et al., 2017; Roseboom et al., 2019), we hypothesized that the brain may derive a representation of event timing from the dynamics of neural responses to that event in sensory processing areas, rather than relying on specialized central pacemakers or processes.”

The hypothesis that visual timing-tuned responses emerge from monotonic responses of the early visual areas that reflect the dynamics of sensory processing makes a specific prediction: monotonic responses should be limited to the retinotopic location of the stimulus within these early visual brain areas. Timing-tuned responses, on the other hand, may then be largely abstracted from stimulus position, particularly as these are topographically organized by event timing rather than visual field position. Methodologically, we test this hypothesis by comparing the model fits in different retinotopic locations for both models. Again here, model comparisons are used to test the predictions of our specific theoretical framework.

We now explain how these predictions follow from our central hypothesis as follows (p. 3, lines 55 – 58):

“Following our hypothesis that the brain’s representation of a visual event’s timing emerges from response dynamics in retinotopically organized early visual areas, we predicted that these monotonic responses would be restricted to the stimulus’s retinotopic location.

After introducing timing-tuned responses, we then go on to introduce the prediction that these would be abstracted from stimulus position, as follows (p. 4, lines 73 – 76):

“As these quantity-tuned responses are topographically mapped by the state of the stimulus quantity, rather than its position, we predicted that timing-tuned responses would not be restricted to the stimulus’ position on the sensory organ, but encoded in an abstracted, quantity-based frame of reference.”

Having introduced our hypotheses and their predictions, we then end the introduction by stating our research questions and outlining our methodological approach (because the full methods section is at the end of the manuscript). We also use this paragraph to explicitly state that we are reanalyzing an existing data set, as follows (p 5., lines 88 – 96):

“Here we ask whether and how monotonic and tuned neural responses to visual event timing are related throughout the brain’s hierarchy of both timing maps and visual field maps. We further ask how both monotonic and tuned responses to timing are related to visual position preferences. We answer these questions by reanalyzing ultra-high-field (7T) fMRI data that was acquired during the presentation of repetitive visual events which gradually varied in event duration and/or period in our previous study (Harvey et al., 2020). We compare the fits of monotonically increasing and tuned neural response models

throughout the brain and investigate how these relate to visual spatial selectivity throughout the visual field map hierarchy.”

2) There is no description of the experiment or justification how the experimental manipulation could disambiguate models. I had to read the 2020 paper in order to see the stimuli and understand the experiment. The paper needs to be self-encapsulated even if the author reanalyze prior data from the lab. The authors also need to be transparent about the fact that the data in this paper has been published before.

We agree that it was not possible to understand the experiment without reference to either the methods section (at the end of the manuscript) or our previous study. Looking again, the brief description of the methodology at the end of the introduction (quoted at the end of our last response) was not enough. We agree that any manuscript should contain all the information needed to understand it at the first reading. We have now addressed this issue by expanding the description of the experiment, and justification of how its design lets us disambiguate monotonic and tuned response models as follows (p. 6, lines 111 – 131):

“During fMRI scanning, we showed a circle repeatedly appearing and disappearing (visual events). Because of fMRI’s slow time scale, we varied event timing gradually in both duration and/or frequency, allowing the resulting measurements to distinguish response amplitudes to different event timings. Both event duration and period ranged from 50 ms to 1 s in 50 ms steps. Event duration changes are inevitably coupled with changes in either mean display luminance (if events begin at regular intervals) or event period (if each event is immediately followed by another). We therefore characterized four stimulus configurations (Fig. 1b, Supplementary Movie 1) to distinguish between responses to event duration, event period and display luminance. The constant luminance configuration matched duration and period so an event was always ongoing. The constant duration configuration fixed all events’ durations at 50 ms while their periods varied. The constant period configuration fixed all events’ periods at 1000 ms while their durations varied. Finally, the gaps configuration varied both duration and period to sample timings absent in other configurations. Participants made no duration or period judgments, but reported when white circles were presented rather than black (performance >80%). This happened pseudo-randomly, equally frequently for all timings.

This stimulus set let us disambiguate monotonic and tuned response models by testing these models’ predictions against the response amplitudes seen for many combinations of event duration and frequency. For each voxel, we used both monotonic and tuned response models (Fig. 1 and Supplementary Fig. 1) to predict the time course of response amplitudes to the presented event timings...”

We then continue to describe our model fitting process in detail as in the original submission.

3) There is no explanation of the models in figure 1 or biological justification of their validity. It is also unclear why it is important and what is implication if the neural response is monotonic or temporally tuned.

We now explain in more detail how the monotonic response models we describe are related to known transient and sustained components of early visual responses (i.e. we outline their

biological validity), and how these are sufficient for the computation of event duration and frequency, as follows (p. 3, lines 46 – 55):

“Early visual cortical responses increase monotonically but sub-linearly with event duration and frequency (and therefore decrease with event period, i.e. 1/frequency). This can be described in terms of the summed amplitude of transient and sustained neural responses respectively (Stigliani et al 2017; Zhou et al, 2018), for a stimulus of fixed strength (e.g. fixed contrast). But these early visual response components’ amplitudes both also increase with stimulus strength, so don’t unambiguously represent event timing. Nevertheless, their amplitudes provide a signal from which event timing can be quantified: the ratio of sustained to transient response amplitude gives event duration (if both scale similarly with stimulus strength), while the sum of transient responses over time is proportional to event frequency (at fixed contrast or after contrast normalization).”

Tuned responses are found throughout visual processing, and are closely linked to perception in the representation of other stimulus properties such as visual motion direction, edge orientation or position. We have recently described timing-tuned responses in an extensive network of brain areas implicated in visual motion processing, attentional allocation, multisensory integration and action planning (Harvey et al, 2020). While it is not clear exactly how these responses are computed, similar timing-tuned responses have also been described by other groups in human fMRI (Hayashi et al, 2015; Hayashi & Ivry, 2020; Protopapa et al., 2019) and cat single-neuron recordings (Duysens et al., 1996). So again there is existing evidence of their biological validity. These timing-tuned responses have also been linked to timing perception (Hayashi & Ivry, 2020), so overall seem to be a higher-level, more abstracted representation of event timing that underlies perception of visual event timing. This is the main implication of whether a neural response to timing is tuned or monotonic. We now explain this as follows (p. 4, lines 59 – 66):

“Beyond early visual areas, an extensive network throughout the human association cortices shows visual timing-tuned responses, with maximum response amplitudes at specific preferred durations (the time from event onset to offset) and periods (the time between repeating event onsets; i.e., 1/frequency) (Harvey et al, 2020). Tuned neural responses are common throughout sensory and cognitive processing, and are closely linked to the perception of many stimulus features (Blakemore & Cooper, 1970; DeAngelis, Cumming & Newsome 1998), including visual motion (Salzman, Britten & Newsome 1990), somatosensory vibrational frequency (Romo, Hernández, Zainos & Salinas, 1998) and other quantities like numerosity (Kersey & Cantlon, 2017; Nieder & Miller, 2004; Piazza, Izard, Pinel, Le Bihan & Dehaene, 2004) (recently reviewed by Tsouli et al., 2021). Even for visual event timing, changes in duration-tuned responses and duration perception are linked (Hayashi et al., 2015; Hayashi et al., 2018; Hayashi & Ivry, 2020).”

4) Related to (1) if the goal of the paper is model comparisons, the authors should compare the data to additional temporal models of the visual system including (i) standard linear model, (ii) 2-channel quadratic model (Stigliani 2017), (iii) temporal compression (Zhou 2018), (iv) dynamic divisive normalization (Zhou 2019), and (v) 2-channel model with compression and adaption (Stigliani 2019).

One of the goals of our study is indeed model comparison, specifically determining where a timing-tuned response function is necessary to predict the fMRI responses and where a monotonic response function is sufficient. Although we did not describe this in the original submission, we chose the monotonic function that is best able to predict the neural responses: this model contained compressive exponents on both event duration and frequency. Comparisons to simpler models containing linear relationships between response amplitude and duration and/or frequency were not included here, because Zhou and colleagues (2018) had already shown that a response model with compressive exponents on both parameters captures early visual responses better. Instead, we cited Zhou and this point and say (in the Methods, new clarifying text underlined) (p. 40, lines 857 – 861):

“The monotonic response model has previously been demonstrated to capture effects of event timing on fMRI responses in early visual areas better than simpler monotonic response models. This model has two components that scale independently with event duration and frequency (1/period) (Stigliani et al., 2017). The frequency and duration components were each scaled by free compressive exponent parameters (Zhou et al., 2018).”

Zhou and colleagues (2018) formulate the mathematics of the compressive monotonic model slightly differently from ours because they describe the timing of their events in terms of interstimulus interval rather than period (i.e. interstimulus interval + duration). For tuned responses, we have previously shown (Harvey et al., 2020) that a Gaussian function in a duration and period space captures tuned responses better than a Gaussian function in a duration and interstimulus interval space. We prefer to be consistent between our monotonic and tuned models, rather than using a compressive exponent on interstimulus interval (which would not make sense as it ignores the effect of event duration on event frequency). Zhou and colleagues determine the “length of ISI required for the response to two 100 ms pulses to approach the linear prediction.” Nevertheless, our model’s predictions are equivalent to Zhou et al. (2018), the only difference being that we treat event period as a single component while Zhou et al. separate the same period into an event duration and an ISI, so capture some of the variance in event period in their event duration component.

In the revision, we have expanded on this by showing that, also in our data, the model with compressive nonlinearities on both duration and frequency is the best fitting monotonic model across both the visual field maps and timing maps. Specifically, we now explain in the methods that (p. 41, lines 871 – 882):

“We also tested simpler models: a model where there was a linear relationship between duration and response amplitude (so $expDur$ was fixed at 1) and another model that additionally had a linear relationship between frequency and response amplitude (so $expFreq$ was also fixed at 1) (Stigliani et al., 2017). In a comparison of cross-validated model fits (using a Wilcoxon signed-rank test of the medians in each map), the monotonic model with variable compressive exponents on event duration and frequency predicted responses more closely than monotonic models with linear scaling of response amplitude with event duration (visual field maps: $Z = 15.59$, $p < 0.001$, $n = 455$ pairs; timing maps: $Z = 13.77$, $p < 0.001$, $n = 284$ pairs) or with both event duration and frequency (visual field maps: $Z = 8.43$, $p < 0.001$, $n = 455$ pairs; timing maps: $Z = 8.80$, $p < 0.001$, $n = 284$ pairs). This finding is in line with those of Zhou and colleagues (2018). Therefore, all subsequent analyses used the model with variable compressive exponents as the best fitting monotonic response model.”

The fourth response model the reviewer suggests (dynamic divisive normalization), does indeed capture responses better when fit on ECoG data. However, fitting the parameters of this model relies on a stimulus sequence with a highly variable relationship between the timing of the previous event and the timing of the current event. Our stimulus sequence does not have such variable relationships because it is designed for fMRI experiments and uses repetitive events. Indeed, Zhou and colleagues (2019) state that “A limitation of this model primarily lies within the measurement method—fMRI samples coarsely in time, and allows only for tests on the sum, instead of the dynamics, of the time course predictions.” Therefore, we did not fit a model with dynamic divisive normalization on the current fMRI data.

The fifth response model (a 2-channel model with compression and adaptation, Stigliani et al., 2019) includes additional complexity by having a compressive sigmoid nonlinearity on responses to event onset and offset. When onsets and offsets are close in time (either when onset precedes offset, i.e. short duration, or offset precedes onset, i.e. short interstimulus interval) these interact to reduce response amplitude. This has a similar effect to a compressive exponential nonlinearity on event duration. However, because the transient responses are modeled using biphasic functions, it is possible in their model for an intermediate event frequency to produce larger responses than both a higher and lower event frequency. As such, this model is not limited to a monotonic relationship between event frequency and response amplitude. As our fundamental question is where monotonic responses change to tuned responses, we therefore do not evaluate the predictions of this model.

5) While they can find that the temporal tuned model better explains the variance of visual responses than the monotonic model, there is insufficient evidence that this reflects a qualitative rather than quantitative difference between early and higher level areas.

Our results clearly demonstrate a gradual transition from monotonic to tuned responses. This is a major result of our study. We propose in the discussion that monotonic and tuned responses are found together throughout the processing hierarchy, but contribute to the population response to different extents in different areas. As such, we do not claim that there is a qualitative difference between early visual areas and later visual areas, except that very early areas show no evidence at all of timing tuned response components. It is exactly the quantitative and gradual progression of responses that is the core finding.

The gradual hierarchical changes in model parameters that we show in response to the reviewer’s next comment further highlight this gradual transition of quantitative response properties. However, there are a couple of parameters that do actually show a more qualitative difference (see Figure 4, copied into our next response). First is the exponent on the frequency component of the monotonic model. Here, visual field maps where monotonic models best predict responses consistently have high exponents (near 1) while those where tuned models fit best have far lower exponents (below 0.5). This is consistent with a transition from responses that simply increase amplitude with frequency to responses that can peak at specific frequencies. The frequency exponent on the tuned model shows a similarly sudden change. Second is the response function extent (major sigma) in the tuned

model. In visual field maps where monotonic models predict responses best, this tuning function is consistently larger than maps where tuned models fit better. This appears to be because monotonic responses do not actually have a peaked response function. Therefore, there are some qualitative differences between the responses of areas with monotonic and tuned responses, but primarily the results show a gradual and quantitative transition from one to the other.

6) Models are useful because they explain how different voxels/regions behave under different conditions. So presenting the model fits as presented in the paper is necessary but this is just the first step of the results. The interesting results that are missing are (i) understanding how model parameters vary across the visual system (ii) under which conditions the models succeed or fail, and (iii) what is the implication of these parameters on how different visual areas process visual stimuli with different durations and periods

We agree that these are interesting aspects of our data and we are happy to show them to give a more comprehensive description of our data. We did not include these model parameters in different visual areas in our original submission because they have been shown before by previous studies analysing monotonic responses in many of the visual field maps (Zhou et al., 2018) and tuned responses in the timing maps (Harvey et al., 2020). Therefore, these additions are in some cases replications, but it is nice to show how our results are consistent with previous studies. Importantly, we show for the first time in great detail how the parameters of the tuned response model progress over various visual field maps. Furthermore, we show additional visual field maps in our analysis of the monotonic response model's parameters compared to Zhou and colleagues, who often grouped visual field maps (TO maps, LO maps, and IPS maps). We have not included the model parameters the timing maps here, because we have shown these before with the same data and ROIs (Harvey et al., 2020).

We describe the additional analyses on model parameters in the method section as follows (p. 45 - 46, lines 991 – 1004):

“Differences in response model parameters between visual field maps

As well as differing in which response model best captured the measured responses, different parameters within either response model can predict very different responses to the presented event durations and frequencies. For each visual field map, we took the median parameter values from both response models, for each hemisphere and the two cross-validation splits. We excluded from this median voxels for which the variance explained in that cross-validation split and model was 0.2 or less in the data on which the model was fit. In the ratio of response amplitudes of the duration and frequency components of the monotonic response model, voxels without a positive response to one component (i.e. where the ratio of response amplitudes was either 0 or infinite) were excluded from this median. We then compared the population of median parameter values between visual field maps using a two-factor ANOVA (factors: participant, map). To determine which visual field maps differed from each other for each parameter, we followed this by post-hoc comparisons using Dunn's test (Dunn, 1964) with Holm–Šidák correction for multiple comparisons (Šidák, 1967).”

We included the following section (including a new figure, now Figure 4) to show how model parameters change between visual field maps (p. 14 – 18, lines 250 - 343):

“Changes in response model parameters between visual field maps

Both response models can capture very different relationships between event timing and response amplitude using different free parameters. Differences between visual field maps in monotonic response model parameters (Stigliani et al., 2017; Zhou et al., 2018) and between timing maps in tuned response model parameters (Harvey et al., 2020) have been described in previous studies, tuned model parameters in the visual field maps have not been described. We therefore grouped the voxels in each visual field map, took the median model parameter in each visual field map and used a two-factor ANOVA to assess how these parameters differed between visual field maps and participants.

In the monotonic response model, the ratio of the response amplitudes of the duration component divided by the frequency component differed significantly between visual field maps ($F_{(16, 403)} = 4.41$, $p < 0.001$, $\eta_p^2 = 0.15$) (Fig. 4a). Post-hoc multiple comparisons showed this ratio progressively decrease from early to later visual field maps. This is consistent with previous reports (Stigliani et al., 2017), where this result is proposed to reflect a faster decrease in the amplitudes of sustained than transient neural response components through the visual processing hierarchy. The second model parameter, the compressive exponent on event duration, also differed significantly between visual field maps ($F_{(16, 438)} = 2.29$, $p = 0.003$, $\eta_p^2 = 0.08$) (Fig. 4b), though there was no evidence of a systematic progression through the visual hierarchy. The final parameter, the compressive exponent on event frequency, also differed significantly between visual field maps ($F_{(16,438)} = 12.33$, $p < 0.001$, $\eta_p^2 = 0.31$) (Fig. 4c). Post-hoc multiple comparisons showed this parameter progressively decreased from early to later visual field maps (consistent with previous reports (Zhou et al., 2018)), particularly maps where tuned models predict responses better than monotonic models (Fig. 3). Therefore, up the visual hierarchy responses to repeated events are progressively integrated until response amplitudes no longer increase monotonically with event frequency and instead show tuned responses that peak at specific frequencies.

Similarly, for the tuned response model, the compressive exponent on event frequency differed between visual field maps ($F_{(16,500)} = 55.24$, $p < 0.001$, $\eta_p^2 = 0.64$) (Fig. 4d), again decreasing from early to later visual field maps where tuned responses become dominant. This decrease is consistent with a decrease among the timing map hierarchy (Harvey et al., 2020), which largely overlaps with the visual field map hierarchy from LO1 onwards. We found that the median preferred event duration and period differed significantly between visual field maps (duration: $F_{(16, 500)} = 3.80$, $p < 0.001$, $\eta_p^2 = 0.11$; period: $F_{(16, 500)} = 2.97$, $p < 0.001$, $\eta_p^2 = 0.09$) (Fig. 4e and 4f), though without a systematic hierarchical progression in either case, again in agreement with a lack of progression through the timing map hierarchy (Harvey et al., 2020). The major and minor extent of the tuned response function also differed significantly between visual field maps (major: $F_{(16, 500)} = 2.72$, $p < 0.001$, $\eta_p^2 = 0.08$; minor: $F_{(16, 500)} = 4.81$, $p < 0.001$, $\eta_p^2 = 0.14$) (Fig. 4g). Post-hoc multiple comparisons showed a decrease in the major extent and an increase in the minor extent of the response function from early to later visual field maps, with the major extent highest and the minor response extent lowest where monotonic models predict responses better than tuned models. This may be because these monotonic responses have no response peak, so responses were best predicted by a function that responds similarly to many event timings to capture the large monotonic changes in amplitude with event frequency. Because of these

simultaneous but opposite changes in the response function's major and minor extents, their ratio also differed between visual field maps ($F_{(16, 500)} = 12.45, p < 0.001, \eta_p^2 = 0.29$) (Fig. 4h) and progressed similarly to the response function's major extent, again consistent with changes through the timing map hierarchy.

The response function's orientation also differed significantly between visual field maps ($F_{(16, 500)} = 6.65, p < 0.001, \eta_p^2 = 0.18$) (Fig. 4i). Post-hoc multiple comparisons showed a progressive decrease along the visual hierarchy, such that the response function was narrower in the duration direction in the early visual field maps (particularly where monotonic models fit better than tuned models) and narrower in the period direction in later visual field maps. Where the monotonic model fit better, again this orientation appears to describe the tuned response function that best captures monotonic changes in response amplitude. However, where tuned responses fit better, there is hierarchical transformation of timing-tuned responses to give a finer representation of event period than event duration (as in the timing map hierarchy (Harvey et al., 2020)). This may provide better information for frontal premotor areas to plan precisely timed actions to interact with visual events.

We can visualize how these differences in response model parameters predict different response amplitudes to different event timings by taking the median response model parameters in each visual field map and rendering response functions with these parameters (Fig. 4j-k). We restrict this to possible event timings, where event duration is less than event period. These response functions change considerably between visual field maps. Nevertheless, in all visual field maps, these two response models predict very different response functions regardless of the response function parameters within that map.

Fig. 4 Comparison of monotonic and tuned response model parameters between visual field maps. *a* In the monotonic response model, the ratio of response amplitudes of the duration component divided by the frequency component decreased along the visual field map hierarchy. *b* The duration component's compressive exponent did not systematically change along the visual hierarchy. *c* The frequency component's compressive exponent decreased along the visual hierarchy. *d* In the timing tuned response model, the compressive exponent on event frequency similarly decreased along the visual hierarchy. *e-f* The median preferred event duration and period did not systematically change along the visual hierarchy. *g-h* There was a decrease in the major extent and an increase in the minor extent of the response function (and consequently a decrease in their aspect ratio) along the visual hierarchy, particularly changing where tuned response models begin to dominate later visual field maps' responses. *i* The response function's orientation progressively rotated along the visual hierarchy, so the response function was narrower in the duration direction in the early visual field maps (particularly where monotonic models fit better than tuned models) and narrower in the period direction in the later visual field maps. Points show the median parameter value across recordings from different hemispheres, error bars show 95%

confidence intervals computed from 1000 bootstrap iterations. Brackets and stars show significant differences in multiple comparisons between all maps: all brackets to the left of the star are significantly different from all brackets to the right of the star (two-sided Dunn's test, corrected for multiple comparisons, $p < 0.05$). n indicates the number of hemispheres included for the monotonic or tuned response model in each visual field map. n deviates for the monotonic component's ratio, as maps with all voxels' ratios of 0 or infinite were excluded. Light colors show visual field maps where the shown response model captured responses significantly less well than the other response model. j-k Visualizations of response functions given by the median response model's parameters in representative visual field maps. Source data are provided as a Source Data file."

We discuss some aspects of these parameters in the discussion, specifically a small difference between our parameters and those reported by Zhou and colleagues that appears to result from a difference in the way our models parameterize event timing (p. 26 – 27, lines 521 - 531):

"In contrast to these previous experiments, we characterized the time between events in terms of the frequency of event onsets, rather than interstimulus interval (the time from one single event's offset to another single event's onset). This describes our stimulus design more straightforwardly because we use a repeating periodic stimulus that includes a large variation in event durations (the time from event onset to offset) for any particular frequency. However, unlike in Zhou and colleagues, we found no decrease in event duration's compressive exponent along the visual field map hierarchy. This appears to be because Zhou and colleagues separate an event's period into two components, event duration and interstimulus interval. By doing this, event period's compressive exponent may be reflected in both their components, while in our approach it is all captured by the event period component and so does not affect the event duration component."

Finally, the reviewer also asks "under which conditions the models succeed or fail". In response to Reviewer 3's first comment, we have now performed an extensive model validation procedure using simulated responses in which the parameters generating the responses are known. This allows us to test how accurately we can identify which response function the responses actually follow, how this is affected by different levels of noise, and how it is affected by model parameters. There are extensive changes to the methods and results to explain this procedure, which are quoted in the reply to Reviewer 3's first comment if the reviewer wants to see the details. But briefly, we summarize the findings of this validation procedure in the discussion as follows, highlighting situations in which the response may be misclassified by our procedure (p. 29 – 30, lines 589 – 605):

"Furthermore, our model validation procedure (Supplementary Fig. 2) demonstrated that ground-truth monotonic responses are very rarely classified as tuned. This gives high confidence that visual field maps identified as showing tuned responses do indeed show tuned responses, and indeed their response function parameters in tuned response models should facilitate identification as tuned responses. However, ground-truth tuned responses can be misclassified as monotonic when their response function approximates a monotonic function: where the preferred timing is outside the presented timing range, the response function is very large, or the compressive exponent on event frequency is high. Therefore, for early visual field maps identified as showing monotonic responses, some borderline

cases may actually have tuned responses that our procedure cannot reliably identify. Specifically, tuned response models fit here have high estimates of the compressive exponent and slightly larger response functions than later visual field maps showing tuned responses. This does not strongly affect our conclusions because such tuned response functions closely resemble monotonic response functions, there is a gradual transition from monotonic to tuned responses, and tuned responses may have monotonic components. Nevertheless, it is possible that tuned responses emerge slightly earlier in the visual hierarchy than our model comparison suggests.”

7) The presentation of the results can be improved by (i) explaining the models and their biological basis, (ii) explaining the experiment and how it can disambiguate models, (iii) adding justifications and predicted outcomes for each section, and (iv) reporting all the intricate stats in tables rather than the main text; I found these long paragraphs of stats unreadable.

We hope the reviewer agrees that points (i) and (ii) have now been explained in our previous replies. Regarding point (iv), we agree with the reviewer that tables are preferable here, and other reviewers also made this point. We have moved all of the detailed stats into tables 1-3. There are a lot of these stats to include to comply with the journal’s requirements.

Regarding point (iii), we agree the justification and predictions for each section could be improved. We now carry the theoretical framework described in previous replies into each section of the results.

Beginning with the results section “Transition from monotonic to tuned responses to timing along the visual field map hierarchy”, we now start this section by clarifying our goal for this section, our hypothesis and its predictions as follows (p. 5 - 6, lines 104 – 111):

“We first asked whether and how monotonic and tuned neural responses to visual event timing are related throughout the hierarchy of both timing maps and visual field maps. We hypothesized that visual event timing-tuned responses are likely to be derived from the inherent modulation of visual responses by event timing: monotonic increases in early visual responses with event duration and frequency. This predicts a transition from monotonic to tuned response functions through the visual field map hierarchy. To determine whether each visual field map’s responses followed monotonic or tuned functions, we recorded neural responses to visual events of variable timing.”

We follow this by describing the stimulus in the experiment, as quoted in our previous reply.

In the section “Changes in response model parameters between visual field maps”, we prefer not to present our predictions as based on specific hypotheses resulting from our theoretical framework, because we largely predict that the parameters would change between visual field maps as described in previous studies. Our hypotheses do not really make predictions about these parameters. We begin by explaining our motivation for this analysis, as follows (p. 14, lines 251 – 255):

“Both response models can capture very different relationships between event timing and response amplitude using different free parameters. Differences between visual field maps in monotonic response model parameters (Stigliani et al, 2017; Zhou et al, 2018) and between timing maps in tuned response model parameters (Harvey et al., 2020) have been described in previous studies, tuned model parameters in the visual field maps have not been described.”

In the section, “Relationship between model fits and retinotopic location” we now begin by explaining our predictions in relation to theoretical framework of the introduction, as follows (p. 18, lines 346 – 352):

“Since the monotonic response model performed well in early visual field maps, it is likely that monotonic responses are computed from low-level stimulus properties. This predicts they would be limited to the retinotopic location of the stimulus (the central visual field representation), rather than elsewhere (the peripheral visual field representation). Furthermore, as timing tuned responses are topographically mapped by their timing preferences (Harvey et al., 2020; Protopapa et al., 2019), we predicted that they would not be restricted to stimulus’s retinotopic location but instead be encoded in an abstracted, quantity-based frame of reference.”

In the final results section, “Transition from monotonic to tuned responses to timing along the timing map hierarchy” we now introduce our motivation and predictions as follows (p. 23 - 24, lines 458 – 476):

“In previous work the locations of timing-tuned responses were defined as timing maps, as the tuned response model performed best over the brain as a whole (Harvey et al., 2020). Such timing maps overlap with visual field maps, but do not have the same borders so do not include the same set of voxels. In the previous analyses we have used visual field map borders, which may include both a spatial representation responding monotonically to temporal contrast and a temporal representation with timing-tuned responses. The gradual transition from monotonic to increasingly timing-tuned responses along the visual field map hierarchy suggests a similar gradual transition along the timing map hierarchy. Therefore, to focus on the temporal representation more specifically, we compared the models specifically within the timing map borders.”

Reviewer #2 (Remarks to the Author):

This paper aimed to determine whether and where monotonic responses (i.e., a gradual increase of response according to the increase of period/duration) are transformed into timing-tuned responses and how the monotonic and time-tuned responses are related to the visual field position. To address this issue, the authors collected fMRI data while participants were exposed to visual stimuli of various frequencies and durations. The data was analyzed using a neural response model that combined a tuned response model (Harvey et al., 2020) with monotonic response models for duration and frequency. The relationship between monotonic /timing-tuned responses and location preferences was examined in each visual field map. The authors reported that there is a gradual transition from monotonic response to tuned response in the visual processing hierarchy: while early visual cortices responded

monotonically, higher visual regions, beginning in area MT/V5, exhibited timing-tuned responses.

Importantly, while monotonic responses in early visual cortices were tightly associated with the location preferences, timing-tuned responses were more abstract and observed independent of the location preferences.

Overall, I think this study was conducted with a sophisticated design and analysis, and the results were clearly presented. I find the main claim of this study is mostly supported by the results and the findings may potentially advance the knowledge of the field. Nevertheless, I have some major concerns regarding the methods and interpretations of the results. Also, there are some limitations in the data analysis which should clearly be stated in the discussion section (if correct). The details are listed below.

We thank the reviewer for their interest in our study and their positive appraisal of our manuscript. We are happy to address their comments point by point below.

Comments:

How does the neural response model for monotonic response can simultaneously and independently capture the effect of 'temporal summation' (i.e., increase of BOLD response according to the increase of frequency in stimulus presentation) and the increase of BOLD response according to the increase of 'period'? Since the period is the inverse of frequency, the prediction for BOLD response would be opposite according to the change in stimulus frequency parameter. I might be missing something but would appreciate it if you could clarify.

The reviewer is absolutely correct that response amplitudes in the monotonic model should increase with increasing event frequency (and therefore decrease with increasing period), and indeed they do. We understand that it can be difficult at times to follow the distinction between frequency and period. In short, in our timing tuned models we use period as a metric for quantifying frequency because period is expressed in seconds (like duration also is) rather than in hertz (as frequency itself is). Conversely, in our monotonic response models we use frequency (in hertz) to quantify this because we hypothesize that there are distinct responses to every event, which accumulate monotonically at the slow temporal resolution of fMRI.

The second and third paragraphs of the introduction introduce these two response models and how each treats frequency. As this is rather brief, it may not have been clear how period is related to frequency. We have clarified this as early as possible in the manuscript by beginning the second paragraph as follows (new text underlined) (p. 3, lines 46 – 47):

“Early visual cortical responses increase monotonically but sub-linearly with event duration and frequency (and therefore decrease with event period, i.e. 1/frequency).”

To clarify the distinction between the parameters of these models, we have also changed the methods to clarify how the tuned model uses both period and frequency as parameters, and why, as follows (p. 42, lines 898 – 903):

“In the tuned response model, the neural response amplitude to each event is described by a 2-dimensional anisotropic Gaussian function. This response model describes the timing of each event in terms of duration and period (rather than frequency) because both of these are expressed in seconds. The model describes the response amplitudes to each event separately, so at the temporal resolution of fMRI these grouped response amplitudes over several events also increase with event frequency.”

Fig. 1. It is not clear to me why fMRI response predictions for the monotonic response for duration and frequency have to be 'weighted sum' ('ratio' in panel g). To create fMRI response predictions, do they need to be integrated rather than being modeled separately? Please clarify.

In the monotonic response model, we have two predictors, event duration and event frequency. These are in different units, seconds and hertz, and we have no reason to hypothesize a particular proportionality of these responses. In other words, we don't have any reason to assume an increase in BOLD response amplitude per increase of 1 Hz would correspond to a particular increase per second of event duration. Indeed, Stigliani and colleagues (2017) show that this proportionality is different between visual field maps. Therefore, these two predictors (for the increase per second and per hertz) are independent components in a general linear model, which calculates their scaling factors. The predictors are then multiplied by these scaling factors (i.e. weighted) and summed. Therefore, the scaling factors are determined by the same GLM and then integrated, it is simply that we don't fix their scaling factors. In other words, for these two predictions, scaling factor 2 can be seen as scaling factor 1 multiplied by some ratio.

This is somewhat different in our cross-validation process. In evaluating the resulting model's goodness of fit in the 'test' half of the data, we allow a change in the scaling factor between the 'training' and 'test' data, because fMRI data can vary considerably in response amplitudes between different measurements. However, the tuned model has only one scaling factor because it contains only one prediction. The monotonic model has two scaling factors because it contains two predictions. This would give an extra degree of freedom to the monotonic response model, so the model comparison would be biased. To deal with this, during cross-validation we take the ratio of these two scaling factors as estimated in the training data, account for this ratio, and then sum the two scaled predictions to give a single prediction. We describe this ratio as a free parameter throughout, and say that all free parameters were fixed for cross-validation.

In the manuscript, we don't want to rely on our readers being familiar with the calculations inside a GLM. Therefore, we don't talk about the scaling between the response prediction and the fMRI data where we can avoid it. For example, we say 'any free parameters were fit to **maximize the correlation between the predicted response and the actual data**', as correlation is easier to understand than a GLM for many readers. Similarly, in Equation 1 of the Methods we say that the response amplitude is proportional to the parameters of the Equation. Within this framework, in Figure 1G and also Equation 1 of the methods, we avoid talking about scaling factors for different predictors by treating this as a ratio of the responses to the two predictors. Again, for these two predictions, scaling factor 2 can be seen as scaling factor 1 multiplied by some ratio.

For a reader that is familiar with general linear models, we explain this in describing the monotonic model as follows (with reference to Equation 1) (p. 40, lines 865 – 868):

“AmplitudeRatio captures the relative amplitudes of the duration and frequency components, and was linearly solved by dividing the duration component’s response amplitude by the frequency component’s response amplitude from a general linear model.”

In the revision, we clarify what the ratio means in Figure 1 in the figure caption by adding (p. 9, lines 162 - 167):

“The monotonic and tuned predictions were compared to the recording from each voxel. The free parameters of both the monotonic and tuned response models were found that maximized the correlation (R^2) between predicted and measured fMRI response time courses. For the monotonic response model these parameters were the compressive exponents on duration and frequency components, and the weighting of these two components (ratio). This ratio is the ratio of the scaling factors for the two components in a general linear model.”

Please clarify the rationale of using the liberal threshold of $R^2 = 0.2$ to map monotonic and timing-tuned responses. To me, $R^2 = 0.2$ appears arbitrary and too low to claim that the model was 'able to predict' the fMRI responses. I would like to see some justification to use this threshold.

This threshold is low because it is applied to cross-validated model fits from only half of the data. Rather than claiming that a voxel with a cross-validated variance explained of 0.2 is able to predict responses (which we did not say), we are aiming to find voxels whose response is convincingly modulated by event timing at all. Notably, in the visual field map analysis there are large parts of each visual field map (in the peripheral visual field representation) that show no response at all, so including these would reduce our ability to distinguish between models of the response we do observe. Other voxels may have some response modulation, but their responses would not allow us to convincingly distinguish between models or accurately determine representative model parameters in each map. On the other hand, in the model comparison in our previous study (Harvey et al., 2020, Current Biology) we found that the timing-tuned model fit the responses in the timing maps with a mean cross-validated variance explained of 0.32 in this data set, so a much higher threshold would exclude a lot of voxels that seem to have some information about which model performs better.

Furthermore, we performed a model validation (see Reviewer 3’s first comment for details) which showed that monotonic responses were almost always correctly classified when the variance explained in the data on which the models were fit was above 0.2. The correct classification of tuned responses also increases towards a variance explained of 0.2 in the data on which the models were fit and then remains relatively stable. We now explain why we use this threshold in the manuscript as follows (p. 9 - 10, lines 172 – 187):

“We validated our procedure’s ability to distinguish monotonic and tuned responses by generating simulated responses that followed either monotonic or tuned response functions with a broad and homogenous range of response function parameters and different

levels of noise (Supplementary Fig. 2). This showed that monotonic responses were almost always correctly identified when their variance explained (R2) was above 0.2 in the data on which the models were fit. Tuned responses were also reliably identified using the same variance explained threshold of 0.2. [...].

Our subsequent analyses therefore excluded voxels where the variance explained was 0.2 or less for both models in the data on which the models were fit (to avoid selecting voxels using the cross-validated fit values we compared). This excludes many voxels in each visual field map (often far from the retinotopic location of the stimulus) where responses do not systematically vary with timing, while including voxels where responses were convincingly modulated by event timing.”

Nevertheless, our choice of threshold does not really affect our conclusions regarding where in the visual field map hierarchy timing-tuned responses arise, the only analysis where a threshold was applied. We have now added supplementary figures (below) showing that very similar patterns of significant differences in model fits across visual field maps is found whether the threshold is set at 0, 0.2 or 0.4. After describing how the model fits compare in different visual fields, we now state that (p. 12, lines 225 – 226):

“Similar results were found at variance explained thresholds of 0, 0.2 and 0.4 (Supplementary Figs. 6 and 7).”

“Supplementary Fig. 6: Model comparisons using medians per hemisphere show similar results with a variance explained threshold above 0 for both visual field maps and timing maps. The threshold for voxel selection was a variance explained of either 0 (left column), 0.2 (middle column), or 0.4 (right column) for the best fitting model in the data on which the model was fit. A threshold of 0.2 and 0.4 result in a gradual transition from monotonic to tuned responses, starting in similar areas. The threshold of 0 deviates from this pattern because there are plenty of voxels within the maps that don't convincingly respond at all to any model. In all cases, the median value of cross-validated variance explained was taken after thresholding in each measured visual field map example. Points show the median across measured visual field map examples, error bars show 95% confidence intervals (from 1000 bootstrap iterations) and two-sided Wilcoxon signed rank tests were used. * $p < 0.05$ ** $p < 0.01$, and *** $p < 0.001$, FDR corrected for multiple comparisons. n indicates the amount of hemispheres included in the comparison in each map. Source data are provided as a Source Data file.”

“Supplementary Fig. 7: Model comparisons using means per hemisphere show similar results regardless of the variance explained threshold used for both visual field maps and timing maps. The threshold for voxel selection was a variance explained of either 0 (left column), 0.2 (middle column), or 0.4 (right column) for the best fitting model in the data on which the model was fit. All thresholds result in a gradual transition from monotonic to tuned responses, starting in similar areas. In all cases, the mean value of cross-validated variance explained was taken after thresholding in each measured visual field map example. At a threshold of 0, these means were not normally distributed, so points show the median across measured visual field map examples, error bars show 95% confidence intervals (from 1000 bootstrap iterations) and two-sided Wilcoxon signed rank tests were used. At other thresholds, the means were normally distributed, points show the mean across measured visual field map examples, error bars show the standard error of the mean, and two-sided paired t-tests were used. * $p < 0.05$ ** $p < 0.01$, and *** $p < 0.001$, FDR corrected for multiple comparisons. n indicates the amount of hemispheres included in the comparison in each map. Source data are provided as a Source Data file.”

One of my major concerns is that this study appears not providing any direct evidence that time-tuned responses were "derived" from monotonic responses. This claim requires some computational explanation regarding how time-tuned responses can be derived from monotonic responses and the supporting evidence for it. If the evidence is rather indirect, this claim should be tone down.

We agree that we do not show causally that tuned responses are derived from monotonic responses, and it is generally not possible to show causation in fMRI experimental designs. We have therefore changed the wording in several places (including the title) to make clear that tuned responses appear to be derived from earlier monotonic responses, but that we cannot directly demonstrate that. We therefore present this hypothesis more tentatively throughout to avoid suggesting to the reader that we conclusively show this derivation. To clarify this point, we sometimes see our study as exploring the relationship between these two types of responses.

These changes are as follows. We have changed the title to (p. 1, lines 1- 2):

“Visual timing-tuned responses in human association cortices and response dynamics in early visual cortex”

In the abstract (p. 2, lines 23 – 24, lines 26 – 28, lines 29 - 32):

“Here we ask how these timing-tuned responses are related to the responses of early visual cortex, which monotonically increase with event duration and frequency.”

“Therefore, across successive stages of visual processing, timing-tuned response components gradually become dominant over the inherent modulation of sensory responses by event timing.”

“We propose that this hierarchical emergence of timing-tuned responses from sensory processing areas quantifies sensory event timing while abstracting temporal representations from the spatial properties of their inputs.”

In the introduction, we have also changed the wording of our research question (p. 4, lines 88 – 89):

“Here we ask whether and how monotonic and tuned neural responses to visual event timing are related throughout the brain’s hierarchy of both timing maps and visual field maps.”

We have also changed the titles of a couple of sections in the results (p. 5, lines 102 - 103):

“Transition from monotonic to tuned responses to timing along the visual field map hierarchy”

and (p. 23, line 457):

“Transition from monotonic to tuned responses to timing along the timing map hierarchy”

In the Results section, we also change how we describe the motivation of one analysis (p. 19, lines 366 – 367):

“The gradual progression from monotonic to tuned response model fits along the visual field map hierarchy suggests tuned responses could be derived from monotonic inputs.”

While we are happy to explain this idea more speculatively and make clear that our results do not establish that tuned responses are necessarily derived from monotonic responses, we still feel that this is how many readers will interpret our results, and that this needs some discussion. Towards the end of the discussion, we therefore explain why our results do not conclusively show this relationship, but also why this relationship nevertheless seems likely from our results and the context of other studies. We explain this as follows (p. 33, lines 683 – 694):

“Our experimental design cannot conclusively demonstrate that timing-tuned responses are derived from monotonic early visual responses, because we don’t disrupt early visual responses and show effects on later timing-tuned responses. Nevertheless, several findings suggest timing-tuned responses are derived from monotonic early visual responses. First, almost all visual inputs to the cortex come through V1, which responds monotonically to event duration and frequency. No other known pathway could pass visual inputs to timing-tuned neurons. Second, monotonic response model fits gradually decrease as timing tuned model fits gradually increase through the visual hierarchy, suggesting a transformation from monotonic to tuned responses. Third, the first areas showing evidence for timing tuned responses are the temporal-occipital visual field maps of hMT+/V5 (Amano, Wandell & Dumoulin, 2009). Transcranial magnetic stimulation here decreases visual duration discrimination performance (Bueti, Bahrami & Walsh, 2008), which is thought to depend on duration-tuned neural responses (Heron, Aaen-Stockdale, Hotchkiss et al., 2012).”

Regarding the computational mechanism for this apparent transformation, it is not yet clear how timing-tuned responses are computed. Many of the proposed mechanisms (like pacemaker-accumulator models) do not specifically rely on activity of sensory cortices, although converging recent evidence (including this study, and work using TMS and adaptation) suggests timing-tuned responses are derived from the activity of sensory cortices. More work is required to determine these mechanisms, which are beyond the scope of the current study. We do not yet want to take a position on this point in print. However, in processes that are better understood, like the derivation of similar numerosity-tuned responses, tuned responses have been convincingly demonstrated to be derived from earlier monotonic responses.

In this study, timing-tuned responses were not associated with the location preferences. How does this finding reconcile with the behavioral studies showing that timing-tuned cells may be spatially specific (Fulcher et al., 2016 Proc Roy Soc B; Li et al., 2021 Brain Res)?

We agree that the spatial spread of repulsive temporal adaptation effects is relevant here, as these adaptation effects are usually understood to reflect timing-tuned neural responses. The study by Li and colleagues is in the tactile sensory domain. We do not yet know how tactile timing-selective neural responses are related to visual timing-tuned responses, so we don't discuss how this tactile adaptation effect relates to our results. However, the study by Fulcher and colleagues is more relevant, as are two other studies on the spatial spread of the visual duration aftereffect (Maarseveen, J., Hogendoorn, H., Verstraten, F. A., & Paffen, C. L. (2017). An investigation of the spatial selectivity of the duration after-effect. *Vision research*, 130, 67-75; Li, B., Yuan, X., Chen, Y., Liu, P., & Huang, X. (2015). Visual duration aftereffect is position invariant. *Frontiers in Psychology*, 6, 1536).

Towards the end of the discussion, we now address the aforementioned studies as follows (p. 30 – 31, lines 618 – 642):

“Only the timing-tuned response component is independent of the retinotopic location of the neural population: monotonic components of the same neural population’s response decrease with retinotopic distance from the stimulus location. In perception, visual duration aftereffects, which are thought to reflect duration-tuned neural responses (Heron et al., 2012), may have a limited though large spatial spread (Fulcher et al., 2016), though others studies show an unlimited spread within and between visual hemifields (Li, Yuan, Chen, Liu, & Huang, 2015; Maarseveen et al., 2017). If the visual duration aftereffects depends on effects on timing-tuned neural populations and is spatially limited, this appears inconsistent with retinotopically-independent timing-tuned response components. This spatial limit may result from interactions between timing-tuned neurons and retinotopically-specific monotonically responding neurons in the same neural population, or effects of adaptation on earlier monotonic responses from which timing-tuned responses are derived. Alternatively, timing-tuned response components might depend on retinotopic location if we mapped a larger area of the visual field: in studies quantifying the duration aftereffect’s spatial spread, at 5° from the adapter it is around 50% of the strength at the adapted location (Fulcher et al., 2016), , and our visual field mapping stimulus covered only 5.6° from the stimulus area. So, timing-tuned neurons may have large receptive fields that the stimulus must fall within

(Fulcher et al., 2016). However, analogous numerosity aftereffects in visual perception (Burr & Ross, 2009, Current Biology) appear to reflect changes in the responses of numerosity-tuned neurons (Piazza et al., 2004; Tsouli et al., 2021) and have a similarly limited spatial spread (Zimmerman, 2018), but numerosity-tuned neurons do not require spatial receptive fields overlapping the stimulus area (Harvey, Fracasso et al, 2015; Harvey & Dumoulin, 2017; Viswanathan & Nieder, 2020; Paul et al., 2021). Therefore, relationships between the spatial spread of perceptual quantity adaptation effects and the receptive fields of quantity-tuned neural populations may be more complex than they appear: rather than perceptual quantity aftereffects providing clear evidence demonstrating that quantity-tuned responses follow spatial receptive field properties, neural tuning for different quantities (Harvey, Fracasso et al., 2015; Hofstetter et al., 2021) and visual position (Harvey, Fracasso et al, 2015; Harvey & Dumoulin, 2017; Viswanathan & Nieder, 2020; Paul et al., 2021) seem to be independent dimensions of neural responses.”

There is an fMRI decoding study (Hayashi et al., 2018 Commun Biol) showing that, while the occipital cortex represents time by monotonic responses, parietal and frontal cortex represent time by multivoxel activity patterns (which may reflect timing-tuned representations as reported in the Hayashi et al., 2015 PLOS Biol). I think their results are highly relevant and consistent with the present study, so it would be important to discuss and clarify what the present study would advance the previous knowledge.

We agree, these are certainly some of the most closely related studies to our current findings, and it was an oversight not to discuss how they are related. We have added a paragraph to the discussion to cover these points, as follows (p. 27, lines 545 – 558):

“A network of previously described areas then shows timing-tuned responses (Harvey et al., 2020). A similar network allows event duration decoding from multivoxel activity patterns (Hayashi et al, 2018). Early visual and lateral occipital areas also allow such decoding, apparently from the monotonic responses we show in these areas. While we find widespread monotonic responses, in that study these were limited to the occipital pole, i.e. the early visual cortex. This may be because our monotonic responses can reflect an increase in response amplitude with event frequency (not only duration) and allow a nonlinear increase in response amplitude with duration. Previous fMRI experiments have also shown repetition suppression by repeated presentation of the same event duration (consistent with duration tuning) in the supramarginal gyrus of the inferior parietal lobule (Hayashi et al, 2015), near our TPCI and TLS timing maps. Here we extend these findings by demonstrating that timing-tuned responses may be derived from the response dynamics of visual field maps, demonstrating contributions of both tuned and monotonic response components in many areas, and demonstrating relationships between these responses and visual spatial responses.”

If I understand correctly, the neural response model for timing-tuned response would not be able to predict the response of the voxels if timing-tuned neurons with different timing preferences were intermixed in the same voxel (thus such voxels would be missed in the results). This point might be trivial for the authors but not necessarily so for the readers, so this limitation should be noted in the manuscript.

We agree it is important to make this clear. Following directly from the paragraph quoted above, we have now added the following (p. 28, lines 559 – 565):

“Either monotonic or tuned responses should allow decoding of event timing, though only tuned responses should produce repetition suppression. Some spatial separation of neural populations with different timing preferences (for example in topographic maps (Protopapa et al, 2019; Harvey et al, 2020)) is required to allow decoding of event timing (Hayashi et al, 2018), or measure different timing preferences at the spatial scale of fMRI (Protopapa et al, 2019; Harvey et al, 2020). Importantly, this does not require that an fMRI voxel contains only neurons with similar timing preferences, simply that neurons with different timing preferences are not homogeneously intermixed.”

Fig. 1, legends: Legends for a. and b. are erroneously switched.

Thanks for pointing this out. The captions are not switched, there is a mistake that makes this hard to follow: the sentence ‘In each case, a compressive exponent parameter captures a non-linear scaling with increases in event duration (left) or frequency (middle and right).’ should be part of the caption for panel a rather than b. We have fixed this as follows (p. 8, lines 146 – 151):

‘a Monotonic response model (left) and tuned response model (right). In each case, a compressive exponent parameter captures a non-linear scaling with increases in event duration (left) or frequency (middle left and middle right). The monotonic response predictions for event duration and frequency changes are combined as a weighted sum. Together, these make a prediction of the per-event response amplitude for every event timing shown in the stimulus (dots).’

Reviewer #3 (Remarks to the Author):

This study uses forward-modelling of fMRI responses to infer the encoding of fast temporal properties of visual stimuli, duration and frequency (period). The authors compare a monotonic model where the fMRI response increases steadily with duration and frequency with a "tuned" model where neuronal populations are selective for these temporal stimulus properties. Comparisons across the extended hierarchy of visual cortical regions, they find that the monotopic model explains data in earlier visual regions but they observe greater evidence for the tuned model in higher parietal regions. This is an elegant and well-executed study and the findings should be of wider interest. However, I think there are a few issues that the authors need to resolve before publication (I apologise for the length of this review but while extensive, most of these comments can probably be dealt with fairly swiftly):

We thank the reviewer for their enthusiasm and interest in our study. We agree that some of these issues are relatively straightforward to address, but we are happy the reviewer takes their time to express their concerns in detail.

1. Validating models

The description of the model fit on page 30, lines 663-675 suggests a fairly complex decision tree dealing with contingencies for when particular parameter fits fail/fall

outside the range. I'd be worried about some unforeseen consequences of this. To me this is a perfect case of where some validation with a simulated ground truth would be highly informative (see e.g. Lerma-Usabiaga et al, 2020, PLoS Comp Biol). This is generally good advice (trust me, I'd know...) but it is particularly crucial for novel models on as-yet untested tuning properties such as this. This need not be an enormous addition and perhaps can even be relegated to the Supplementary Information but some validation of this would help a great deal. One particular issue is how models perform depending on the signal-to-noise ratio. Does the pattern of results observed empirically simply arise because of different noise levels in higher vs early regions? More generally, it can also help instill confidence that the models can actually capture these parameters in the first place.

We have now added a model validation section. In this model validation we show the pattern of the data both when the parameters fall inside and outside of the range. Furthermore, we show what happens to the percentage of correct classifications for various signal-to-noise ratios.

We describe this in the method section as follows (p. 44 - 45, lines 956 – 989):

“Model Validation

To test whether our model comparison procedure correctly identified monotonic and tuned responses, we performed a model validation process using simulated responses with a known ground-truth state. First, we generated simulated fMRI responses time courses for both monotonic and tuned responses, with a broad and homogenous distribution of response function parameters in each case, using the same procedure we use to generate candidate fMRI time course predictions during model fitting. We normalized these simulated responses by subtracting their mean amplitude and dividing the resulting amplitudes by their standard deviation, to give simulated responses with a mean amplitude of zero and standard deviation of one. We degraded these responses by adding normally distributed noise with standard deviations between 0 and 6, giving simulated signals with known signal-to-noise ratios from $\frac{1}{6}$ to infinite (i.e. no noise). We repeated this twice to give pairs of simulated responses with the same signal but independent noise for our cross-validation procedure. We then multiplied these by the average standard deviation of the observed fMRI responses within our visual field maps and added the mean response amplitude of those observed responses, matching the amplitudes and ranges in our observed responses. We then passed these simulated responses through our response model fitting and comparison procedures, including cross-validation against data with the same signal but independent noise.

For both the simulated monotonic and tuned responses, we then compared the cross-validated variance explained from monotonic and tuned response models. We separated responses where the tuned response model returned preferred event duration and period estimates inside the presented range (which we take as evidence of tuned responses) and outside the presented range (which we do not take as evidence of tuned responses because the responses change monotonically within the presented timing range). For each of these cases, we determined the proportion of responses that were correctly classified as monotonic or tuned, as a function of both signal-to-noise ratio and cross-validated variance explained.

The parameters underlying our simulated data were broadly and homogeneously distributed. However, the tuned response model parameters estimated from our fMRI data had specific distributions that differed between brain areas. We therefore asked which ground-truth response function parameters would lead to incorrect response model classification in responses that would pass our threshold of 0.2 variance explained in the best fitting response model in the data on which the response model was fit. We determined the proportion of correctly identified responses for each of the ground-truth response function's parameters, both where preferred event duration and period were estimated inside and outside the presented range."

Furthermore, we describe the results of this model validation in the result section as follows (new text underlined) (p. 9 - 10, lines 172 – 192):

"We validated our procedure's ability to distinguish monotonic and tuned responses by generating simulated responses that followed either monotonic or tuned response functions with a broad and homogenous range of response function parameters and different levels of noise (Supplementary Fig. 2). This showed that monotonic responses were almost always correctly identified when their variance explained (R^2) was above 0.2 in the data on which the models were fit. Tuned responses were also reliably identified using the same variance explained threshold of 0.2. In both cases, responses were more likely to be classified as monotonic where the best fitting tuned response function approximated a monotonic response function: where its preferred duration or period estimate was outside the presented range, its extent was very large, or its compressive exponent on event frequency was high.

Our subsequent analyses therefore excluded voxels where the variance explained was 0.2 or less for both models in the data on which the models were fit (to avoid selecting voxels using the cross-validated fit values we compared). This excludes many voxels in each visual field map (often far from the retinotopic location of the stimulus) where responses do not systematically vary with timing, while including voxels where responses were convincingly modulated by event timing. We also classified voxels where tuned models gave preferred duration or period estimates outside the presented range (i.e. below 60 ms or above 990 ms) as showing monotonic responses, setting their variance explained to zero to avoid using a tuned response model that approximates a monotonic response function.

We then compared these models' cross-validated fits, in data independent from that on which model parameters were fit. We found a clear ..."

We added the following figure as supplementary figure 2:

“Supplementary Fig. 2: Proportion of correctly classified simulated monotonic and tuned responses during model validation. *a* Simulated responses following known monotonic and tuned functions were passed through our response model fitting and comparison procedures. The proportion of each simulated data set classified correctly increased when less noise was added. This proportion was higher for simulated monotonic responses. For simulated tuned responses, a higher proportion was correctly classified when the estimated parameters in the tuned response model were inside the range allowed by our tuned response model voxel selection. At higher signal-to-noise ratios, a higher proportion of simulated monotonic responses were correctly classified when the estimated parameters in

the tuned response model were outside the range allowed by our tuned response model voxel selection. In both cases, this demonstrates that our tuned response model voxel selection correctly excludes tuned responses that cannot reliably be distinguished from monotonic responses. b The same data ordered by the variance explained in the best-performing model in the data on which the model was fit (i.e. before cross-validation). This demonstrates that the ability to correctly classify both response types was consistent above our variance explained threshold (0.2) then dropped sharply. n indicates the amount of simulated data points and is identical for panel a and b. c Proportion of correctly classified tuned responses changes with various parameters of the underlying known response function, in all responses with a variance explained above 0.2. Some parameters reduced the ability to correctly identify tuned responses, and these parameters made the tuned response function approximate a monotonic function. Such parameters were rarely found in the fMRI responses from areas showing evidence of tuning. n indicates the amount of included simulated data points. Source data are provided as a Source Data file.”

We discuss these findings in the discussion section as follows, highlighting situations in which the response may be misclassified by our procedure, as follows (p. 29 – 30, lines 589 – 605):

“Furthermore, our model validation procedure (Supplementary Fig. 2) demonstrated that ground-truth monotonic responses are very rarely classified as tuned. This gives high confidence that visual field maps identified as showing tuned responses do indeed show tuned responses, and indeed their response function parameters in tuned response models should facilitate identification as tuned responses. However, ground-truth tuned responses can be misclassified as monotonic when their response function approximates a monotonic function: where the preferred timing is outside the presented timing range, the response function is very large, or the compressive exponent on event frequency is high. Therefore, for early visual field maps identified as showing monotonic responses, some borderline cases may actually have tuned responses that our procedure cannot reliably identify. Specifically, tuned response models fit here have high estimates of the compressive exponent and slightly larger response functions than later visual field maps showing tuned responses. This does not strongly affect our conclusions because such tuned response functions closely resemble monotonic response functions, there is a gradual transition from monotonic to tuned responses, and tuned responses may have monotonic components. Nevertheless, it is possible that tuned responses emerge slightly earlier in the visual hierarchy than our model comparison suggests.”

2. Comparing variance explained

In Figures 3 and 6, the authors show the variance explained by the two models across the range of brain areas tested. The results are quite clear but something is odd. Only voxels with $R^2 > 0.2$ for either model were included. I interpret this that at least one of the models was above threshold for the voxel to be included (logical-OR operation). It must be because otherwise the means of any condition could never be below this threshold.

The reviewer’s interpretation was correct. We agree that the wording was confusing, so we have changed the use of “either model” to “*the best fitting model*” here and throughout the manuscript.

However, this logical-OR data selection is susceptible to statistical artifacts because the data selection is not independent. Consider if condition A is on average lower than condition B: by selecting only the voxels that are above 0.2 in A or B you will artifactually inflate the mean of condition B. It amplifies the difference already there. As far as I can tell, it should not produce a difference where isn't one - so this is probably not a major concern - but it may mean that the distinction of the models isn't actually as clear as shown here. My suggestion is to redo this analysis but for -all- voxels with $R^2 > 0$ (as the authors already do in some of the later analyses in Fig 5). It doesn't really make sense to threshold the data here anyway if the aim is to compare the goodness-of-fit of the two models.

The motivation for including this threshold is that large parts of the early visual field map ROIs do not respond to the timing of the stimulus. For example in V1, only the representation of the central 1.5 degrees convincingly responds, while the rest does not. Nevertheless, because of measurement noise many of the non-response voxels in the peripheral visual field map do have model fits above 0 variance explained. Therefore, the mean is strongly drawn towards the variance explained in unresponsive voxels, which are a clearly distinct part of the visual field map. We could address this by using a voxel selection based on pRF position, but each visual field map has a different relationship between model fits and eccentricity: unresponsive voxels in the peripheral visual field map are mostly found early in the hierarchy. We used a variance explained threshold so the same analysis could be used for all visual field maps.

In these early visual field maps, the median value of variance explained (as suggested in the reviewer's next point) is the value for a voxel that doesn't convincingly respond at all, so really doesn't allow any meaningful comparison. As a result, the combination of not thresholding and using a median creates a big problem in the case of visual field maps where only a small part responds in any model. We find using the median a better-motivated choice, but these choices don't make much difference to the results, as the reviewer expects.

When we look at the differences between models that we see with the threshold at 0.2 and at 0 (below) we see that the mean variance explained decreases with a threshold at 0. While some differences cease to be significant, others become significant. Regardless, the results in both cases show a transition from monotonic to tuned models as we go up the hierarchy. Therefore, the choice of the threshold makes little difference to the overall pattern of results and has a clear motivation (excluding non-responsive parts of early visual field maps).

To illustrate this we added supplementary figure 7:

“Supplementary Fig. 7: Model comparisons using means per hemisphere show similar results regardless of the variance explained threshold used for both visual field maps and timing maps. The threshold for voxel selection was a variance explained of either 0 (left column), 0.2 (middle column), or 0.4 (right column) for the best fitting model in the data on which the model was fit. All thresholds result in a gradual transition from monotonic to tuned responses, starting in similar areas. In all cases, the mean value of cross-validated variance explained was taken after thresholding in each measured visual field map example. At a threshold of 0, these means were not normally distributed, so points show the median across measured visual field map examples, error bars show 95% confidence intervals (from 1000 bootstrap iterations) and two-sided Wilcoxon signed rank tests were used. At other thresholds, the means were normally distributed, points show the mean across measured visual field map examples, error bars show the standard error of the mean, and two-sided paired *t*-tests were used. * $p < 0.05$ ** $p < 0.01$, and *** $p < 0.001$, FDR corrected for multiple comparisons. *n* indicates the amount of hemispheres included in the comparison in each map. Source data are provided as a Source Data file.”

Using a threshold at 0 also introduces a problem for interpreting another effect that we see here. As we go up the early visual field map hierarchy, the fits of the monotonic model increase (with either threshold). But with the threshold at 0, this can be explained simply by the inclusion of more non-responsive voxels in the earliest visual field maps with the smallest receptive fields and the most spatially localised responses. The threshold at 0.2 removes these non-responsive voxels, so supports our interpretation that there is a concentration of timing-modulated responses up the early visual hierarchy before timing-tuned responses emerge.

So overall, we feel that the choice of threshold makes little difference, but is clearly motivated and allows us to (1) use medians and (2) show how monotonic responses increase through the early visual hierarchy.

Notably, Reviewer 2 asks whether the threshold at 0.2 might be too low to focus on clearly responsive voxels. Our validation procedure (discussed in response to the reviewer’s first comment) also provides a good motivation for using a threshold at 0.2: this shows that our ability to distinguish between monotonic and tuned responses drops sharply below 0.2 variance explained, but changes little with variance explained for simulated data with variance explained above 0.2.

While we find that removing thresholding entirely causes other problems, we do agree with the reviewer that this is likely to exaggerate real differences. Fundamentally, we were selecting the voxels based on the same variance explained values that we then test, which is somewhat circular even if it doesn’t favour one model over the other. So we have come up with a better way of thresholding that we don’t believe suffers from these problems. In this new method we select voxels based on the models’ variance explained in the data on which the models are originally fit. In the statistical test underlying the model comparison, we then use the models’ variance explained in cross-validation, on the complementary half of the data. Therefore, we select the voxels based on fits in data that is independent from the values used in the test, avoiding circularity.

We have explained our new thresholding approach and the motivation for using thresholds at all in the results section, just after the model validation results, as follows (p. 9 - 10, lines 182 – 192):

“Our subsequent analyses therefore excluded voxels where the variance explained was 0.2 or less for both models in the data on which the models were fit (to avoid selecting voxels using the cross-validated fit values we compared). This excludes many voxels in each visual field map (often far from the retinotopic location of the stimulus) where responses do not systematically vary with timing, while including voxels where responses were convincingly modulated by event timing [...]

We then compared the fits of these models on cross-validated data, independent from that on which models were fit.”

We explain this again in the methods section, as follows (p. 43, lines 934 – 939):

“We excluded from model comparison voxels for which the variance explained of both models was 0.2 or less in the data on which the models were fit. Then, the cross-validated model fits (i.e. fits in data independent from that used for model fitting) were statistically compared to each other in each visual field map and timing map ROI. For each hemisphere, we then took the median model variance explained across the selected voxels in each ROI, separately for the two cross-validation splits.”

On a smaller but related point, it seems the authors take the arithmetic mean of R^2 across a ROI for each participant. R^2 distributions are distinctly non-normal so it makes more sense to use the median or another robust estimator of central tendency for that (but this probably makes no practical qualitative difference).

We agree that these distributions of variance explained are not normal, so indeed using a median would be preferable. However, as explained in our previous response, combining a median with a threshold at 0 creates a problem: the median voxel is a non-responsive voxel. Again, this fundamentally happens because our ROI definition is a visual field map, much of which does not respond to event timing, and the response to event timing is what we want to investigate.

For the model comparisons (i.e. does the monotonic or tuned model fit better?) we have now used the median value of cross-validated variance explained among the selection of voxels where at least one model has a variance explained above 0.2 in the other half of the data (below, left panels). This produces a very similar result to using the means (below, right panels), though note here that the error bars are larger because they are confidence intervals for the medians rather than standard errors (across participants) for the means. The exact same visual field maps and timing maps show significant differences in model fits, though now in non-parametric (Wilcoxon) tests because the participant medians were not normally distributed while the participant means were. We have now replaced the figures in the manuscript with the median-based version.

Figure: Model comparisons using medians (left) and means (right) per hemisphere show similar results with a variance explained threshold above 0 for both visual field maps and timing maps with a variance explained threshold of 0.2 for the best fitting model in the data on which the model was fit. Both summary statistics result in a gradual transition from monotonic to tuned responses, starting in similar areas. The medians per map were not normally distributed, so points show the median across measured visual field map examples, error bars show 95% confidence intervals (from 1000 bootstrap iterations) and two-sided Wilcoxon signed rank tests were used. The means were normally distributed, so points show the mean across measured visual field map examples, error bars show the standard error of the mean, and two-sided paired t-tests were used. * $p < 0.05$ ** $p < 0.01$, and *** $p < 0.001$, FDR corrected for multiple comparisons. n indicates the amount of hemispheres included in the comparison in each map.

In the manuscript's supplementary figures, we present the results from means (shown in our last response) and medians (shown below) separately, so we can show that neither the median/mean or threshold makes much difference to our overall result. But notably, using a median with a variance explained threshold at zero doesn't work well because the median voxel is non-responsive. Conversely, using a higher threshold of 0.4 causes problems because little data survives that threshold, so the error bars become very large although the overall pattern of results remains.

“Supplementary Fig. 6: Model comparisons using medians per hemisphere show similar results with a variance explained threshold above 0 for both visual field maps and timing maps. The threshold for voxel selection was a variance explained of either 0 (left column), 0.2 (middle column), or 0.4 (right column) for the best fitting model in the data on which the model was fit. A threshold of 0.2 and 0.4 result in a gradual transition from monotonic to tuned responses, starting in similar areas. The threshold of 0 deviates from this pattern because there are plenty of voxels within the maps that don't convincingly respond at all to any model. In all cases, the median value of cross-validated variance explained was taken after thresholding in each measured visual field map example. Points show the median across measured visual field map examples, error bars show 95% confidence intervals (from 1000 bootstrap iterations) and two-sided Wilcoxon signed rank tests were used. * $p < 0.05$ ** $p < 0.01$, and *** $p < 0.001$, FDR corrected for multiple comparisons. n indicates the amount of hemispheres included in the comparison in each map. Source data are provided as a Source Data file.”

We also compare different eccentricity ranges. Here things work differently because we are also interested in the variance explained in a particular eccentricity range even if it is zero: that eccentricity range does not respond, which is important information. But this causes a problem when looking at the medians in many eccentricity ranges, because these are simply zero (see below, left panels). Again, this comes from the difficult combination of unthresholded data, much of the visual field map being unresponsive to timing, and taking the median. This gives the misleading impression that the variance explained of the tuned response model is just zero in both eccentricity ranges and there is no difference between eccentricity ranges (when there often is, see below right panels for the same result using the mean variance explained in each eccentricity range). We should point out here that even if we do use the medians within each eccentricity range in each participant, we still see our important result that there is an effect of eccentricity on the difference between model fits only in visual field maps where the monotonic model fits better, but we see the basis of this result as misleading.

As a result of this misleading effect of using medians in a comparison where it is important not to use a threshold, here we still use the mean value of variance explained in each eccentricity range in each participant, as this contains important information about how each eccentricity range responds that is lost by using the median. We tried other measures than the mean and median (75th and 90th percentiles), but these caused other problems: we found that we would have to use some really convoluted measures to avoid a problem that the mean simply doesn't have. So while we agree that a mean of a strongly skewed distribution is not ideal, we couldn't find a better option. We then plot the medians and 95% confidence intervals of these means across participants, and compare them using Wilcoxon tests. So the summary of each map's variance explained is not the best choice from a theoretical viewpoint, but the subsequent stats on this summary value are treated non-parametrically.

Figure: Median (top) and mean (bottom) cross-validated variance explained in the two eccentricity ranges. These analyses are thresholded at a variance explained of 0 for the best fitting model in the data on which the model was fit. Presented are the variance explained by the monotonic (left) and tuned response model (middle). Also differences between these model fits (right).

3. Clarity of results

The presentation of some of the results could be clearer. I understand why the authors want to show everything they show in Figure 1 but this diagram is not going to be easy for a reader to digest, especially someone who isn't versed in pRF-type analyses. For one thing, it would help adding titles over the three columns labelling them as monotonic duration, frequency, and the tuned model. More generally, I wonder if less isn't more here. You could have a figure outlining the modelling more conceptually in the main text, and relegate this more detailed one to the supplementary information. Or perhaps it would help to split this figure. In any case, it would probably help running this by some non-expert readers to aid understanding.

We simplified the representation of the monotonic response model by moving the scaling factors of the duration and frequency components to the top. The revised figure (below, p. 8 – 9, lines 144 - 170) now has two main columns (with titles) that align with the two response models. Moreover, we added a simplified conceptual flow chart on the left to highlight the main steps. Computations that were applied to both models are now presented in the middle. We feel that this is far easier to understand, but essentially the modelling procedure is complex and we feel it is so important to the study that it needs to be described fully in the main text.

“Fig. 1 Monotonic and tuned response model fitting procedures. a Monotonic response model (left) and tuned response model (right). In each case, a compressive exponent parameter captures a non-linear scaling with increases in event duration (left) or frequency (middle left and middle right). The monotonic response predictions for event duration and frequency changes are combined as a weighted sum. **b** Together, these make a prediction of the per-event response amplitude for every event timing shown in the stimulus (dots). **c** This gives a prediction of the response amplitudes that would be seen for each stimulus condition for a specific candidate set of response model parameters. **d** The times of event offsets in each stimulus condition, which vary in frequency. **e** Combining the per-event response amplitudes predicted by the response models (in c) and the times of the event offsets (in d) gives neural response amplitude predictions for each condition, equal to the amount of color under the curve. **f** The hemodynamic response function. **g** The neural response amplitude predictions (in e) are convolved with a hemodynamic response function (in f) to get the predicted fMRI response time courses. Note here that these predictions are for both ascending and descending sweeps of duration and/or period, while neural response predictions (in c and e) are for ascending sweeps only. **h** The recorded fMRI response time course for an example voxel. The monotonic and tuned predictions were compared to the

recording from each voxel. The free parameters of both the monotonic and tuned response models were found that maximized the correlation (R^2) between predicted and measured fMRI response time courses. For the monotonic response model these parameters were the compressive exponents on duration and frequency components, and the weighting of these two components (ratio). This ratio is the ratio of the scaling factors for the two components in a general linear model. For the tuned response model, the free parameters were the compressive exponent on event frequency, preferred duration (x), preferred period (y), response function extent along its major (σ_{maj}) and minor (σ_{min}) axes, and major axis orientation (θ)."

In similar vein, there are a lot of details in the text that detract more than they help understanding. I'm not a big fan of tables but the statistical inferences reported across a large range of brain areas, complete with means, SDs, t-stats, and confidence intervals are a perfect example of information better shows as tables. Otherwise you have these massive blocks of statistical text, like in lines 168-183, 265-279, 285-297, 309-322, 360-369 (I hope I got them all). I'd turn these into tables and keep them in the Supplementary Info.

We agree that these blocks of text are horrible to read, they result from the journal's requirement to present the exact statistical outcomes of every test (not just stars on the figures). They are also horrible tables, but easier for the reader to skip over. We have now included these as tables (Table 1-3) in the main manuscript, but we will spare the reviewer by not including them here.

4. Eccentricity effects

The authors show interesting findings w.r.t. to how the models can explain data from peripheral pRFs far from the actual stimulus location. This raises an obvious question how this relates to cortical magnification and/or larger pRF sizes. In terms of functional significance, in peripheral vision spatial sensitivity is poorer while temporal sensitivity is enhanced. The authors already discuss a magno- vs parvocellular account but it isn't entirely clear what the interaction could be. Related to this, higher regions have larger receptive fields so more peripheral voxels will be stimulated by the central stimulus there while at is less the case in earlier regions. Could this explain the absence of the relationship with eccentricity in higher regions?

First, we should highlight for the reviewer that our stimulus was presented in the central visual field (entirely within 0.75 degrees eccentricity), so the relative magnocellular and parvocellular contributions in different parts of the visual field are not really relevant to our results. When we discuss the different magnocellular and parvocellular contributions to different visual field maps we are talking about the inputs at a single, central visual field position: even the central visual field has both magnocellular and parvocellular pathways.

However, the reviewer's other point here, that pRF sizes increase up the visual hierarchy is important and is indeed missing from our discussion. We now point out that the slower decrease in model fits as we go up the hierarchy is likely due to larger receptive field sizes, as follows (p. 30, lines 615 – 618):

“The decrease in model fits with eccentricity is clearest in early visual field maps with small spatial pRFs. It is more gradual in higher visual field maps with larger pRFs, which is expected as peripheral pRFs in these visual field maps also cover the central visual field where the stimulus was presented.”

We do not believe that this effect alone underlies an absence of an effect of eccentricity in higher visual field maps because there is an effect of eccentricity on model fits in all visual field maps. It is only the difference between tuned and monotonic model fits that is not related to eccentricity (in visual field maps where tuned models fit better than monotonic models).

On that note, I am a bit confused by the confidence intervals in Figure 4a. Is the dashed lines we see for V1 and V2 really the upper bound of the CI for the fit here? I assume the lower bound is effectively at zero? The bins have narrow error bars and the means are far away from this CI. This seems implausible but I'm probably missing something. In general, since the dashed lines are impossible to discern in most of the panels I would suggest using a filled region to denote the confidence interval (transparent or behind the line plot) rather than the dashed lines.

We have adjusted the fitting procedure for the sigmoid curves on the variance explained over eccentricity bins. We used to determine the 95% confidence interval of all parameters of the sigmoid curves separately. Currently, we determine the 95% confidence intervals of the fit of 10,000 bootstrapped sigmoid curves on the data.

We have also implemented the reviewer's suggestion to show the confidence intervals as a filled area rather than dashed lines. The new figure looks as follows (p. 20, lines 386 – 399) :

“Fig. 5 Progression of timing response model fits with preferred visual field eccentricity in a representative selection of visual field maps. a Early and lateral occipital visual field maps show a sharp decrease of model fits moving away from the retinotopic representation of the stimulus position. This decrease then becomes more gradual where tuned response model fits begin to improve. b The difference between the

response model fits (tuned - monotonic) also decreases with eccentricity in the early and lateral occipital visual field maps, but shows no consistent relationship with eccentricity after TO1. Markers show mean variance explained per eccentricity bin, error bars show the standard error of the mean. For all bins, $n \geq 50$ included voxels. Solid lines show the best fit to changes with eccentricity, shaded areas around these lines are their 95% confidence intervals. Note that the data for these plots are not thresholded to a variance explained above 0.2, but above 0 for the best fitting model in the data on which the models were fit. See also Supplementary Fig. 8. Source data are provided as a Source Data file.”

5. Missing ROIs

The authors seemingly analyse every visual and timing-related region under the sun. For that reason it seems odd that as far as I can tell there is no mention of V4 anywhere in the manuscript. Also, how were the ROIs in Figure 4 chosen because some of them are absent from this plot as well (such as LO1). This is not a big issue but it should probably be discussed somewhere.

In (previously) Figure 4 (now Figure 5) we do not show all visual field maps because the figure is already busy and has a lot of panels. We chose representative visual field maps from early vision (V1, V2), MT+ (TO1-2), an area between those (LO2), and parietal (IPS maps) and frontal (sPCS) visual field maps. Together these show quite different relationships between monotonic responses, tuned responses, the difference between them, and visual field position eccentricity. The figures title describes this as “*a representative selection of visual field maps*”. We now also state this in the text where the figure is first mentioned, as follows (p. 18, lines 358 – 359):

“(see Fig. 5 for a representative selection of visual field maps and Supplementary Fig. 8)”

We omitted hV4 from our selection of visual field maps because fMRI measurements in hV4 are strongly affected by an overlying large draining vein, the transverse sinus (Winawer et al 2010). This makes it difficult to take accurate functional images from hV4: the BOLD response measured here can pool responses across large areas of the visual field (contaminating our eccentricity comparisons) and even from other visual field maps (contaminating our model comparison). This omission does not really concern us because our fMRI volume did not consistently cover the (other) ventral visual field maps, so our results focus on early visual, lateral and dorsal visual field maps, which we would expect to be more involved in temporal processing than ventral stream areas. We now discuss these points as follows (p. 31 – 32, lines 643 – 651):

“Notably, we do not analyze responses in ventral stream visual field maps, including hV4. These are generally implicated in object processing, so we focused our data collection on the lateral, dorsal stream, and parieto-frontal visual field maps implicated in motion processing, multisensory integration and attention control. We also investigate early visual field maps that provide their inputs. We omit hV4 (considered either an early visual or ventral stream visual field map) because fMRI recordings here are strongly affected by a large overlying vein, the transverse sinus (Winawer et al., 2010). Responses measured here can

pool responses across large parts of the visual field (contaminating our eccentricity comparisons) and even from other visual field maps (contaminating our model comparison)."

6. Typo on page 12, line 232: captured -> capture

Thank you, we have now corrected this.

7. Data availability

I don't want to make the authors' lives more difficult but given NC is championing data transparency I believe a statement that data will be shared upon "reasonable request" is not good enough. What requests are "reasonable"? Who decides this? I entirely understand that there are legal and ethical limits to data sharing but this must be specified clearly. Quoting the journal's guidelines:

"...reasons for controlled access (eg., privacy, ethical/legal issues), conditions of access must be described precisely including contact details for access requests, timeframe for response to requests, restrictions imposed on data use via data use agreements. [...] Restrictions on controlled access datasets including restrictions on downstream data reuse or authorship requirements must be clearly described in manuscript and to editors at the time of submission."

We agree with the preference to share scientific data publicly. Unfortunately, our ethical review board does not allow sharing of medical imaging data to public repositories, even if anonymised. Their reasoning for this is that the structure of the brain is unique to the individual participant, in theory allowing the participant to be identified from any images showing the brain's structure, which may also contain medically sensitive findings. This is an interpretation of the EU's General Data Protection Regulation (GDPR) that is common in the EU for medical images, and widely applied here in the Netherlands for MRI data.

While we cannot share such data to public repositories, we can share them with specific researchers and collaborators (on request) provided they agree not to share them publicly. Sharing with researchers who contact us unexpectedly from outside the EU (where the GDPR does not apply) may require additional agreements: this would first need to be discussed with our ethics board and the legal interpretation of the GDPR is ongoing. The timeframe for this is not certain. This is why we specify 'reasonable' request: our ethics board decides what is reasonable. We should also mention that the raw data here are around 200 GB, which is generally easiest to share by mailing a hard disk. Such a request would be reasonable for us and our review board if it came from a member of the scientific community, but not necessarily coming from a member of the public that has no clear motivation to request the medical imaging data.

Because sharing our medical image data is a complex issue, instead we share all the numbers that go into our statistical tests (in the supplementary materials of our original submission). In the revision, we share the model parameters from which these are derived, for every voxel in our ROIs. Importantly, neither format contains any information about the structure of our participants' brains.

Because we fundamentally agree with the preference to share as much underlying data as possible, we have also included all the time series data from every voxel in our visual field map and timing map ROIs for all the timing mapping and visual field mapping data we analyse. Therefore, we now share all of our recorded fMRI responses as part of the publication, but in a somewhat unusual way that removes all structural information.

We have modified our data availability statement as follows to clarify these restrictions (p. 48, lines 1054 – 1058):

“Ethical constraints prevent us from sharing the medical imaging data sets (MRI scans) generated in the current study to public repositories. These raw data sets are available from the corresponding author upon reasonable request, depending on agreements not to share these data publicly. Source data for all model fitting and the model parameters underlying all statistical analyses are provided as supplementary materials of this paper.”

REVIEWER COMMENTS

Reviewer #1 (Remarks to the Author):

In this revision, Hendrikx and co-authors made major changes to the manuscript that have substantially improved the paper. Specifically, they have now introduced their theories and implications in the introduction, added analyses comparing model parameters across visual field maps, improved the clarity of the writing, moved the cumbersome statistics to tables, and better situated their results in the context of other studies in the discussion. I appreciate all the amount of work that went into this.

Nonetheless, there are still several outstanding issues that need to be addressed

Theory:

The contextualization of the paper in terms of prior models and findings has substantially improved the scholarly impact of the paper. Now that the models are better described and I can follow them, I believe the authors should better clarify the assumptions and terminology, which I believe is still confusing.

The authors compare the predictions of the monotonic and tuned models on events that last between 50ms-1000ms. This is a restricted regime of timing. Perhaps in this regime it is reasonable to assume monotonic or tuned responses in visual cortex. However, it is likely that neurons responses may deviate widely from these model predictions in temporal regimes that are much shorter or longer. E.g., do you expect a monotonic increase in V1 between events that last a micro and millisecond? Likewise, do you expect that responses in V1 will continue to rise for events that last minutes or hours? While the authors have a better survey of the literature, they should justify the limited temporal range that they used in the experiments, indicate what evidence is there that the monotonic and tuned models are thought to be applicable in this temporal regime, and indicate in the discussion the limitation of their models to explain neural responses to events with timing outside this range.

The authors use duration, frequency, and period to describe the nature of their stimuli and models. This is still a bit confusing for several reasons: (i) the monotonic model is described in terms of frequency and duration but the tuned model is described in terms of period and duration, (ii) the formulas of the models include frequency, period, and duration but $\text{period} = 1/\text{frequency}$, so they are dependent variables, and (iii) event duration is not completely independent from event period as $\text{duration} \leq \text{period}$. I suggest that the paper (i) use frequency & duration or period & duration when describing the models in the text, figures, and formulas (yet keep the information that $\text{frequency} = 1/\text{period}$ for clarity) and (ii) in the main text clarify the relation between duration and period.

Results:

I appreciate that the authors now included new analyses and a new figure with the model parameter results (new Figure 4). However, I am surprised that all the visual regions have very similar preferred durations (Fig 4e) of about .4 s. Is this due to the way the stimulus was presented? That is the stimuli are presented in a continuous way and are not randomized and this is the mid range of the stimuli timing parameters.

This result is also some contrary to the authors' predictions in the introduction:

These timing tuned responses are topographically mapped, such that the preferred durations and periods of neural populations gradually progresses across the cortical surface[19,31].

I wonder if the distributions of duration preference vary systematically across cortex or areas

Ventral Stream ROIs

The authors have been asked about the models' outcomes in ventral visual maps such as hV4; The authors had responded that they did not analyze data in hV4 because of the venous eclipse (Winawer 2010). This is not a compelling response for several reasons: (i) while the venous eclipse may obscure some of the lower visual field representation in hV4 it does not obscure the upper visual field representation (which has been measured successful since 1995). Furthermore, it is an intermittent artifact and most of the times you can measure a complete hemifield in hV4; In fact, there are thousands of papers that have published fMRI data from hV4. (ii) There are other visual field maps in the ventral stream that are medial to the venous eclipse and do not suffer from this issue (VO1/2; PHC1/2), and (iii) There is a large draining vein in the back of the brain along the posterior edge of the occipital lobe (Dural Sinus) traversing V1/V2m but that did not dissuade the authors from reporting data from these areas. For completeness, and to extend the present results in a significant way beyond the group's 2020 paper, it would be interesting to add the ventral stream ROIs and compare the model parameters to the dorsal and lateral stream ROIs.

Effects of eccentricity:

The authors report that there are differences in model variance explained for the monotonic model but not the tuned model across eccentricities (note they are only measuring the central 5 degrees). This data is presented in Figure 5 (see V1, V2, LO1). Given that the stimulus was small (...visual events comprised the presentation of a filled circle with a diameter of 0.4 degrees and was 0.75 degrees away from the fixation cross) and receptive fields of V1 and V2 are the smallest, I wonder if this is an effect of eccentricity or an effect of the stimulus not hitting the receptive fields of neurons more than 2 degrees away from fixation in early visual cortex. In other words, is this a qualitative difference between early and higher level regions, or is this dues to a floor effect as the tuned model explains less than 5% of the variance of V1 and V2 voxels? For LO1 there seems to be a similar trend across eccentricities for both the monotonic and tuned models.

Minor comments:

- 1) I wanted to watch Supplementary Movie 1 to see the experiment, but the movie is not accessible.
- 2) I agree with the other reviewers that the figures are somewhat overbearing and contain too many panels. To aid the readers, I suggest to move all the yellow curves showing differences between monotonic and tuned conditions from Figures 3b, 5b, 7b to the supplementary figures as the differences are already evident in the main plots by comparing the red (tuned) and blue (monotonic) data.

Reviewer #2 (Remarks to the Author):

The authors sufficiently addressed my concerns and I acknowledge that the manuscript is greatly improved. I feel this manuscript is now ready to publish.

Reviewer #3 (Remarks to the Author):

The authors have addressed all my previous points conclusively and I commend them on the hard work. The revised manuscript does a great job communicating this fairly complex piece of research. The validation work is a great addition, also helping to deal with the question about the appropriate threshold.

Lastly, I thank the authors for taking seriously my comment about data availability. I understand (and can relate to) the issues with privacy and data sovereignty - so an explanation for why brain images cannot be shared publicly and a clear detail of the conditions for sharing data is entirely sufficient in my book. So I commend the authors for going the extra mile here in sharing the processed data that should allow interested parties to further interrogate the results without sacrificing data protection requirements.

Sam Schwarzkopf
UoA

Reviewer #1 (Remarks to the Author):

In this revision, Hendriks and co-authors made major changes to the manuscript that have substantially improved the paper. Specifically, they have now introduced their theories and implications in the introduction, added analyses comparing model parameters across visual field maps, improved the clarity of the writing, moved the cumbersome statistics to tables, and better situated their results in the context of other studies in the discussion. I appreciate all the amount of work that went into this.

We are glad that the Reviewer was happy with the changes made in the previous revision.

Nonetheless, there are still several outstanding issues that need to be addressed

Theory:

The contextualization of the paper in terms of prior models and findings has substantially improved the scholarly impact of the paper. Now that the models are better described and I can follow them, I believe the authors should better clarify the assumptions and terminology, which I believe is still confusing.

We agree that some further improvements can be made here. We will reply to specific points as they are raised below.

The authors compare the predictions of the monotonic and tuned models on events that last between 50ms-1000ms. This is a restricted regime of timing. Perhaps in this regime is it reasonable to assume monotonic or tuned responses in visual cortex. However, it is likely that neurons responses may deviate widely from these model predictions in temporal regimes that are much shorter or longer. E.g., do you expect a monotonic increase in V1 between events that last a micro and millisecond? Likewise, do you expect that responses in V1 will continue to rise for events that last minutes or hours? While the authors have a better survey of the literature, they should justify the limited temporal range that they used in the experiments, indicate what evidence is there that the monotonic and tuned models are thought to be applicable in this temporal regime, and indicate in the discussion the limitation of their models to explain neural responses to events with timing outside this range.

We agree with the Reviewer that we have looked at a limited range here, as it is not feasible to look at timings from microseconds to hours in a single experiment. Therefore, we focus on responses to the timing of sub-second events, from 50 ms to 1000 ms. The reviewer asks some questions about what might happen outside this range. As we did not record responses outside of this range, we can only speculate how responses would change. Regarding monotonic responses to event frequency or period, we would expect response amplitudes to increase with event frequency over almost any range, such that the summed responses to events with a period of 50 minutes would be greater than responses to (less frequent) events with a period of 60 minutes for example. Similarly, we would expect summed responses to events with a period of 40 ms to be greater than responses to events with a period of 50 ms. Simply, more frequent events should lead to more frequent sensory responses, with the sum over time therefore increasing. However, there are likely to be limits to this, particularly at very high frequencies where humans cannot distinguish individual events. This is one reason why we stop at 50 ms periods. With monotonic responses to event duration, we would expect similar increases of response amplitude with duration, but also with limits for very short events (where humans cannot accurately perceive event duration) and very long events (where at a certain point the time the event began is hard to remember).

We chose this sub-second range for two main reasons. First, humans can accurately perceive event timing in this range, and it is therefore widely used in perceptual studies of event timing (for example by Heron and colleagues' 2012 study of duration adaptation effects). Second, there is indeed evidence that monotonic and tuned responses apply over similar ranges, in previous fMRI studies (monotonic: Zhou and colleagues 2018, Stigliani and colleagues 2017 & 2019. Tuned: Harvey and colleagues 2020, Protopapa and colleagues 2019) as well as electrocorticography of monotonic responses to visual event timing (monotonic: Zhou and colleagues 2019) and macaque single-cell neurophysiology of responses to motor event timing (monotonic: Merchant and colleagues 2011. Tuned: Merchant and colleagues 2013).

To clarify this, we now introduce our research question at the end of the Introduction by saying (new text underlined):

“Here we ask whether and how monotonic and tuned neural responses to sub-second visual event timing are related throughout the brain’s hierarchy of both timing maps and visual field maps.”

And in the first sentence of the Results by saying:

“We first asked whether and how monotonic and tuned neural responses to sub-second visual event timing are related throughout the hierarchy of both timing maps and visual field maps.”

We also now specify ‘sub-second’ visual event timing in several other places for clarity.

In the second paragraph of the Results, we now give our reasons for focussing on this range as follows (all new text):

“We focus on events with durations and periods from 50 ms to 1000 ms for several reasons. First, it is feasible to sample this limited range in a single experiment. Second, previous studies using fMRI (Hayashi et al., 2015; Hayashi et al. 2018; Hayashi et al. 2020; Harvey et al., 2020; Protopapa et al., 2019; Stigliani et al., 2017; Zhou et al., 2018), animal neurophysiology (Merchant et al., 2011; Merchant et al., 2013) and psychophysics (Bruno et al., 2010; Bruno et al. 2015; Heron et al., 2012; Maarseveen et al., 2017; Morrone et al., 2005) have used similar ranges, allowing us to relate our findings to previous literature. Finally, these studies shown both monotonic (Merchant et al., 2011; Stigliani et al., 2017; Zhou et al., 2018) and tuned (Harvey et al., 2020; Merchant et al. 2013; Protopapa et al., 2019) neural responses within this range.”

Our models also make quantitative predictions of how response amplitudes increase with event duration and frequency using a single exponent parameter in each case. We have no reason to believe these quantitative predictions would be accurate far from the tested range, and it is likely that a more complex model would be needed to capture responses at a very large range of time scales. Furthermore, there is no evidence in the literature of monotonic or tuned responses to visual event timing for durations and periods below 50 ms (which is shown by our 2020 study) or above 3 seconds (which Protopapa and colleagues show) in the human brain, and also no evidence of duration adaptation aftereffects outside of this range. On the other hand, reports of visual event timing tuned responses are new, with the first demonstration from neural measurements dating only from Protopapa’s 2019 study, so they may exist somewhere.

At the end of the discussion, we now speculate on potential responses to broader timing ranges and explain that more research is needed here, as follows:

“Our experiments focus on events with durations and periods from 50 ms to 1000 ms. It remains unclear how the brain responds to visual event timings outside that range. For monotonic responses to event frequency, we would expect that neural response amplitudes summed over time would also increase with event frequency outside this range: each event should produce a neural response, the sum of which should increase with the number of events. Similarly for monotonic responses to event duration, we would expect increases to continue as duration continues to increase. However, there are likely to be limits in both cases. At very high frequencies, humans cannot distinguish visual events, leading to flicker fusion, and we similarly cannot accurately perceive the durations of events that are faster than neural responses. At very low frequencies or for very long events (for example with periods or durations of minutes), the event’s temporal structure is less perceptible, and we must remember how long ago the event began. Our monotonic response models make quantitative predictions outside the tested range, but these predictions are not tested and a single compressive exponent parameter seems unlikely to be able to predict responses over a very large range of time scales. For tuned responses, there is no evidence in the literature of visual event timing tuned responses with preferred durations or periods below 50 ms (Harvey et al., 2020) or above 3000 ms (Protopapa et al., 2019) or indeed adaptation effects operating outside these ranges (Heron et al., 2012). However, descriptions of visual event timing-tuned neural populations are relatively recent, and further research may reveal such responses.”

Harvey, B. M., Dumoulin, S. O., Fracasso, A. & Paul, J. M. A network of topographic maps in human association cortex hierarchically transforms visual timing-selective responses. *Curr. Biol.* **30**, 1424-1434.e6 (2020).

Heron, J. et al. Duration channels mediate human time perception. *Proc. R. Soc. B Biol. Sci.* **279**, 690–698 (2012).

Merchant, H., Pérez, O., Zarco, W. & Gámez, J. Interval tuning in the primate medial premotor cortex as a general timing mechanism. *J. Neurosci.* **33**, 9082–9096 (2013).

Merchant, H., Zarco, W., Pérez, O., Prado, L. & Bartolo, R. Measuring time with different neural chronometers during a synchronization-continuation task. *Proc. Natl. Acad. Sci. U. S. A.* **108**, 19784–19789 (2011).

Protopapa, F. et al. Chronotopic maps in human supplementary motor area. *PLoS Biol.* **17**, 1–34 (2019).

Stigliani, A., Jeska, B. & Grill-Spector, K. Encoding model of temporal processing in human visual cortex. *Proc. Natl. Acad. Sci. U. S. A.* **114**, E11047–E11056 (2017).

Stigliani, A., Jeska, B., & Grill-Spector, K. Differential sustained and transient temporal processing across visual streams. *PLoS Comput. Biol.* **15**, e1007011 (2019).

Zhou, J., Benson, N. C., Kay, K. N. & Winawer, J. Compressive temporal summation in human visual cortex. *J. Neurosci.* **38**, 691–709 (2018).

Zhou, J., Benson, N. C., Kay, K., & Winawer, J. Predicting neuronal dynamics with a delayed gain control model. *PLoS Comput. Biol.* **15**, e1007484 (2019).

The authors use duration, frequency, and period to describe the nature of their stimuli and models. This is still a bit confusing for several reasons: (i) the monotonic model is described in terms of frequency and duration but the tuned model is described in terms of period and duration, (ii) the formulas of the models include frequency, period, and duration but $\text{period} = 1/\text{frequency}$, so they are dependent variables, and (iii) event duration is not completely independent from event period as $\text{duration} \leq \text{period}$. I suggest that the paper (i) use frequency & duration or period & duration when describing the models in the text, figures, and formulas (yet keep the information that $\text{frequency} = 1/\text{period}$ for clarity) and (ii) in the main text clarify the relation between duration and period.

We agree this is a complex point, and that it is important for readers to understand how frequency in period are related. However, we find serious problems with explaining both response models in terms of period only or frequency only that we feel further complicate matters and are straightforwardly solved by simply explaining clearly that $\text{frequency} = 1/\text{period}$ from the start. The problem with explaining the tuned response model in terms of frequency only is that the Gaussian response function is not a Gaussian function of frequency, it is a Gaussian function of period. While it would be possible to change all the period terms to $1/\text{frequency}$, this is mostly simply avoided by simply using the term period and explaining from the start that this $\text{period} = 1/\text{frequency}$ and highlighting this point when helpful. On the other hand, explaining the monotonic response model in terms of period only is difficult because summed response amplitudes increase with duration and frequency, but decrease with period. When we have tried to explain the monotonic model in terms of period, this breaks the intuitive and straightforward implication that responses get larger as frequency increases and also the implication that the events are repetitive. We then have to always repeat that $\text{period} = 1/\text{frequency}$ to make these points clear, while currently they are very clear. For example, the sentence “*We hypothesized that visual event timing-tuned responses are likely to be derived from the inherent modulation of visual responses by event timing: monotonic increases in early visual responses with event duration and frequency*” has to become “*We hypothesized that visual event timing-tuned responses are likely to be derived from the inherent modulation of visual responses by event timing: monotonic increases in early visual responses with event duration and decreases with event period (the time between event onsets, i.e. $1/\text{frequency}$)*”. We find this confuses matters, and is redundant if the reader already understands that $\text{frequency} = 1/\text{period}$. We also feel that ‘frequency’ is widely used in general English to mean what we mean here, while ‘period’ has other meanings than a measure of frequency, so we prefer to use frequency when we can and period only when we must. Furthermore, the study of Stigliani and colleagues, which first introduces a monotonic response model like ours, also describes events in terms of frequency, and we prefer to be consistent with the previous literature where possible. We find it much clearer to occasionally repeat that $\text{period} = 1/\text{frequency}$ (or vice versa) when needed.

Nevertheless, we agree with the reviewer that highlighting this relationship between period and frequency in more places will help readers. When we introduce the equation for the monotonic response model (Equation 1), we already state that:

“*This model has two components that scale independently with event duration and frequency ($1/\text{period}$) (Stigliani et al., 2017).*”

When introducing the tuned response model, we have now clarified the parameters we use and why:

“*This response model describes the timing of each event in terms of duration and period (rather than frequency, i.e. $1/\text{period}$) because both of these are expressed in seconds and the response function is a Gaussian function of these parameters.*” In the equation for this response model, we use frequency for the compressive exponent

parameter for consistency with equation 1 for the monotonic model, which uses the same term, and we feel it would confuse readers to use both $(\text{Frequency}^{\text{expFreq}})/\text{Frequency}$ and $\text{Period}/(\text{Period}^{\text{expFreq}})$ when they are equivalent.

Similarly in the introduction, we already clearly explain that $\text{period}=1/\text{frequency}$ (or vice versa) when introducing monotonic and tuned responses, as follows:

“Early visual cortical responses increase monotonically but sub-linearly with event duration and frequency (and therefore decrease with event period, i.e. $1/\text{frequency}$).”

“Beyond early visual areas, an extensive network throughout the human association cortices shows visual timing-tuned responses, with maximum response amplitudes at specific preferred durations (the time from event onset to offset) and periods (the time between repeating event onsets; i.e., $1/\text{frequency}$).”

The reviewer also asks us to clarify the relationship between duration and period, that $\text{duration}\leq\text{period}$. We have now clarified this as follows:

“Both event duration and period ranged from 50 ms to 1000 ms in 50 ms steps, and any event’s duration was always less than its period so that the event ended before the next began and there was never more than one event happening.”

Results:

I appreciate that the authors now included new analyses and a new figure with the model parameter results (new Figure 4). However, I am surprised that all the visual regions have very similar preferred durations (Fig 4e) of about .4 s. Is this due to the way the stimulus was presented? That is the stimuli are presented in a continuous way and are not randomized and this is the mid range of the stimuli timing parameters.

To clarify this point, Figure 4a-i show the median response function parameter in each map. There is plenty of variation of preferred durations and periods WITHIN each visual field map, as we have previously shown within each timing map (Harvey et al, 2020). We prefer not to repeat these published results here as they are not within the scope of our research question (i.e. the relationship between monotonic and tuned responses). Figure 4 looks at differences BETWEEN visual field maps so uses a summary statistic (the median) to give a single value for each visual field map. The median duration and period are indeed similar between visual field maps, as we have previously shown between timing maps. But as a median, it ignores the range of response function parameters within each visual field map.

An analysis of changes in response function parameters within a visual field map is less well motivated than within a timing map. Each visual field map is a sub-sample of one or more timing maps: while a timing map contains a complete set of timing response model parameters found in a local area of the cortex, a visual field map is a somewhat random sample from that set: the relationship between visual field map borders and timing map borders varies between participants (as we have previously shown). We have now clarified why we don’t look at changes in response function parameters within visual field maps as follow (in the first paragraph of the Results section “Changes in response model parameters between visual field maps”):

“Here we do not analyze how response model parameters change within visual field maps as the distribution of timing response function preferences is better characterized with a timing map, which has a complete set of response properties in a particular region of the brain. We have done this elsewhere (Harvey et al, 2020). The set of voxels within a visual field map represents a variable sample from the set within a complete timing map.”

The reviewer also asks whether the similar values in different maps is because this is the middle of the presented timing range. In our previous study, which analyses the changes within timing maps, we propose that the response preferences seem to be drawn to the middle of the presented range, saying in the abstract that:

“Progressing from posterior to anterior maps, responses to multiple events were increasingly integrated, response selectivity narrowed, and responses focused increasingly on the middle of the presented timing range.” Again, we prefer not to repeat ourselves here.

We have no reason to think that this apparent attraction of timing preferences towards the middle of the presented range is related to the order of presentation, particularly because the central tendency of timing perception (attraction of perceived timing towards the middle of the presented range) is found when timings are randomly ordered. But we haven’t tested this experimentally because it is not feasible to do our experiment with a random order of presentations: sequential presentations of events that vary gradually in timing (or other parameters) produces a signal from which response functions can be estimated more reliably for a fixed amount

of scanning, and we are already scanning each participant for 4 sessions of around 70 minutes. We prefer not to speculate about the outcome of an experiment we did not perform.

This result is also some contrary to the authors' predictions in the introduction:

These timing tuned responses are topographically mapped, such that the preferred durations and periods of neural populations gradually progresses across the cortical surface[19,31].

I wonder if the distributions of duration preference vary systematically across cortex or areas

Again, we have previously shown systematic changes in timing preferences across the cortical surface, forming topographic maps (Harvey et al., 2020). Others have found similar results (Protopapa et al, 2019). But these results have already been published, are cited here (as quoted) and are not related to the research questions of the current study. These systematic changes are not contrary to the findings of the current study, they are changes WITHIN brain areas while the current study looks at changes BETWEEN brain areas.

Ventral Stream ROIs

The authors have been asked about the models' outcomes in ventral visual maps such as hV4; The authors had responded that they did not analyze data in hV4 because of the venous eclipse (Winawer 2010). This is not a compelling response for several reasons: (i) while the venous eclipse may obscure some of the lower visual field representation in hV4 it does not obscure the upper visual field representation (which has been measured successful since 1995). Furthermore, it is an intermittent artifact and most of the times you can measure a complete hemifield in hV4; In fact, there are thousands of papers that have published fMRI data from hV4. (ii) There are other visual field maps in the ventral stream that are medial to the venous eclipse and do not suffer from this issue (VO1/2; PHC1/2), and (iii) There is a large draining vein in the back of the brain along the posterior edge of the occipital lobe (Dural Sinus) traversing V1/V2m but that did not dissuade the authors from reporting data from these areas. For completeness, and to extend the present results in a significant way beyond the group's 2020 paper, it would be interesting to add the ventral stream ROIs and compare the model parameters to the dorsal and lateral stream ROIs.

We have now included responses from hV4, VO1, VO2 and PHC visual field maps in model comparisons. While mapping hV4 can be complex, we could clearly identify hV4 and VO1 in all hemispheres, VO2 in 14 of 16 hemispheres, and PHC in 5 hemispheres (where we could not separate PHC1 and PHC2). We are happy to include these for completeness.

We found that hV4 responded very much like V3, LO1 and LO2, showing convincingly monotonic responses. We have now included hV4 in all of our analyses. However, both monotonic and tuned response models fit responses in VO1, VO2 and PHC poorly, worse than early visual or dorsal stream visual field maps. So it seems that the responses in these ventral visual field maps are not convincingly modulated by visual event timing, at least for our events (consisting of a dot appearing and disappearing, rather than objects). Therefore, we show the poor fits of both responses models in Figure 3. But we do not then analyse the responses of VO1, VO2 and PHC any further, because the underlying response models fit so poorly that we can't conclusively compare response models (which also show huge error bars on variance explained and no significant differences between response models) or reliably estimate response model parameters.

In the revised manuscript, we now show hV4 in all figures showing the other visual field maps. The results seen in hV4 are much like V3, LO1 or LO2, so there is nowhere that the results from hV4 need detailed discussion on their own. We refer to hV4 in many places in the text where the other visual field maps are mentioned, for example (new text underlined):

"Post-hoc multiple comparisons demonstrated that early visual and lateral visual field maps had significantly better fits for the monotonic response model (V1, V2, V3, hV4, LO1, LO2, and TO1; Fig. 3, Table 1)."

We show the response models fits in hV4, VO1, VO2 and PHC in Figure 3 as follows:

Fig. 3 Comparison of fits of monotonic and tuned response models in each visual field map.

We describe the results for the ventral stream visual field maps seen in Figure 3 as follows:

“Ventral stream visual field maps (VO1, VO2 and PHC), and visual field maps that lay in between the early visual (monotonic) and parietal/frontal (tuned) maps (TO2, V3AB and IPS0) showed no significant difference between the fit of the monotonic and tuned response model (Fig. 3, Table 1).”

...

“In the ventral stream visual field maps (VO1, VO2 and PHC) the maximum variance explained by either response model was lower than in other visual field maps, and few hemispheres showed any voxels reaching the variance explained threshold of 0.2. As such, the responses of these visual field maps are not convincingly affected by visual event timing. We have therefore excluded them from further analysis.”

We raise this in the Discussion as follows. Please note that we have removed hV4 from this discussion, as it responds much like the (other) early visual field maps:

“Notably, we do not analyze responses in ventral stream visual field maps (VO1, VO2 and PHC) in detail. These are generally implicated in object processing, so we focused our data collection on the lateral, dorsal stream, and parieto-frontal visual field maps implicated in motion processing, multisensory integration and attention control.

We also investigate early visual field maps that provide their inputs. In the ventral visual field maps, the maximum variance explained by either response model was lower than in other visual field maps, so the responses here were not convincingly modulated by visual event timing, and as a result we could not distinguish between response models or quantify response model parameters here. However, visual field mapping in the ventral visual field maps often uses stimuli including objects (faces, buildings and other objects) to maximize responses (Benson et al., 2018). Analogously, responses in the ventral visual field maps may be more strongly modulated by event timing if the events included objects rather than dots alone.”

Effects of eccentricity:

The authors report that there are differences in model variance explained for the monotonic model but not the tuned model across eccentricities (note they are only measuring the central 5 degrees). This data is presented in Figure 5 (see V1, V2, LO1). Given that the stimulus was small (...visual events comprised the presentation of a filled circle with a diameter of 0.4 degrees and was 0.75 degrees away from the fixation cross) and receptive fields of V1 and V2 are the smallest, I wonder if this is an effect of eccentricity or an effect of the stimulus not hitting the receptive fields of neurons more than 2 degrees away from fixation in early visual cortex. In other words, is this a qualitative difference between early and higher level regions, or is this due to a floor effect as the tuned model explains less than 5% of the variance of V1 and V2 voxels? For LO1 there seems to be a similar trend across eccentricities for both the monotonic and tuned models.

We agree that we can't distinguish effects of eccentricity from effects of overlap between the voxels' population receptive fields and the fixated stimulus area. This is because the distribution of dot positions was centred on the fixated location (within 0.75 degrees of fixation, not placed 0.75 degrees AWAY from fixation as this comment suggests). We interpret this effect of eccentricity as an effect of overlap of the receptive fields and the stimulus area, although we quantify this overlap in terms of eccentricity. We make clear that this decrease with eccentricity is likely to depend on receptive field size, saying (in the results): “*This decrease was steep and sudden in early visual and lateral occipital visual field maps, which have smaller spatial receptive fields and where responses to event timing are best captured by monotonic response models. A decrease in model fits with eccentricity was also apparent in higher visual field maps, where the tuned response model begins to fit better, but this decrease was more gradual.*” and (in the discussion) “*The decrease in model fits with eccentricity is clearest in early visual field maps with small spatial pRFs. It is more gradual in higher visual field maps with larger pRFs, which is expected as peripheral pRFs in these visual field maps also cover the central visual field where the stimulus was presented.*” We are also clear in the discussion that we are only measuring a small part of the visual field and tuned responses may show a clearer dependence on stimulus position if we mapped a much larger area of the visual field, saying “*Alternatively, timing-tuned response components might depend on retinotopic location if we mapped a larger area of the visual field: in studies quantifying the duration aftereffect's spatial spread, at 5° from the adapter it is around 50% of the strength at the adapted location (Fulcher et al., 2016), and our visual field mapping stimulus covered only 5.6° from the stimulus area.*” We feel we are clear about these limitations of our experiments.

The reviewer asks another question that we are not sure we understand, asking “is this due to a floor effect as the tuned model explains less than 5% of the variance of V1 and V2 voxels? For LO1 there seems to be a similar trend across eccentricities for both the monotonic and tuned models.” If we understand this question correctly, we should first clarify that our results do not show that there is no effect of eccentricity on the fit of tuned response models (which LO1 and LO2 show clearly), but rather that there is no effect of eccentricity on the additional response captured by tuned response models. Then, when we look at Figure 5a we generally see that monotonic and tuned model fits plotted against eccentricity converge when the monotonic model fit (blue) is higher, but are roughly parallel when the tuned (red) model fit is higher (i.e. there is no effect of eccentricity on the difference in model fits where the tuned model fits better). For V1 and V2, this could perhaps arise because both lines are converging on zero (i.e. a floor effect). However, the same convergence and differences between eccentricity ranges are also seen in LO1, LO2, and V3A/B, which show no such floor effect but are better fit by monotonic response models and show a clear effect of eccentricity on the fits of both models and the difference between these fits. So, while this could be interpreted in different ways looking at V1 and V2 only, taking all the visual field maps into account we conclude that there is an effect of eccentricity on differences between model fits where monotonic models fit better but not where tuned models fit better. This could indeed result from the larger receptive fields in later visual field maps which are also the visual field maps that are dominated by tuned responses, but even in those later visual field maps we generally see effects of eccentricity on the fits of both models, just not the difference between their fits.

Minor comments:

1) I wanted to watch Supplementary Movie 1 to see the experiment, but the movie is not accessible.

Thank you for pointing this out. Supplementary Movie 1 was included in the original submission, but we accidentally missed it from the revision. This should be accessible in the new revision.

2) I agree with the other reviewers that the figures are somewhat overbearing and contain too many panels. To aid the readers, I suggest to move all the yellow curves showing differences between monotonic and tuned conditions from Figures 3b, 5b, 7b to the supplementary figures as the differences are already evident in the main plots by comparing the red (tuned) and blue (monotonic) data.

We agree that there are a plenty of Figure panels, particularly with the addition of all response model parameters in Figure 4. However, we feel that Figures 3b, 5b and 7b show effects that are important for our conclusions and are far less clear from looking at Figures 3a, 5a and 7a alone. Specifically, Figures 3b and 7b both show the gradual transition from monotonic to tuned models across the visual field map and timing map hierarchies, in both cases by showing the gradual change in the difference of the fits of the two response models. This is less clear in Figures 3a and 7a because the different maps vary considerably in variance explained. Furthermore, we use median differences here, and the median of the differences (shown in Figures 3b and 7b) is not the same as the difference of the medians (shown in Figure 3a and 7a), so these do not show the same data. We feel that removing the two panels showing the data behind our major conclusions most clearly would not help the reader. Regarding Figure 5b, we hope that our reply to an earlier point makes clear that the major result shown here is that the progressions of the difference in model fits with eccentricity show clear changes when the difference is negative but not when the difference is positive. This panel importantly highlights that it is the difference in model fits that we are looking at here, rather than the eccentricity progression of the fits of each response model.

Reviewer #2 (Remarks to the Author):

The authors sufficiently addressed my concerns and I acknowledge that the manuscript is greatly improved. I feel this manuscript is now ready to publish.

Reviewer #3 (Remarks to the Author):

The authors have addressed all my previous points conclusively and I commend them on the hard work. The revised manuscript does a great job communicating this fairly complex piece of research. The validation work is a great addition, also helping to deal with the question about the appropriate threshold.

Lastly, I thank the authors for taking seriously my comment about data availability. I understand (and can relate to) the issues with privacy and data sovereignty - so an explanation for why brain images cannot be shared publicly and a clear detail of the conditions for sharing data is entirely sufficient in my book. So I commend the authors for going the extra mile here in sharing the processed data that should allow interested parties to further interrogate the results without sacrificing data protection requirements.

**Sam Schwarzkopf
UoA**

REVIEWERS' COMMENTS

Reviewer #1 (Remarks to the Author):

The authors have improved the manuscript by better justifying the experimental decisions, adding data from additional ventral ROIs, and extending the discussion to make it more nuanced.

They have addressed my remaining comments and I believe this manuscript is acceptable for publication.

Reviewer #1 (Remarks to the Author):

The authors have improved the manuscript by better justifying the experimental decisions, adding data from additional ventral ROIs, and extending the discussion to make it more nuanced. They have addressed my remaining comments and I believe this manuscript is acceptable for publication.

We thank the reviewer for the positive appraisal of the changes we made in the previous revision.